**Atmospheric CO$_2$ estimates for the Miocene to Pleistocene based on foraminiferal δ$^{11}$B at Ocean**
**Drilling Program Sites 806 and 807 in the Western Equatorial Pacific**
Maxence Guillermic[1,2], Sambuddha Misra[3,4], Robert Eagle[1,2], Aradhna Tripati[1,2]
[1]Department of Atmospheric and Oceanic Sciences, Department of Earth, Planetary, and Space
Sciences, Center for Diverse Leadership in Science, Institute of the Environment and Sustainability,
University of California – Los Angeles, Los Angeles, CA 90095 USA
[2]Laboratoire Géosciences Océan UMR6538, UBO, Institut Universitaire Européen de la Mer, Rue
Dumont d'Urville, 29280, Plouzané, France
[3] Indian Institute of Science, Centre for Earth Sciences, Bengaluru, Karnataka 560012, India
[4] The Godwin Laboratory for Palaeoclimate Research, Department of Earth Sciences, University of
Cambridge, UK
*Correspondence to:* Maxence Guillermic (maxence.guillermic@gmail.com) and Aradhna Tripati
(atripati@g.ucla.edu)
**ABSTRACT**
Constraints on the evolution of atmospheric $CO_2$ levels throughout Earth's history are foundational to
our understanding of past variations in climate. Despite considerable effort, records vary in their
temporal and spatial coverage and estimates of past $CO_2$ levels do not always converge, and therefore
new records and proxies are valuable. Here we reconstruct atmospheric $CO_2$ values across major
climate transitions over the past 16 million years using the boron isotopic composition ($\delta^{11}B$) of
planktic foraminifera from 89 samples obtained from two sites in the West Pacific Warm Pool, Ocean
Drilling Program (ODP) Sites 806 and 807 measured using high-precision multi-collector inductively-
coupled plasma mass spectrometry. We compare our results to published data from ODP Site 872,
also in the Western Equatorial Pacific, that goes back to 22 million years ago. These sites are in a
region that today is near equilibrium with the atmosphere and are thought to have been in equilibrium
with the atmosphere for the interval studied. We show that $\delta^{11}B$ data from this region are consistent
with other boron-based studies. The data show evidence for elevated $pCO_2$ during the Middle
Miocene and Early to Middle Pliocene, and reductions in $pCO_2$ of ~200 ppm during the Middle
Miocene Climate Transition, ~250 ppm during Pliocene Glacial Intensification, and ~50 ppm during
the Mid-Pleistocene Climate Transition. During the Mid-Pleistocene Transition there is a minimum
$pCO_2$ at MIS 30. Our results are consistent with a coupling between $pCO_2$, temperature and ice sheet
expansion from the Miocene to the late Quaternary.
**Highlights**
In this study, we reconstruct atmospheric $pCO_2$ using $\delta^{11}B$ data from ODP Sites 806 and 807 and
compare them with ice core data. We therefore apply the same framework to older samples from these
sites to create a long-term pH and $pCO_2$ reconstruction for the past 16 million years, including
recalculating $pCO_2$ for ODP Site 872 from 17 to 22 million years ago. We find major increases in
surface water pH and decreases in atmospheric $pCO_2$ were associated with decreased temperature in
the Western Equatorial Pacific, including associated with major episodes of ice sheet expansion in the
high latitudes, providing more robust quantitative constraints on the past coupling between $pCO_2$,
temperature, and cryosphere stability.

**Keywords**
Boron isotopes, $CO_2$, ODP Site 806, ODP Site 807, Miocene, climate

## 1. Introduction

Due to concerns about the long-term consequences of anthropogenic emissions and associated climate change (IPCC, 2014, 2018), efforts have been made to quantify past atmospheric $CO_2$ and examine past relationships between $CO_2$ and temperature. Such data are not only critical for constraining Earth-system sensitivity (Lea, 2004; Lunt et al., 2010; Pagani et al., 2010; Hansen et al., 2012, 2013, Foster and Rohling, 2013; Schmittner et al., 2011; Tierney et al., 2020), but are also of broad interest to contextualize the evolution of climate and geological systems through Earth's history (Tripati et al., 2011; Foster et al., 2017; Tripati and Darby, 2018). However, discrepancies between proxy reconstructions still exist, including for major climate transitions of the Cenozoic. In particular, there remains a pressing need for robust and higher-resolution atmospheric $CO_2$ records.

High-resolution and direct determinations of atmospheric $CO_2$ are available for the last 800 kyr through analysis of air bubbles extracted from ice-cores, but these records are limited to the availability of cores (Petit et al., 1999; Siegenthaler et al., 2005; Lüthi et al., 2008; Bereiter et al., 2015). A window into older atmospheric $CO_2$ levels comes from 1 million-year-old blue ice (Higgins et al., 2015) and from a second snapshot from 1.5 Ma (Yan et al., 2019). Most reconstructions of $CO_2$ prior to 800 ka are based on indirect terrestrial and marine proxies. Stomata indices for fossil leaves (Van der Burgh, 1993; Royer , 2001), carbon isotope ratios ($\delta^{13}C$) of paleosols (Retallak et al., 2009), $\delta^{13}C$ of alkenones (Pagani et al., 2005; Zhang et al., 2013), B/Ca ratios of surface-dwelling foraminifera (Yu and Hönisch, 2007; Foster, 2008; Tripati et al., 2009, 2011), and boron isotope ratios ($\delta^{11}B$) of surface-dwelling foraminifera (e.g. Pearson and Palmer., 2000; Hönisch and Hemming, 2009; Seki et al., 2010; Bartoli et al., 2011; Foster, 2008, 2012; Badger et al., 2013; Foster and Sexton, 2014; Greenop et al., 2014; Martinez-Boti et al., 2015a; Chalk et al., 2017; Sosdian et al., 2018; Dyez et al., 2018; deLaVega et al., 2020; Greenop et al., 2021; Rae et al., 2021; Raitzsch et al., 2021; Shuttleworth et al., 2021) have been used to estimate atmospheric $CO_2$.

Each of the above proxy methods has sources of systematic errors that we do not attempt to exhaustively document as they have been discussed in-depth elsewhere (e.g., Pagani et al., 2005; Tripati et al., 2011; Guillermic et al., 2020). However, we note that significant developments in the boron-based proxies include improvements to the accuracy and precision of measurements using multi-collector inductively coupled mass spectrometry (MC-ICP-MS) compared to early work with negative thermal ionization mass spectrometry (N-TIMS), where there were large instrumental mass fractionations and challenges with laboratory intercomparison (Foster et al., 2013; Farmer et al., 2016; Aggarwal and You, 2017). There was also the realization that temperature-dependent $K_D$ and B/Ca sensitivities reported from sediment trap, core-top, and downcore studies (Yu and Hönisch, 2007; Foster et al., 2008; Tripati et al., 2009, 2011; Babila et al., 2010; Osborne et al., 2020) differ from inferences from foraminiferal culture experiments (Allen et al., 2011, 2012) and inorganic calcite (Mavromatis et al., 2015) which complicates the use of the B/Ca proxy, although this type of

discrepancy has also been observed with other elemental proxies (e.g., Mg/Ca). Such differences may
be due to differences in growth rates (Sadekov et al., 2014), ontogenetic changes, a correlation in the
field between temperature and other hydrographic variables that obscure robust statistical
determination of parameter relationships, culture conditions resulting in organisms being stressed,
and/or other factors.

The marine $CO_2$ proxy that appears to be subject to the fewest systematic uncertainties, based
on our current understanding, is the boron isotopic composition ($\delta^{11}B$) of planktic foraminifera as
measured using MC-ICP-MS and N-TIMS (Hain et al., 2018). This proxy provides constraints on
seawater pH, if temperature, salinity, seawater $\delta^{11}B$, and the appropriate mono-specific calibration
between $\delta^{11}B_{carbonate}$ and $\delta^{11}B_{borate}$ are constrained (Pearson and Palmer, 2000; Foster et al., 2008;
Sosdian et al., 2018; Raitzsch et al., 2018; Guillermic et al., 2020). Seawater pH can be used to
calculate seawater $pCO_2$ if there are constraints on a second parameter of the carbonate system (e.g.
alkalinity, DIC). Atmospheric $pCO_2$ can then be constrained if the site being examined is in air-sea
$CO_2$ equilibrium or if the disequilibrium is known and stable through time.

However, there are relatively few studies generating high-precision boron-based records over
major climate transitions in the Cenozoic using recent analytical methods and that incorporate our
current understanding of the proxy (e.g., Greenop et al., 2014; Martinez-Boti et al., 2015b; Chalk et
al., 2017; Dyez et al., 2018; Sosdian et al., 2018; de la Vega et al., 2020; Rae et al., 2021; Raitzsch et
al., 2021). Furthermore, of the existing studies using boron-based proxies, an additional uncertainty
frequently exists, namely the short time interval of study (e.g., emphasizing on a climate transition)
(Martinez-Boti et al., 2015b; Chalk et al., 2017) and whether the study sites remain in air-sea $CO_2$
equilibrium with the atmosphere (Martinez et al., 2015a). Moreover, although estimation of
atmospheric $pCO_2$ from seawater pH using this proxy is relatively straightforward, reconstructions are
still impacted by uncertainties including the lack of robust constraints on a second parameter of the
carbonate system, and our limited understanding of secular variations in the $\delta^{11}B$ of seawater (Tripati
et al., 2011; Greenop et al., 2017; Sosdian et al., 2018; Rae et al., 2021).

Therefore, to provide additional constraints on the evolution of atmospheric $pCO_2$ from the
Miocene through Pleistocene, we developed new records from the western tropical Pacific. We use
foraminiferal $\delta^{11}B$ and trace elements in the planktic foraminiferal species *Trilobus sacculifer* and
*Globigerinoides ruber* to reconstruct past seawater pH and atmospheric $CO_2$ at Ocean Drilling
Program (ODP) Sites 806 and 807 in the Western Equatorial Pacific (WEP) over the last 16 million
years (Myr). The sites are located on the western border of the tropical Pacific Ocean, the largest open
ocean region on the globe, and the warmest open ocean region at present.

These two sites have been examined in other boron-based studies (Wara et al., 2003; Tripati et
al., 2009, 2011; Shankle et al., 2020), as has the region more broadly (Pearson and Palmer, 2000;
Sosdian et al., 2018), because it is understood to be in equilibrium with the atmosphere and have

relative stable hydrography. The region experiences equatorial divergence but is not strongly affected by upwelling and has a current estimated annual air-sea $CO_2$ difference of +28 ppmv (Takahashi et al., 2014). The pre-industrial air-sea $CO_2$ difference is calculated to be +16 ppm, (GLODAP database corrected from anthropogenic inputs), with a value of 298 ppm, compared to the ice core value of 282 ppm at 1.08 ka. This $pCO_2$ difference is similar to our $pCO_2$ uncertainty (an average of ~17 ppm (2 SD) for the youngest samples). If trade winds were much stronger, and equatorial divergence greater, than this could drive some disequilibrium in the past. However, a few lines of evidence suggest the region was in quasi-equilibrium in the past: 1) zonal temperatures are at a maximum in pre-industrial times and during the Pleistocene, and we are able to reconstruct atmospheric $pCO_2$ values from the ice cores, 2) temperature proxies indicate the region is relatively stable with respect to temperature compared to other parts of the ocean, and also indicate a weak and stable zonal temperature gradient during the Miocene and Pliocene which would support air-sea stable conditions and air-sea (dis-)equilibrium conditions (e.g., Nathan and Leckie, 2009; Zhang et al. 2014; Liu et al., 2019).

Thus, this study builds on prior low-resolution reconstructions for these sites (Wara et al., 2003; Tripati et al., 2009, 2011; Shankle et al., 2020), Site 872 in the tropical Pacific (Sosdian et al., 2018), and other published boron isotope work, to provide additional data to constrain past seawater pH and $pCO_2$ for the WEP using MC-ICP-MS, thereby providing a new perspective on reconstructing past atmospheric $CO_2$ via marine sediment archives. We explore various constraints on the second carbonate system parameter using a number of different scenarios, following on the systematic work done by Tripati et al. (2009) and (2011) for B/Ca. We interpret these data using recent constraints on seawater $\delta^{11}B$ (Lemarchand et al., 2000; Raitzsch and Hönisch, 2013; Greenop et al., 2017). For temperature estimation, we utilize a multi-variable model for Mg/Ca correcting for salinity, pH and seawater Mg/Ca (Gray and Evans, 2019), that builds on prior work with clumped isotopes in planktic foraminifera for Site 806 and other WEP sites demonstrating that for the Last Glacial Maximum to recent times, salinity-corrected Mg/Ca values are needed to yield convergent estimates of mixed-layer temperatures (Tripati et al., 2014).

## 2. Materials and Methods

Below we describe site locations, analytical methods used, and principal figures. The supplemental methods section describes screening for potential contamination, equations used for calculations, and error propagation.

### 2.1 Site locations

Samples are from three ODP holes recovered during Leg 130 in the WEP (Fig. 1, Table 1): Hole 806A (0°19.140'N, 159°21.660'E, 2520.7 m water depth), Hole 806B (0°19.110'N, 159°21.660'E, 2519.9 m water depth), and Hole 807A (3°36.420'N, 156°37.500'E, 2803.8 m water

depth) (Berger et al., 1993). Sites 806 and 807 are not likely to have experienced major tectonic
changes over the last 20 million years.

**2.2 Preservation**
Microfossils in sediments at these sites, as with any sedimentary sequences, have the potential
to be influenced by diagenesis. Despite evidence of authigenic carbonate formation, recent modeling
work concluded the influence of dissolution and reprecipitation at Sites 806 and 807 was relatively
minor (Mitnik et al., 2018). Prior work has also found minimal impacts on the B/Ca ratio of Pliocene
foraminifera from Site 806 (White and Ravelo, 2020), and on the Mg/Ca ratio of Miocene *D. altispera*
shells at Site 806 (Sosdian et al., 2020). The weight/shell ratio is commonly used to monitor
dissolution, and the only published record at Site 806 for the Pliocene does not show a trend
consistent with dissolution of *T. sacculifer* (Wara et al., 2005). We do note that while the "coccolith
size-free dissolution" index reported in Si and Rosenthal (2019) indicates higher dissolution rates in
the Miocene, their records were thought to be biased from changes in foraminifera assemblages as
discussed in White and Ravelo (2020).
To further assess the potential impact of dissolution in our geochemical data, the weight/shell
ratio was examined in our samples. The weight/shell data used to monitor dissolution does not exhibit
any trend within the interval studied consistent with dissolution. Absolute weights/shell are increasing
in the Miocene, which is not consistent with dissolution influencing the record (Fig. 2E).
Additionally, reconstructed pH and $pCO_2$ values also exhibit reasonable correspondence with the ice
core data. Downcore $\delta^{11}B$ values from Sites 806 and 807 are similar, despite evidence for higher
authigenic carbonate at Site 807 relative to Site 806 (Mitnik et al., 2018). Further, despite different
sedimentation rates, our $\delta^{11}B$ and Mg/Ca results are consistent between Sites 806 and 807, and with
data from Site 872 (Sosdian et al., 2018), which implies that diagenesis is not a primary driver of the
reconstructed trends. A comparison of raw data, and derived parameters, is shown in Figs. 2 and 7.

**2.3 Age models**
The age model for Site 806 from 0-1.35 Ma is based on Medina-Elizalde and Lea (2005);
calculated ages correspond well with ages from the Lisiecki and Raymo LR04 stack (Fig. 2A). The
fourth polynomial regression-based biostratigraphy from Lear et al. (2015) was used for the rest of the
record, following other work (Sosdian et al., 2020). Ages for Site 807 are based on published
biostratigraphy (Berger et al., 1993) with additional constraints placed by Zhang et al. (2007) for the
interval from 0-0.55 Ma. Benthic $\delta^{18}O$ values from Sites 806 and 807 show good correspondence for
the last 0.55 Myr, and the low-resolution benthic $\delta^{18}O$ record for Site 806 (Lear et al., 2003; 2015) is
consistent with the stack from Lisiecki and Raymo, (2005) for the period studied (Fig. 3).

**2.4 Species and trace element cleaning**

Samples were picked and cleaned to remove clays at UCLA (Los Angeles, CA) and the University of Western Brittany (Plouzané, France). 50-100 foraminifera shells were picked from the 300-400μm fraction size for *T. sacculifer* (w/o sacc) and from the 250-300 μm for *G. ruber* (white sensu stricto). Picked foraminifera were gently crushed, clays removed, and checked for coarse-grained silicates. Samples were then cleaned using a full reductive and oxidative cleaning protocol following Barker et al. (2003). A final leach step with 0.001N HCl was done prior dissolution in 1N HCl. Boron purification used a published microdistillation protocol (see Misra et al., 2014b, Guillermic et al., 2020 for more detailed methods).

**2.5 Chemical purification and geochemical analysis**

Chemical separation was performed in a boron-free clean lab at the University of Cambridge (Cambridge, UK). Calcium concentrations were measured on an ICP-AES ®Ultima 2 HORIBA at the Pôle Spectrometrie Océan (PSO), UMR6538 (Plouzané, France). Elemental ratios (e.g. X/Ca ratios) were analyzed on a Thermo Scientific ®Element XR HR-ICP-MS at the PSO, Ifremer (Plouzané, France). Boron isotopic measurements were carried out on a Thermo Scientific ®Neptune+ MC-ICP-MS equipped with $10^{13}$ Ohm resistor amplifiers (Lloyd et al., 2018) at the University of Cambridge (Cambridge, UK).

**2.6 Standards**

Variations in B isotope ratios are expressed in conventional delta (δ) notation with $\delta^{11}B$ values reported against the reference standard NIST SRM 951 (NIST, Gaithersburg, MD, USA):

$$\delta^{11}B \text{ (‰)} = 1000 \times \left( \frac{^{11}B/^{10}B_{Sample}}{^{11}B/^{10}B_{NIST\ SRM\ 951}} - 1 \right) \qquad \textbf{eq. 1}$$

Multiple analyses of external standards were performed to ensure data quality. For boron isotopic measurements, $JC_P$-1 (Geological Survey of Japan, Tsukuba, Japan, Gutjahr et al., 2020) was used as a carbonate standard, and NEP, a *Porites sp* coral from University of Western Australia and Australian National University was also used (McCulloch et al., 2014). A boron isotope liquid standard, ERM© AE121 (certified $\delta^{11}B = 19.9 \pm 0.6$ ‰, SD), was used to monitor reproducibility and drift during each session (Vogl and Rosner, 2012; Foster et al., 2013; Misra et al., 2014b). For trace elements, external reproducibility was determined using the consistency standard Cam-Wuellerstorfi (University of Cambridge) (Misra et al., 2014b).

## 2.7 Figures of Merit

### 2.7.1 $\delta^{11}B$ analyses

Samples measured for boron isotopes typically ranged in concentration from 10 ppb B (~5ng B) to 20 ppb B samples (~10ng B). Sensitivity was 10mV/ppb B (eg. 100mV for 10ppb B) in wet plasma at 50µl/min sample aspiration rate. The intensity of $^{11}B$ for a sample at 10 ppb B was typically $104 \pm 15$ mV (2 SD, typical session) and closely matched the $98 \pm 6$ mV (2 SD, typical session) of the standard. Procedural boron blanks ranged from 15 pg B to 65 pg B (contributed to less than 1 % of the sample signal). The acid blank during analyses was measured at $\leq 1$mV on $^{11}B$ (which also is $< 1$ % of the sample intensity), and no memory effect was seen within and across sessions.

External reproducibility was determined by analyzing the international standard JC$_P$-1 (Gutjahr et al., 2020) and a *Porites sp.* coral (NEP). The boron isotopic composition of JC$_P$-1 was measured at $24.06 \pm 0.20$ ‰ (2 SD, n=6) within error of published values of $24.37 \pm 0.32$ ‰, $24.11 \pm 0.43$ ‰ and $24.42 \pm 0.28$ ‰ from Holcomb et al. (2015), Farmer et al. (2016) and Sutton et al. (2018), respectively. Average values are $\delta^{11}B_{NEP} = 25.72 \pm 0.79$ ‰ (2 SD, n=31) determined over 13 different analytical sessions, with each number representing a separately processed sample from this study. These results are within error of published values of $26.20 \pm 0.88$ ‰ (2 SD, n = 27) and $25.80 \pm 0.89$ ‰ (2 SD, n = 6), from Holcomb et al. (2015) and Sutton et al. (2018), respectively. Data are reported in Supplementary Table B.

### 2.7.2 X/Ca analyses

Trace element (TE) analyses were conducted at a Ca concentration of either 10 or 30 ppm. Typical blanks for a 30 ppm Ca session were: $^7Li < 2$ %, $^{11}B < 7$ %, $^{25}Mg < 0.2$ % and $^{43}Ca < 0.02$ %. Additionally, blanks for a 10 ppm Ca session were: $^7Li < 2.5$ %, $^{11}B < 10$ %, $^{25}Mg < 0.4$ % and $^{43}Ca < 0.05$ %. Analytical uncertainty of a single measurement was calculated from the reproducibility of the CamWuellestorfi standard: 0.6 µmol/mol for Li/Ca, 8 µmol/mol for B/Ca and 0.02 mmol/mol for Mg/Ca (2 SD, n=48). Data are reported in Supplementary Table B.

## 2.8 Calculations

Detailed calculations can be found in the supplemental materials. Briefly, Mg/Ca was used to reconstruct sea surface temperature (SST) using the framework from Gray and Evans. (2019) correcting for influences of pH, salinity, and secular variation in seawater Mg/Ca. $\delta^{11}B_{carbonate}$ was corrected using an empirical $\delta^{11}B_{carbonate}$-weight/shell ratio relationship. $\delta^{11}B_{borate}$ was determined using species dependent sensitivities of $\delta^{11}B_{carbonate}$ to $\delta^{11}B_{borate}$ (Guillermic et al., 2020). pH was calculated using $\delta^{11}B_{borate}$ with different scenarios of secular seawater $\delta^{11}B$ changes (Lemarchand et al., 2002; Raitzsch and Hönisch, 2013; Greenop et al., 2017). $pCO_2$ was reconstructed using pH based $\delta^{11}B_{carbonate}$ and different scenarios of alkalinity (Tyrell and Zeebe, 2004; Ridgwell and Zeebe, 2005;

Caves et al. 2016 and Rae et al. 2021). Further details including equations are provided in the
Supplement.
**3. Results and discussion**
**3.1 Geochemical results**
Geochemical data used in this study are presented in Figure 2. Mg/Ca data (Fig. 2C) are
consistent with previously published Mg/Ca values for Site 806 on *T. sacculifer* (Wara et al., 2005;
Tripati et al., 2009; Nathan and Leckie, 2009). Although the record we generated does not overlap
with Site 872, they are 1 Myr apart (15.7 and 16.7 Ma); there is a good correspondence between our
Mg/Ca data and the published Mg/Ca record from *T. trilobus* at Site 872 (Sosdian et al., 2018). Mg/Ca
from a different species, *D. altispira* (Sosdian et al., 2020), is also plotted with an offset, for
comparison.
Comparison with Site 872 data that is part of the compilation from Sosdian et al. (2018)
shows that their $\delta^{11}B$ data are in line with our dataset (Figure 2B), and all sites examined in the WEP
(Sites 806, 807, and 872) are above the lysocline (Kroenke et al. 1991). The $\delta^{11}B$ data for *T. sacculifer*
exhibit a significant decrease (4.2 ‰) from the Miocene to present. Figure 2B also compares the $\delta^{11}B$
data used in this study with published data from other sites and shows that raw $\delta^{11}B$ data for the WEP
can be lower than values for other regions.
**3.2 Reproducing pCO$_2$ from ice cores**
We sought to assess if there is evidence for air-sea equilibrium or disequilibrium in the WEP
during the large amplitude late Pleistocene glacial/interglacial cycles, in order to validate our
approach. We reconstructed pCO$_2$ for the last 800 kyr (n=16, Fig. 3). For the last 800 kyr,
reconstructed pCO$_2$ values for Sites 806 and 807 are in the range from ice cores (Fig. 3, Petit et al.,
1999, Siegenthaler et al., 2005, Lüthi et al., 2008; compilation from Bereiter et al., 2015). The two
critical diagnostics we used for method validation are: 1) that the $\delta^{11}B$-based reconstruction of pCO$_2$
is consistent with ice core atmospheric CO$_2$ and 2) the boron-based reconstruction empirically
reproduces interglacial-glacial amplitudes from ice cores. Fig. 3B shows that both of these criteria are
met despite large scatter. We also created a crossplot comparing these two independent constraints on
pCO$_2$ (Fig. 3C). Two regressions between ice core pCO$_2$ and boron-based pCO$_2$ are shown, a simple
linear regression (grey line) and a Deming regression that takes into account error in variables (blue
line). Bootstrapping was used to calculate uncertainties in the regression models (n=1000, Figure 3C,
Table S6). While slopes and intercepts are not statistically different from a 1:1 line, the regressions do
not reach a high significance level (p=0.25); boosting the resolution of the record could help provide
better constraints for this type of comparison. No significant difference in variability was observed at
either site. The age models for the sites are based on comparisons of the benthic $\delta^{18}O$ records for both

Sites 806 and 807 (Fig. 3A, Zhang et al., 2007; Lear et al., 2003; Lear et al., 2015) to the published isotopic stack (Lisiecki and Raymo, 2004).

We also note that reconstructed $pCO_2$ uncertainties (both accuracy and precision) could potentially arise from Mg/Ca-derived estimates of temperature; these uncertainties could be reduced using independent temperature proxies for the WEP such as clumped isotope thermometry (Tripati et al., 2010; 2014), a technique that is not sensitive to the same sources of error as Mg/Ca thermometry, and therefore is an area planned for future work. Other sources of uncertainty that have a larger effect on $pCO_2$ calculations are the weight/shell correction, while the TA and seawater boron isotope composition have a minor effect over this time interval.

Between MIS 7 and 6, our reconstructions exhibit a decrease in temperature ($\Delta T$) of 1.2 °C, an increase in pH ($\Delta pH$) of 0.08 and a decrease in $pCO_2$ ($\Delta pCO_2$) of 58 ppm. Between stage 3 and 1, we observed an increase of temperature of 2.0 °C, a decrease of pH of 0.13 and an increase in $pCO_2$ of 76 ppm. We also compare results with recent reconstructions in Figs. S1 and S2 (Sosdian et al., 2018; Rae et al., 2021). These results highlight that we are able to reproduce the range of atmospheric $pCO_2$ in the ice core record, and reproduce the amplitude of changes between transitions, with uncertainties typical for this type of work (Hönisch et al., 2019).

**3.3 Sea surface temperature in the WEP**

Mg/Ca data are consistent at Site 806 (Wara et al., 2005; Tripati et al., 2009, 2011; Nathan and Leckie, 2009) and Site 872 (Sosdian et al., 2018) in the WEP. The Mg/Ca in *T. sacculifer* has to date not shown a pH dependency (Gray and Evans, 2019) but Mg/Ca of *G. ruber* does and was therefore corrected from this effect (see supplemental material). Data for both species were corrected from salinity and seawater Mg/Ca changes. Mg/Ca-temperatures for Site 872 were reconstructed using published data and the same framework we use here and are presented in Figure 4. Recalculated values for Site 872 are from *D. altispera*, with an offset applied relative to *T. sacculifer*, and show similar variations to our record for the MCO-MMCT periods (Sosdian et al., 2020). Temperatures from $Tex_{86}$ and $U^{K'}_{37}$ are plotted for comparison but those records are limited to the last 12 and 5 Myrs, respectively (Zhang et al., 2014).

The Mg/Ca data support high temperatures of 35.2 ± 1.3 °C (2SD, n=11) for the early Miocene until the MMCT, with relatively small (ca. 1°C) change into the MCO, and larger changes out of the MCO. Similarly warm SST for the MCO were reconstructed in the North Atlantic at Site 608 from $Tex_{86}$ (Super et al., 2018). Despite a gap in our compilation from 11.5 to 9.5 Ma, there is a SST decrease of ~6 °C from the MCO to ~7 Ma where temperatures similar to present day values are observed. A decline in temperature during the MMCT is coincident with the timing of a constriction of the Indonesian Seaway, the pre-closure of the trans-equatorial circulation and subsequent formation of a proto-warm pool (Nathan and Leckie, 2009; Sosdian et al., 2020). From 12 to 7 Ma, the Mg/Ca-

SST record diverges from $Tex_{86}$ and $U^{K'}_{37}$-based reconstructions, with higher temperatures. At the
same time, a record for the North Atlantic showed a decrease of ~10 °C from the MCO to ~9 Ma
(Super et al., 2018). From 7 Ma to present, the record from multiple proxies – Mg/Ca, $Tex_{86,}$ and
$U^{K'}_{37}$, in the WEP agree.

**3.4 Scenarios of seawater $\delta^{11}B$ and alkalinity used for $pCO_2$ reconstructions**

Figures 5 and 6 show the different histories of seawater $\delta^{11}B$ and alkalinity used in our

calculations, respectively. Details of calculations are provided in the Supplemental methods.
Following the approach of Tripati et al. (2009, 2014) and recent literature (Sosdian et al., 2018; Rae et
al., 2021), we explored multiple scenarios for the evolution of seawater boron geochemistry (Fig. 5)
and alkalinity for calculations of $pCO_2$ (Figs. 6, S1 and S2). During the interval overlapping with the
ice core record, we observe that the choice of model used does not make a significant difference in
reconstructed values. During earlier time intervals, we see there is a greater divergence, reflecting
larger uncertainties in seawater $\delta^{11}B$ and alkalinity further back in Earth history.

Prior to 10 Ma and during the early Pliocene (~4.5 to 3.5 Ma), calculations of pCO2 diverge

from published values largely because of the different assumptions each study has used for past
seawater $\delta^{11}B$ (Fig. 5). However, we find that when the uncertainty in reconstructed pH is fully
propagated, the differences in reconstructed pH values calculated using each of the $\delta^{11}B_{seawater}$ histories
is not significantly different (Fig. 5 and 6; see also Hönisch et al., 2019). In contrast to the results
from Greenop et al. (2017), the record from Raitzsch and Hönisch, (2013) exhibits substantial
variations on shorter timescales. Such variability is a challenge to reconcile with the Li isotope record
of Misra and Froelich, (2012), given that Li has a shorter residence time than boron while having
similar sources and sinks. For the remainder of this study, we use the $\delta^{11}B_{seawater}$ history from Greenop
et al. (2017) because it is in good agreement with seawater $\delta^7Li$ (Misra and Froelich, 2012). The
recent calculations of seawater pH (Sosdian et al., 2018; Rae et al., 2021) agree with values from our
study when uncertainties are taken into account (Fig. 5).

The four alkalinity models used in this study diverge prior to 9 Ma, with a maximum

difference at ~13 Ma that is also reflected in reconstructed pCO2 values (Fig. 6). However, all four
models yield pCO2 estimates that are within error of each other when the full uncertainty is
considered. Uncertainty in the evolution of seawater alkalinity and seawater $\delta^{11}B$ leads to differences
in the absolute values of pCO2 reconstructed (Fig. S2), and a divergence in pCO2 values reconstructed
that is largest in the Miocene. The two scenarios that produce the highest divergence in values are
those calculated using constant alkalinity relative to those calculated using values from McCaves et al.
(2016), with a maximum difference at 15.06 Ma of up to 250 ppm CO2, and with the latter model
producing lower values (Figs. 6B and 6E). Thus, for the MCO, alkalinity is a critical parameter in
calculations of absolute pCO2 values. For the Miocene and earlier intervals, improved constraints on

past secular variations of seawater $\delta^{11}B$ and alkalinity will yield more accurate reconstructions of $pCO_2$.

For the remainder of this paper, we use the model of Caves et al. (2016) to estimate alkalinity and $\delta^{11}B_{seawater}$ determined by Greenop et al. (2017) (e.g. Fig. 6E). We note that two recent syntheses of boron isotope data have been published and compare our results to these findings (Figs. 8 and S2). Sosdian et al. (2018) reports values that are in line with our results in the Miocene but their study does not replicate results from ice cores. Rae et al. (2021) presents reconstructed values that are higher in the Miocene, due to the utilization of different scenarios of seawater $\delta^{11}B$ and alkalinity compared to this work.

### 3.5 Time intervals

### 3.5.1 Miocene

The study of Miocene climate is thought to provide insights into drivers and impacts of global warming and melting of polar ice (Flower and Kennett, 1994). The Miocene epoch (23-5.3 Ma) is characterized by a warm interval, the Miocene Climate Optimum (~17-14.7 Ma - MCO), and an abrupt cooling during the Middle Miocene Climate Transition (~14-13 Ma – MMCT) that led to the expansion of ice on Antarctica and Greenland. Climate modeling supports a role for decreasing $CO_2$ in this transition (DeConto and Pollard, 2003). However, reconstructions for the Miocene are still relatively limited (Sosdian et al., 2018; Rae et al., 2021; Raitzsch et al., 2021). Boron isotope and alkenone-based $pCO_2$ reconstructions support higher $pCO_2$ during the MCO and a decrease over the MMCT (Sosdian et al. 2018; Stoll et al., 2019), consistent with what was previously inferred from B/Ca (Tripati et al., 2009, 2011; Sosdian et al., 2020).

We applied the same framework we used for calculations at Sites 806 and 807 to published boron isotope data from Site 872 (Sosdian et al., 2018) in order to extend the WEP record to the early Miocene (Figs. 7, 8). The Miocene data between Sites 806 and 872 do not overlap as both are low in resolution, but do show excellent correspondence in their trends in $\delta^{11}B$ and reconstructed pH. For example, the closest datapoints in time at the two sites are at 15.6 Ma at Site 806 with a $\delta^{11}B = 14.47\pm$ 0.21 ‰, and at 16.7 Ma at Site 872, with a $\delta^{11}B = 15.12\pm 0.25$ ‰. The pH values we reconstruct are within error of published estimates from Site 872 (Sosdian et al. 2018, Figs. 7D and 8D). Collectively, these data suggest that the early Miocene WEP was characterized by a mixed-layer pH of 8.1 ± 0.1 (2 SD, n=4) between 19.4 and 21.8 Ma, which decreased to reach a minimum during the MCO of 7.7 ($\pm^{0.11}_{0.14}$).

Given the sensitivity in absolute $pCO_2$ to assumptions about the second carbonate system parameter, a few scenarios were explored for the combined 806/807/872 reconstructed pH values. For all alkalinity scenarios we used, reconstructed $pCO_2$ shows an increase from the Early Miocene to the MCO, with the highest values in the MCO. Recalculated $pCO_2$ for Site 872 between 19.4 and 21.8 Ma

is 232 ± 92 ppm (2 SD, n=4), lower but within error of the ones presented in Sosdian et al. (2018) and
also within error of a constant alkalinity scenario (Fig 8D). The main difference between our
calculations and published reconstructions occurs between 19.4 and 21.8 Ma, when the same $\delta^{11}$B
data for Site 872 from Sosdian et al. (2018) recalculated in Rae et al. (2021) yield higher $pCO_2$, with
an average value of 591 ± 238 ppm (2 SD, n=4) because of the different assumptions used in their
calculations. This difference is important because the assumptions from Rae et al. (2021) would imply
a relatively high and stable $pCO_2$ from the early Miocene to MCO (Fig. S2), which would imply a
decoupling between $pCO_2$ and temperature with no $pCO_2$ change during an interval of decreasing
benthic $\delta^{18}$O. However, our reconstructed $pCO_2$ data increase towards the MCO is in line with the
observed benthic $\delta^{18}$O decrease and $\delta^{13}$C increase and suggest a coupling between temperature and
$pCO_2$ over this period. This highlights the critical need for the use of a common set of assumptions for
studies. Assumptions may vary between studies depending of the timescales studied, but a common
framework is needed. In addition, further constraints on the second carbonate system parameter and
on secular changes in seawater $\delta^{11}$B will reduce uncertainties in reconstructed $pCO_2$, with improved
precision.
The highest $pCO_2$ values we reconstruct are found during the MCO (Fig. 6E). For the MCO,
our estimates are 511 ± 201 ppm (2 SD, n=3, Table 2). The middle Miocene values we reconstruct are
in line with previous studies (Greenop et al., 2014; Sosdian et al., 2018). Published $\delta^{11}$B-based
reconstructions also support higher $pCO_2$ for the MCO of ~350-400 ppm (Foster et al., 2012) or 300-
500 ppm (Greenop et al., 2014) that was recalculated by Sosdian et al. (2018) to be ~470-630 ppm
depending on the model of $\delta^{11}B_{seawater}$ chosen. During the MCO relative maxima in $pCO_2$, our data
support very warm sea surface temperatures in the WEP (35.6 ± 0.6 °C 2SD, n=3; Fig. 8C), that
merits further examination in future studies. In fact, the highest temperatures recorded in our samples
occur when there is a minimum in the global composite record of $\delta^{18}$O of benthic foraminifera
(Zachos et al., 2001, 2008; Tripati and Darby, 2018).
At the end of the MMCT, we find evidence for changes in $p$CO$_2$ and temperature in the WEP
(Fig. 8). From 13.5 to 12.7 Ma, we reconstruct an increase of pH of ~0.21 and a major decrease of
$pCO_2$ of ~215 ppm during an interval highlighted by Flower and Kennett, (1996), who observed
changes in $\delta^{18}$O indicative of rapid East Antarctic Ice Sheet growth and enhanced organic carbon
burial with a maximum $\delta^{13}$C reached at ~13.6 Ma (Shevenell et al., 2004; Holbourn et al., 2007). As
discussed in section 3.4 the alkalinity model used for the calculations have an important impact during
the Miocene which is likely responsible for the different absolute $pCO_2$ values over the MCO. In
comparison, a scenario of constant alkalinity would lead to a $pCO_2$ during the MCO of 714 ± 313 ppm
(2 SD, n=3) and a decrease of ~540 ppm during the MMCT. Both those reconstructions could
simulate the large-scale advance and retreat of Antarctic ice with such low $pCO_2$ values (Gasson et al.,
2016). At the same time, we find evidence for a decline in SST of 3.4 °C to minimum values of 33.3
°C. The synchronous shifts in the $\delta^{13}C$ and $\delta^{18}O$ of benthic foraminifera are consistent with increased
carbon burial during colder periods, thus feeding back into decreasing atmospheric $CO_2$, and
supporting the hypothesis that the drawdown of atmospheric $CO_2$ can in part, be explained by
enhanced export of organic carbon (Flower and Kennett, 1993, 1996). However, given the limited
sampling of this study, we are only able to resolve a $pCO_2$ decrease toward the end of the MMCT
(~13.5 Ma). The higher resolution $\delta^{11}B$-$pCO_2$ from Site 1092 for the MMCT (Raitzsch et al. 2021)
reports eccentricity-scale $pCO_2$ variability; the authors reported that low $pCO_2$ during eccentricity
maxima was consistent with an increase in weathering due to strengthened monsoonal circulation,
which would increase nutrient delivery and supporting higher productivity that in turn would impact
carbon drawdown and burial, in line with modeling from Ma et al. (2011).
The resolution of our data during the late Miocene is low, with a data gap from 12.5 to 9.2
Ma, and another gap between 6.5 and 5 Ma. We note the $pCO_2$ peak at ~9 Ma observed by Sosdian et
al. (2018) is not seen in our record although this is likely due to the low resolution of our dataset.
Between 9.5 and 7.1 Ma we find evidence for a decrease in atmospheric $CO_2$ of 100 ppm associated
with a decrease in temperature of 1.3 °C. $pCO_2$ estimates derived from alkenones for Site 1088
(Tanner et al., 2020) do not show the same trend as boron-based reconstructions from the WEP or
other regions (Figure 6), which might be due to other controls on the alkenone proxy (Badger et al.,
2019). A recent publication from Raitzsch et al. (2021) reports a $\delta^{11}B$ reconstruction of $pCO_2$ that is
within error of other $\delta^{11}B$ isotope data from the Southern Ocean (Sosdian et al., 2018), although not
for the same period as Tanner et al. (2020). $pCO_2$ differences between our reconstruction and that of
Sosdian et al. (2018) and Raitzsch et al. (2021) (Fig. 8) likely reflect assumptions made for
calculations (of $\delta^{11}B$, TA) and the specific mono-specific calibrations used for each study, as well as
potential geographic differences in air-sea $pCO_2$. These differences do not invalidate the boron isotope
proxy but illustrate the impact that specific seawater parameters and calibrations can have on
reconstructed $pCO_2$ values, as well as potential inferences of air-sea disequilibrium.

**3.5.2 Pliocene**

Oxygen isotope data from a global benthic foraminiferal stack show that the Pliocene epoch
(5.3-2.6 Ma) was initially characterized by warm conditions followed by the intensification of
glaciation that occurred in several steps, including during MIS M2 (3.312-3.264 Ma), followed by the
Middle Pliocene Warm Period (Lisiecki and Raymo, 2005). The Middle Pliocene Warm Period
(mPWP – 3.29-2.97 Ma) is considered a relevant geological analogue for future climate change given
~3°C warmer global temperatures and sea levels that were ~20 m higher than today (Dutton et al.,
2015; Haywood et al., 2016), and is a target for model intercomparison projects, for which accurate
paleo-atmospheric $pCO_2$ estimates are critical (Haywood et al., 2016).
We calculate high $pCO_2$ values of $419 \pm 119$ ppm (2 SD, n=3, Table 2) between 4.7 to 4.5 Ma

during the Early Pliocene warm interval (Figure 9). The $pCO_2$ data we report provide a higher data density for the Early Pliocene, and exhibit a trend that is in line with the reconstruction from Rae et al. (2021). Our data support values of 530 ± 110 ppm over the mPWP (2 SD, n = 4), higher than previously published data (Figs. 9, S2 and Table 2), although we acknowledge our low data density may not fully sample variability over this period. The similarity between our reconstructed values and those published for Site 871 in the Indian Ocean (Sosdian et al., 2018) suggests that changes in Indonesian through-flow do not induce substantial changes in air-sea exchange in the WEP.

The warmth and local $pCO_2$ maxima of the mPWP (mid-Pliocene Warm Period) was followed by a strong decrease of temperature in upwelling and high latitude regions from 3.3 to 2.7 Ma, coincident with glacial intensification in the Northern Hemisphere. This climate transition was hypothesized to be driven by the closure of the Panama seaway the opening of the high latitudes and subsequent modifications of oceanic circulation (Haug and Tiedemann, 1998). However, modeling from Lunt et al. (2008) supports an additional major role for $CO_2$ in the glaciation. $pCO_2$ thresholds have been proposed to explain the intensification of Northern Hemisphere Glaciation, with values proposed ranging from 280 ppm (DeConto et al., 2008) to 200 to 400 ppm (Koening et al., 2011).

The $pCO_2$ concentrations that we calculate indicates a reduction to 350 ppm by 2.7 Ma, ~280 ppm by 2.6 Ma, and ~210 ppm by 2.4 Ma, in several steps. These results support roughly a halving of $CO_2$ values when compared to values of ~530 ppm at 3.3 Ma. These values are consistent with the $pCO_2$ thresholds proposed by both DeConto et al. (2008) and Koening et al. (2011) for the intensification of Northern Hemisphere glaciation and the low atmospheric $CO_2$ (280 ppmv) scenario from Lunt et al. (2008). Mg/Ca SST decline from 30°C to 26°C, supporting an Earth System sensitivity of ~4°C/doubling of $CO_2$ over this range, although given uncertainties, higher values of ~6°C/doubling of $CO_2$ that have recently been proposed (Tierney et al., 2020) can not be excluded.

We speculate that associated with Pliocene glacial intensification, at 4.42, 3.45 and 2.67 Ma, it is possible that the declines in $CO_2$ and ice growth in turn drove substantial changes in pole-to-equator temperature gradients and winds, that in turn may have impacted iron cycling (Watson et al., 2000; Robinson et al., 2005; Martinez-Garcia et al., 2011), stratification (Toggweiler, 1999; Sigman et al., 2010), and other feedbacks that impact the amplitude of glacial/interglacial cycles and have been implicated as factors that could have contributed to Pliocene glacial intensification. Specifically, as the mean climate state of the planet became cooler, and glacial-interglacial cycles became larger in amplitude, enhanced windiness and dust transport and upwelling during glacials (Martinez-Boti et al., 2015b) may have enhanced iron fertilization and subsequent carbon export (Martinez-Garcia et al., 2011). While data resolution is limited, we speculate this could explain why glacial/interglacial amplitudes in WEP $pCO_2$ values decrease from the mPWP towards the Pleistocene, whereas variations in $\delta^{18}O$ are increasing – a speculation that could be tested with increased data resolution.

### 3.5.3 Pleistocene

During the Pleistocene (2.58-0.01 Ma), the climate system experienced a transition in glacial/interglacial (G/I) variability from low amplitude, higher frequency and obliquity-dominated oscillations (i.e., ~ 41 kyr) of the late Pliocene to the high amplitude, lower frequency (~100 kyr) cycles of the last 800 kyr. This transition is termed the Middle Pleistocene Transition (1.2-0.8 Ma – MPT). Questions have been raised about the role of atmospheric $CO_2$ during this transition, including using boron-based proxies (Hönisch et al., 2009; Tripati et al., 2011; Chalk et al., 2017). Previous boron isotope studies for ODP Sites 668 and 999 in the tropical Atlantic Ocean have suggested that a decline in atmospheric $CO_2$ did occur during glacial periods in the MPT, but not during interglacials (Hönisch et al., 2009; Chalk et al., 2017; Dyez et al., 2018).

Our $pCO_2$ concentrations for Sites 806/807 reported here are in good agreement with those determined from ice cores from the early Pleistocene (Yan et al., 2019, Figs. 9 and 10), and with the boron-derived $pCO_2$ from a recent compilation (Rae et al., 2021). Results for the MPT are broadly in the range of values reported by Hönisch et al. (2009) and Chalk et al. (2017). Although our data are relatively limited, we note they have greater resolution for the middle and later part of the transition than prior publications that have drawn conclusions about the MPT (Hönisch et al., 2009; Chalk et al., 2017; Dyez et al., 2018) (Fig. 10D) and therefore we explore their implications.

Taken alone, or when combined with the published data from Chalk et al. (2017) (that is also based on MC-ICPMS), our results support a possible reduction of both glacial and interglacial $pCO_2$ values. We also find evidence that during the MPT, glacial $pCO_2$ declined rapidly from 189 ±30 ppm at MIS 36 (Chalk et al., 2017) to reach a minimum of 170 ($\pm_{24}^{52}$) ppm during MIS 30. We note that $pCO_2$ concentrations are within error when uncertainty is fully propagated, and then remained relatively stable until the end of the MPT whereas interglacial $pCO_2$ values decrease gradually to reach post-MPT values.

In our record for the last 16 Myr, the lowest $pCO_2$ is recorded at MIS 30 during the MPT, with values of 164 ($\pm_{35}^{44}$) ppm, which supports an atmospheric $CO_2$ threshold that leads to large sheet generation. During this transition, the $pCO_2$ threshold needed to build sufficiently large ice sheets that were able to survive the critical orbital phase of rising obliquity to ultimately switch to a 100 kyr world, was likely reached at MIS 30, but a higher $pCO_2$ resolution of the MPT is needed for confirmation. The multiple feedbacks resulting from stable ice sheets (iron fertilization/productivity/changes in albedo/ changes in deep water formation) might have sustained larger mean global ice volumes over the subsequent 800 kyr. An asymmetrical decrease between $pCO_2$ values during interglacials relative to glacials, with glacials exhibiting the largest change across the MPT, would have led to increased sequestration of carbon during glacials in the 100 kyr world, as discussed by Chalk et al. (2017), with increased glacial dust input and iron fertilization.

## 3.6 Changes in volcanic activity and silicate weathering, and long-term $pCO_2$

On million-year timescales, atmospheric $CO_2$ is controlled by its input through mantle degassing in the form of sub-aerial and sub-aqueous volcanic activity and its removal by chemical weathering of continental silicate rocks. Over the last 16 Myr, two relative maxima in atmospheric $pCO_2$ are observed in our record, one during the MCO (at 15.67 Ma) and a second around the late Miocene/early Pliocene (beginning at 4.7 and 4.5 Ma) (Fig. 11), though the timing for the latter is not precise. The strong $pCO_2$ increase from the early Miocene to MCO occurs when there is increasing volcanic activity associated with the eruption of the Columbia River Flood Basalts (Hooper et al., 2002; Foster al. 2012; Kasbohm and Schoene, 2018), with recent geochronologic evidence published supporting higher eruption activity between 16.7 and 15.9 Ma (Kasbohm and Schoene, 2018) reinforcing the idea of an episodic $pCO_2$ increase during the MCO due to volcanic activity. Underestimation of net $CO_2$ outgassing from specific continental flood basalt eruption is possible, as both sub-aqueous and sub-aerial flood basalts, under right climatic conditions, are prone to enhanced chemical weathering. For example, the 4-5‰ drop in $\delta^7Li$ record at the Cretaceous–Paleogene (K-Pg) boundary (Misra and Froelich, 2012) is attributed to rapid quasi-congruent weathering of Deccan Traps (Rene et al. 2015) during their eruption. Courtillot and Rene (2003) estimate that about 50% of emitted $CO_2$, roughly equivalent to the amount emitted by the eruption of a million cubic kilometers of Deccan Traps, may be missing due to chemical and physical weathering. Additionally, the early Eocene (at ~50 Ma) 3-4‰ rise in seawater $\delta^7Li$ at a time where there is not significant uplift of the Himalayas (Misra & Froelich, 2012) is also attributed to incongruent weathering of previously erupted Deccan Trap basalts as the Indian subcontinent moved from arid mid-latitudes to the wet low latitudes (Kent and Muttoni, 2008). Thus, a significant part of the outgassed $CO_2$ can be consumed by chemical weathering of freshly erupted hot basalts (Courtillot et al., 2003). However, the congruency of chemical weathering of basalts, depending on regional climatic conditions (warm-wet vs. cold-arid), will determine the shape and position of inflection points in the seawater $\delta^7Li$ record. The possible quantification of increased rates of silicate weathering inferred from $\delta^7Li$ (mentioned below) can be utilized to determine total eruptive volume (missing + existing) and volatile emissions from the Columbia River Flood Basalts. At the same time as continental flood basalt emissions, enhanced seafloor production could also be a second possible source of $CO_2$; however, we note there is evidence that the rate of seafloor production has remained virtually invariant over the last 60 million years (Rowley, 2002; Muller et al. 2016).

The second $CO_2$ peak can possibly be caused either by the observed increase in global volcanism during the early/middle Pliocene (Kennett and Thunell, 1977; Kroenke et al., 1993), and/or by a change in silicate weathering regime. Strontium and lithium isotopes ($^{87/86}Sr$ and $\delta^7Li$) have been used as proxy for silicate weathering flux and congruency. Although the strontium isotope record exhibits a monotonous increase, lithium isotope data (Misra and Froelich, 2012) are more variable

with a transition from a period of increasing seawater $\delta^7Li$ (e.g. non-steady state weathering) to stable
seawater $\delta^7Li$ (e.g., steady state weathering) beginning at roughly 6.8 Ma (Fig. 11).

It is interesting to note that the rise in $\delta^7Li$ (Fig. 11B) from the early Miocene to the MCO is

synchronous with the rise in $pCO_2$. Before 18.5 Ma, the $pCO_2$ is relatively stable, $\delta^7Li$ is increasing,
suggesting non-steady state / incongruent nature of continental chemical weathering. From 18.6 to
16.7 Ma, the $\delta^7Li$ record decreases by ~2 ‰, consistent with decreasing weathering rates and an
associated increase in $pCO_2$. Between 16.7 and 15.9 Ma, when the eruption of the Columbia River
Flood Basalts is at a maximum, $\delta^7Li$ increases, in line with higher weathering rates that could arise
from higher atmospheric $CO_2$ and the presence of fresh basalts. The $\delta^7Li$ record then decreases again
until the end of the MCO at ~14.7 Ma, in line with a decrease in the eruption rate, sustaining high
atmospheric $CO_2$. A constant increase in $\delta^7Li$ is then observed, until the early Pliocene, where there is
evidence for a shift to a steady-state weathering regime. This increase in $\delta^7Li$ is also consistent with
the decrease in $pCO_2$ observed until the early Pliocene.

**3.9 Conclusions**

We developed a reconstruction of atmospheric $pCO_2$ based on $\delta^{11}B$ of planktic foraminifera

from ODP Sites 806 and 807 located in the Western Equatorial Pacific for the past 16 million years
and extended the record to 22 Ma by reprocessing data from Site 872 (Sosdian et al., 2018). We build
on past efforts to reconstruct atmospheric $pCO_2$ using different proxies from this region, including
from carbon isotopes in marine organic matter (Rayno et al., 1996) and alkenones (Pagani et al.,
2010), as well as foraminiferal B/Ca ratios (Tripati et al., 2009, 2011), all of which have been shown
to have a number of complexities and potential sources of systematic error (e.g., Tripati et al., 2011).
It also builds on efforts using boron isotopes in other regions using MC-ICP-MS (Seki et al., 2010;
Foster et al., 2012, 2014; Greenop et al., 2014; Martinez-Boti et al., 2015b; Stap et al., 2016; Chalk et
al., 2017; Dyez et al., 2018; de la Vega et al., 2020), and our recent work constraining fractionation
factors and measuring small samples of foraminifera (Guillermic et al., 2020).

Our study contributes a new long-term reconstruction of atmospheric $pCO_2$ for the Neogene

derived from boron isotopes from the tropical Pacific Ocean. Although the record is not continuous,
with variable resolution, it captures both long-term and short-term variability associated with several
key transitions and demonstrates the utility of examining sites in the Western Equatorial Pacific for
future higher-resolution studies. Results for Sites 806 and 807 in the Western Equatorial Pacific
reproduce the amplitude of late Pleistocene glacial-interglacial cycles in $pCO_2$. These observations are
consistent with the sites being in equilibrium with the atmosphere, although further work would be
useful to explore sources of uncertainty and differences relative to ice core $pCO_2$.

$pCO_2$ values increase from the early Miocene to the MCO with estimated MCO $pCO_2$ values

of 511 ± 201 ppm (2 SD, n=3). These elevated values are potentially linked to the eruption of the
Columbia River Flood Basalts, with values declining into the early Pliocene, including during
Pliocene glacial intensification. The changes in $pCO_2$ we observed are in line with changes in $\delta^7Li$, a
proxy of silicate weathering, and future modeling of multiple proxy records should be insightful.
Early Pliocene data for ~4.7-4.5 Ma support high $pCO_2$ of $419 \pm 119$ ppm, and elevated values during
the mid-Pliocene Warm Period of $530 \pm 110$ ppm for the time interval ~3.3-3.0 Ma. These data are
low in resolution, thereby not fully sampling orbital and millennial scale variability. The higher
resolution record for the Pliocene glacial intensification supports a reduction in $pCO_2$ during several
steps, with values at 2.7 Ma of 350 ppm, 2.6 Ma of ~280 ppm, and 2.4 Ma of ~210 ppm. We find
support for a larger reduction in glacial $pCO_2$ during the Mid-Pleistocene Transition compared to
interglacial $pCO_2$, and a minimum in $pCO_2$ during glacial MIS 30. These findings confirm a role for
$CO_2$ in the transition from a 41 kyr to a 100 kyr world.
Higher-resolution boron isotope records from the WEP would allow for further resolution of
these changes. Additional constraints on temperature, such as from clumped isotopes (Tripati et al.,
2010) in the WEP (Tripati et al., 2014), could allow for uncertainties in $pCO_2$ estimates from boron
isotopes to be reduced and for new constraints on Earth climate sensitivity. Future constraints on the
vertical structure of the tropical Pacific (Shankle et al., 2021) during these transitions may also
potentially be illuminating.

**Data availability**
All data are available in the supplemental materials. Reconstructed climate parameters and proxy data
will be archived at the *NOAA's* NCEI World Data Service for Paleoclimatology on acceptance at
https://www.ncei.noaa.gov/products/paleoclimatology.

**Author Contributions**
AT developed the project and wrote the proposals that funded the work. All authors contributed to the
experimental design. MG performed the measurements with assistance from SM. MG conducted data
analysis with input from AT. MG drafted the paper, which was edited by all authors. Interpretation
was led by MG and AT, with input from SM and RE.

**Competing interests**
The authors declare that they have no conflict of interest.

**Acknowledgments**
The authors wish to thank the Tripati Lab, including Lea Bonnin and Alexandra Villa, for assistance
with picking samples; the IODP core repository for provision of samples; Mervyn Greaves for
technical support and use of laboratory space at the University of Cambridge; Yoan Germain,
Emmanuel Ponzevera, Céline Liorzou and Oanez Lebeau for technical support and use of laboratory
space at IUEM and Ifremer (Plouzané, France). We thank Thomas Chalk, another anonymous
reviewer, and Hubertus Fischer for their helpful comments on the manuscript, and Mathis Hain for
discussion of this work.

**Financial support**


This research is supported by DOE BES grant no. DE-FG02-13ER16402 to AKT, by the International
Research Chair Program that is funded by the French government (LabexMer ANR-10-LABX-19-01)
to AKT and RAE, and IAGC student research grant 2017.

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

 **Figure captions**

**Figure 1:** Modern hydrography of sites. **A.** Map of air-sea $pCO_2$ ($\Delta pCO_2$, ppm, data from Takahashi
et al. (2014) and plotted using Ocean Data View from Schlitzer, (2016) showing the location of ODP
Sites 806 and 807 (black circles) and Site 872 (black square, Premoli et al., 1993). Depth profiles are
for preindustrial parameters, **B.** pH calculated from GLODAP database and corrected for
anthropogenic inputs, **C.** Boron isotopic composition of borate ion ($\delta^{11}B_{borate}$) with associated
propagated uncertainties.
**Figure 2:** Foraminiferal data for Miocene to recent times. **A.** Benthic foraminiferal $\delta^{18}O$ data (blue
line – stack from Lisiecki and Raymo, 2005; black line – compilation from Zachos et al., 2008). **B.**
$\delta^{11}B$ of *T. sacculifer* (blue circles) and *G. ruber* (blue triangles) at Sites 806 (light blue), 807 (dark
blue), Grey filled square represent data from Site 872 located in the WEP (Sosdian et al., 2018). Open
symbols are $\delta^{11}B$ data from published studies (Hönisch and Hemming, 2009; Seki et al., 2010; Foster
et al., 2012; Greenop et al., 2014; Martinez-Boti et al., 2015a; Chalk et al., 2017; Dyez et al., 2018;
Sosdian et al., 2018; de la Vega et al., 2020; Raitzsch et al., 2021), grey open symbols are *T.*
*sacculifer*, brown open symbols are for *G. ruber*. **C.** Mg/Ca ratios of *T. sacculifer* and *G. ruber* at
Sites 806, 807 and fourth-order polynomial regression from Sosdian et al. (2020) representing secular
variations of $Mg/Ca_{sw}$ (blue dotted line). **E.** Calculated weight per shell for *T. sacculifer* and *G. ruber*.
For Panels B-D: Circles = *T. sacculifer*, Triangles = *G. ruber*.
**Figure 3: A.** Reconstruction of surface $pCO_2$ (ppm) for the past 0.8 Myr from *T. sacculifer* at ODP
Sites 806 and 807 (blue symbols) using boron-based pH calculated from $\delta^{11}B_{seawater}$ (Greenop et al.,
2017) and alkalinity from Caves et al. (2016). Planktonic foraminiferal $\delta^{18}O$ at site 806 with isotope
stages labeled (black line – Medina-Elizalde and Lea, 2005) and benthic foraminiferal $\delta^{18}O$ stack
(grey line - Lisiecki and Raymo, 2005), benthic $\delta^{18}O$ at Site 806 (dark red line) from Lear et al. (2003,
2015). **B.** $pCO_2$ values calculated from boron isotopes (colored symbols - this study) with data from
the literature (open gray triangles – compilation B are data recalculated in Rae et al., 2021) and ice
core $pCO_2$ (black line – Petit et al., 1999, Lüthi et al., 2008, Bereiter et al., 2015). **C.** Cross plot for the
last 0.8 Myr of $pCO_{2\ \delta11B}$ from this study and $pCO_{2\ ice\ core}$ (from ice core compilation, Bereiter et al.,
2015), grey line is a simple linear regression (p = 0.25, $R^2$=0.09), blue line is a Deming regression
taking both x and y uncertainties into account (p = 0.25). Details of the regression parameters are in
Table S6. Ice core $CO_2$ error was calculated based on 2 SD of reported values, and $\pm$ 1 ky for the age
of sediment samples. Boron-based $pCO_2$ error is calculated based on error propagation described by
eq. S17. Data compiled are from: Foster et al., 2008; Hönisch and Hemming, 2009; Seki et al., 2010;
Foster et al., 2012; Badger et al., 2013; Greenop et al., 2014; Martinez-Boti et al., 2015a; Chalk et al.,
2017; Dyez et al., 2018; Sosdian et al., 2018; Greenop et al., 2019; de la Vega et al., 2020.
**Figure 4:** Compilation of temperatures from Site 806 in the WEP. Mg/Ca based temperatures were
derived using the same framework (see supplemental information). Blue filled symbols are from Sites
806 and 807 with blue circles for *T. sacculifer* and triangles for *G. ruber*; filled gray squares are data
from Site 872 (Sosdian et al., 2018). Open symbols are SST derived from Mg/Ca at Site 806 (Wara et
al., 2005; Tripati et al., 2009; Nathan and Leckie, 2009). $Tex_{86}$ and $U^{K'}_{37}$ are also plotted for
comparison (Zhang et al., 2014). Orange open circles are SST data calculated with our framework
from the species *D. altispera* at ODP Site 806 (Sosdian et al., 2020) with an offset of +8°C. Blue line
is a smooth line (Lowess) going through the data.
**Figure 5:** Different models for the evolution of the boron geochemistry explored as part of this work.
Due to the 1 ‰ uncertainty propagated for $\delta^{11}B_{seawater}$, all scenarios yield reconstructed seawater pH
values that are within error of each other. Propagated uncertainties were calculated using eq. S14 (see
Supplement). **A.** Different models for $\delta^{11}B_{seawater}$ used for the reconstruction of $pCO_2$ in this study
(blue – Lemarchand et al., 2000; green – Greenop et al., 2017; red – Raitzsch and Hönisch, 2013). **B.**
Reconstructed pH based on our measured $\delta^{11}B_{carbonate}$ values using different models for $\delta^{11}B_{seawater}$
(blue – Lemarchand et al., 2000; green – Greenop et al., 2017; red – Raitzsch and Hönisch, 2013),

compilations of pH from Sosdian et al. (2018) (compilation A - open squares) and Rae et al. (2021) (compilation B - open triangles) are also shown for comparison. Data for compilation A are from: Hönisch and Hemming, 2009; Seki et al., 2010; Foster et al., 2012; Badger et al., 2013; Greenop et al., 2014; Martinez-Boti et al., 2015a; Chalk et al., 2017; Sosdian et al., 2018. Data for compilation B are from: Foster et al., 2008; Hönisch and Hemming, 2009; Seki et al., 2010; Foster et al., 2012; Badger et al., 2013; Greenop et al., 2014; Martinez-Boti et al., 2015a; Chalk et al., 2017; Dyez et al., 2018; Sosdian et al., 2018; Greenop et al., 2019; de la Vega et al., 2020.

**Figure 6:** Different models for the evolution of a second carbonate (e.g. alkalinity) system parameter explored as part of this work. The propagated uncertainties were calculated using eq. S16 (see Supplement). **A.** Different models for alkalinity used for the reconstruction of $pCO_2$ in this study (brown – constant alkalinity of 2330 μmol/kg, blue - Ridgwell and Zeebe, 2005; green - Tyrell and Zeebe, 2004; violet - Caves et al., 2016. Colored symbols are reconstructed $pCO_2$ based on our measured $\delta^{11}B_{carbonate}$ values , alkalinity scenario and $\delta^{11}B_{seawater}$ from Greenop et al., 2017; open squares (compilation A) are $pCO_2$ compilation from Sosdian et al. (2018), open triangles (compilation B) are from the compilation by Rae et al. (2021), black symbols are from site 872. **B.** Reconstructed $pCO_2$ using constant alkalinity of 2330 μmol/kg and $\delta^{11}B_{seawater}$ from Greenop et al. (2017). **C.** Reconstructed $pCO_2$ using the constant alkalinity scenario from Ridgwell and Zeebe, (2005) and $\delta^{11}B_{seawater}$ from Greenop et al. (2017). **D.** Reconstructed $pCO_2$ using constant alkalinity scenario from Tyrell and Zeebe, (2004) and $\delta^{11}B_{seawater}$ from Greenop et al. (2017). **E.** Reconstructed $pCO_2$ using constant alkalinity scenario from Caves et al., (2016) and $\delta^{11}B_{seawater}$ from Greenop et al. (2017). In black are published estimates from ice core data (circles - Yan et al., 2019). Compilations of $pCO_2$ from Sosdian et al. (2018) (compilation A - open squares) and Rae et al. (2021) (compilation B - open triangles) are also shown for comparison. Data for compilation A are from: Hönisch and Hemming, 2009; Seki et al., 2010; Foster et al., 2012; Badger et al., 2013; Greenop et al., 2014; Martinez-Boti et al., 2015a; Chalk et al., 2017; Sosdian et al., 2018. Data for compilation B are from: Foster et al., 2008; Hönisch and Hemming, 2009; Seki et al., 2010; Foster et al., 2012; Badger et al., 2013; Greenop et al., 2014; Martinez-Boti et al., 2015a; Chalk et al., 2017; Dyez et al., 2018; Sosdian et al., 2018; Greenop et al., 2019; de la Vega et al., 2020. Stars indicate $pCO_2$ values reconstructed from alkenones by Tanner et al. (2020) (simulation 6) at Site 1088 in the Southern Ocean.

**Figure 7:** Proxy data for the past 22 million years in the Western Equatorial Pacific compared to benthic oxygen isotope data. **A.** Benthic $\delta^{18}O$ (blue line – stack from Lisiecki and Raymo, 2005; black line – compilation from Zachos et al., 2008). **B.** Benthic $\delta^{13}C$ (black line – compilation from Zachos et al., 2008). **C to E**, color indicates the site (filled light blue=806, filled dark blue=807), symbols represent the species (circle=*T. sacculifer* and triangle=*G. ruber*), filled grey squares are recalculated data based on Sosdian et al. (2018) at site 872. **C.** SST reconstructed at ODP Sites 806 and 807 using Mg/Ca ratios (see supplemental information for reconstruction details), open symbols are reconstructed temperatures based on literature Mg/Ca at site 806 (see text or Fig. 4). **D.** Seawater pH reconstructed from $\delta^{11}B$ of *T. sacculifer* and *G. ruber* using $\delta^{11}B_{seawater}$ from Greenop et al. (2017) (refer to text and supplement for calculations, this study), open squares (compilation A) represent data from the $CO_2$ compilation of Sosdian et al. (2018) and open triangles (compilation B) are compilation data from Rae et al. (2021). **E.** Reconstructed $pCO_2$ (ppm) using boron-based pH and alkalinity from Caves et al. (2016), data presented are from this study. Propagated uncertainties are given by eq. S17 for the dark blue envelope, while the light blue envelope are the uncertainties calculated based on eq. S16 (taking into account uncertainty in $\delta^{11}B_{seawater}$). Crosses are original $pCO_2$ values calculated in Sosdian et al. (2018) at Site 872; asterisks are recalculated $pCO_2$ values at Site 872 by Rae et al. (2021).

**Figure 8:** Proxy data from 22 to 6 million years, including the Middle Miocene Climate Transition (MMCT) and Miocene Climate Optimum (MCO), in the Western Equatorial Pacific compared to benthic oxygen isotope data. **A.** Benthic $\delta^{18}O$ (black line – compilation from Zachos et al., 2008). **B.** Benthic $\delta^{13}C$ (black line – compilation from Zachos et al., 2008). **C and D**, color indicates the site

(filled light blue=806, filled dark blue=807), symbols represent the species (circle=*T. sacculifer* and triangle=*G. ruber*), filled grey squares are recalculated data based on Sosdian et al. (2018) at site 872. **C**. SST reconstructed at ODP Sites 806 and 807 using Mg/Ca ratios (see supplemental informations for reconstruction details), open symbols are reconstructed temperatures based on literature Mg/Ca at site 806 (see text or Fig. 4). **D**. Reconstructed $pCO_2$ (ppm) from this study (blue symbols) using boron-based pH and alkalinity from Caves et al. (2016). Propagated uncertainties are given by eq. S17 for the dark blue envelope, while the light blue envelope reflects the uncertainties calculated based on eq. S16 (taking into account uncertainty on $\delta^{11}B_{seawater}$). Orange datapoints and envelope are calculated $pCO_2$ values and associated uncertainty from our study using our framework and a constant alkalinity scenario. Open squares (compilation A) are compilation data from Sosdian et al. (2018), open triangles are data from Raitzsch et al. (2021) at Site 1092. Crosses are original $pCO_2$ calculated in Sosdian et al. (2018) at Site 872; asterisks are recalculated $pCO_2$ at Site 872 by Rae et al. (2021); dark red triangles are from Site 1092 (Raitzsch et al., 2021). Data for compilation A are from: Hönisch and Hemming, 2009; Seki et al., 2010; Foster et al., 2012; Badger et al., 2013; Greenop et al., 2014; Martinez-Boti et al., 2015a; Chalk et al., 2017; Sosdian et al., 2018. Data for compilation B are from: Foster et al., 2008; Hönisch and Hemming, 2009; Seki et al., 2010; Foster et al., 2012; Badger et al., 2013; Greenop et al., 2014; Martinez-Boti et al., 2015a; Chalk et al., 2017; Dyez et al., 2018; Sosdian et al., 2018; Greenop et al., 2019; de la Vega et al., 2020.

**Figure 9:** Proxy data from 7 to 1 million years, including the Warm Pliocene Transition (WPT), in the Western Equatorial Pacific compared to benthic oxygen isotope data. **A.** Benthic $\delta^{18}O$ (black line – compilation from Zachos et al., 2008). **B.** Benthic $\delta^{13}C$ (black line – compilation from Zachos et al., 2008). **C and D**, color indicates the site (filled light blue=806, filled dark blue=807), symbols represent the species (circle=*T. sacculifer* and triangle=*G. ruber*), filled grey squares are recalculated data based on Sosdian et al. (2018) at ODP Site 872. **C.** SST reconstructed at ODP Sites 806 and 807 using Mg/Ca ratios (see supplemental informations for reconstruction details), open symbols are reconstructed temperatures based on litearature Mg/Ca at site 806 (see text or Fig. 4). **D.** Reconstructed $pCO_2$ (ppm) from this study (blue symbols) using boron-based pH and alkalinity from Caves et al. (2016). Propagated uncertainties are given by eq. S17 for the dark blue envelope, while the light blue envelope reflects the uncertainties calculated based on eq. S16 (taking into account uncertainty on $\delta^{11}B_{seawater}$). Open squares (compilation A) are $pCO_2$ compilation from Sosdian et al. (2018), open triangles (compilation B) are from the compilation by Rae et al. (2021). Data for compilation A are from: Hönisch and Hemming, 2009; Seki et al., 2010; Foster et al., 2012; Badger et al., 2013; Greenop et al., 2014; Martinez-Boti et al., 2015a; Chalk et al., 2017; Sosdian et al., 2018. Data for compilation B are from: Foster et al., 2008; Hönisch and Hemming, 2009; Seki et al., 2010; Foster et al., 2012; Badger et al., 2013; Greenop et al., 2014; Martinez-Boti et al., 2015a; Chalk et al., 2017; Dyez et al., 2018; Sosdian et al., 2018; Greenop et al., 2019; de la Vega et al., 2020. In black are published estimates from ice core data (circles - Yan et al., 2019).

**Figure 10:** Proxy data from 1.5 to 0.5 million years, including the Middle Pleistocene Transition (MPT), in the Western Equatorial Pacific compared to benthic oxygen isotope data. **A.** Benthic $\delta^{18}O$ (blue line – stack from Lisiecki and Raymo, 2005). **B.** Benthic $\delta^{13}C$ (black line – compilation from Zachos et al., 2008). **C and D** color indicates the site (filled light blue=806, filled dark blue=807), symbols represent the species (circle=*T. sacculifer* and triangle=*G. ruber*), filled grey squares (compilation A) are recalculated data based on Sosdian et al. (2018) at site 872. **C.** SST reconstructed at ODP Sites 806 and 807 using Mg/Ca ratios (see supplemental informations for reconstruction details), open symbols are reconstructed temperatures based on litearature Mg/Ca at site 806 (see text or Fig. 4). **D.** Reconstructed $pCO_2$ (ppm) from this study (blue symbols) using boron-based pH and alkalinity from Caves et al. (2016). Propagated uncertainties are given by eq. S17. In black are published estimates from ice core data (line – Bereiter et al., 2015; black circles - Yan et al., 2019). Open triangles (compilation B) are from the compilation by Rae et al. (2021). Data for compilation B are from: Foster et al., 2008; Hönisch and Hemming, 2009; Seki et al., 2010; Foster et al., 2012; Badger et al., 2013; Greenop et al., 2014; Martinez-Boti et al., 2015a; Chalk et al., 2017; Dyez et al., 2018; Sosdian et al., 2018; Greenop et al., 2019; de la Vega et al., 2020.

**Figure 11:** Proxy data from 1.5 to 0.5 million years, including the Middle Pleistocene Transition
(MPT), in the Western Equatorial Pacific compared to benthic oxygen isotope composites. **A.** Benthic
$\delta^{18}O$ (blue line – compilation from Lisiecki and Raymo, 2005, black line – compilation from Zachos
et al. 2008). **B.** Records from Lithium isotopes ($\delta^7Li$, orange, Misra and Froelich, 2012) and
Strontium isotopes ($^{87/86}Sr$, grey, Hodell et al., 1991, Farrel et al., 1995, Martin et al., 1999, Martin et
al., 2004), both proxies for silicate weathering. Orange arrows represent the different weathering
regimes as indicated by the $\delta^7Li$, black crosses indicates when changes in weathering regime occur. **C.**
Reconstructed $pCO_2$ (ppm) using boron-based pH and alkalinity from Caves et al. (2016), color
indicates the site (filled light blue=806, filled dark blue=807), symbols represent the species (circle=*T.*
*sacculifer* and triangle=*G. ruber*), filled grey squares (compilation A) are recalculated data based on
Sosdian et al. (2018) at site 872. Data for compilation A are from: Hönisch and Hemming, 2009; Seki
et al., 2010; Foster et al., 2012; Badger et al., 2013; Greenop et al., 2014; Martinez-Boti et al., 2015a;
Chalk et al., 2017; Sosdian et al., 2018. Propagated uncertainties are given by eq. S17 for the dark
blue envelope, while the light blue envelope are the uncertainties calculated based on eq. S16 (taking
into account uncertainty on $\delta^{11}B_{seawater}$). Also shown is the timing of major events. The rose band and
dark rose band indicate the eruption of the Columbia River flood basalts (Hooper et al., 2002) and
time of maximum eruption (Kasbohm and Schoene, 2018), respectively.

**Table 1:** Core information.

| Cruise | Leg | Hole | N (°) | E (°) | Depth (m) |
|---|---|---|---|---|---|
| ODP | 130 | 807 | 3.61 | 156.62 | 2804 |
| ODP | 130 | 806 | 0.32 | 159.37 | 2520 |

**Table 2:** Comparison of reconstructed $pCO_2$ values for key intervals in the last 16 Myr.

**Mid-Pleistocene transition (1.2-0.8 Ma)**

| MIS (G) | $pCO_2$ (ppm) | Reference | MIS (IG) | $pCO_2$ (ppm) | Reference | $pCO_2$ amplitude IG-G (ppm) |
|---|---|---|---|---|---|---|
| 20 | 179 | This study | 21 | 254 | This study | 75 |
| 22 | 187 | This study | 23 | 230 | This study | 43 |
| 24 | nd | | 25 | 298 | This study | nd |
| 26 | nd | | 27 | nd | | nd |
| 28 | 174 | This study | 29 | nd | | nd |
| 30 | 170 | This study | 31 | 295 | Hönisch et al., 2009 (N-TIMS) | 125 |
| 32 | 218 | Chalk et al., 2017 | 33 | 323 | Chalk et al., 2017 | 105 |
| 34 | 197 | Chalk et al., 2017 | 35 | 315 | Chalk et al., 2017 | 118 |
| 36 | 189 | Chalk et al., 2017 | 37 | 295 | This study, Chalk et al., 2017 | 106 |
| | | | 39 | 306 | This study | nd |

**Middle Pliocene Warm Period (3.29-2.97 Ma)**

| $pCO_2$ (ppm) | Reference |
|---|---|
| 530 ± 110 | This study (2 SD, n=4) |
| 320 ± 130 | Martinez-Boti et al., 2015b (2 SD, n=8) |
| 360 ± 85 | de la Vega et al., 2020 (2 SD, n=59) |

**Early Pliocene Warm Period (4.7-4.5 Ma)**

| $pCO_2$ (ppm) | Reference |
|---|---|
| 419 ± 119 | This study (2 SD, n=3) |

**Miocene Climate Optimum (17-14 Ma)**

| $pCO_2$ (ppm) | Reference |
|---|---|
| 511 ± 201 | This study (2 SD, n=3) |
| 350-400 | Foster et al., 2012 |
| 300-500 | Greenop et al., 2014 |
| 470-630 | Sosdian et al., 2018 |
| 687 ± 421 | Rae et al., 2021 (2 SD, n=58) |

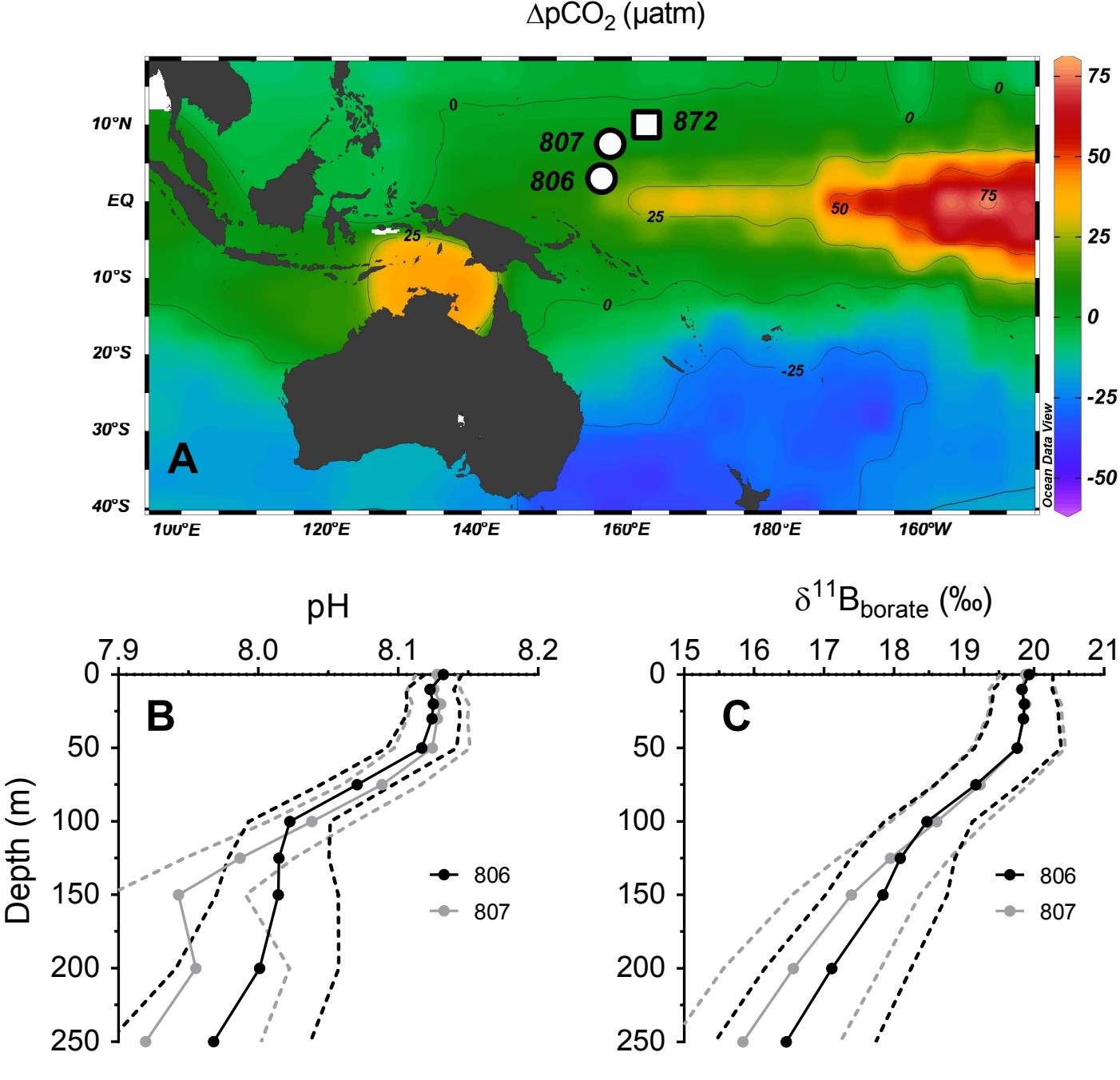

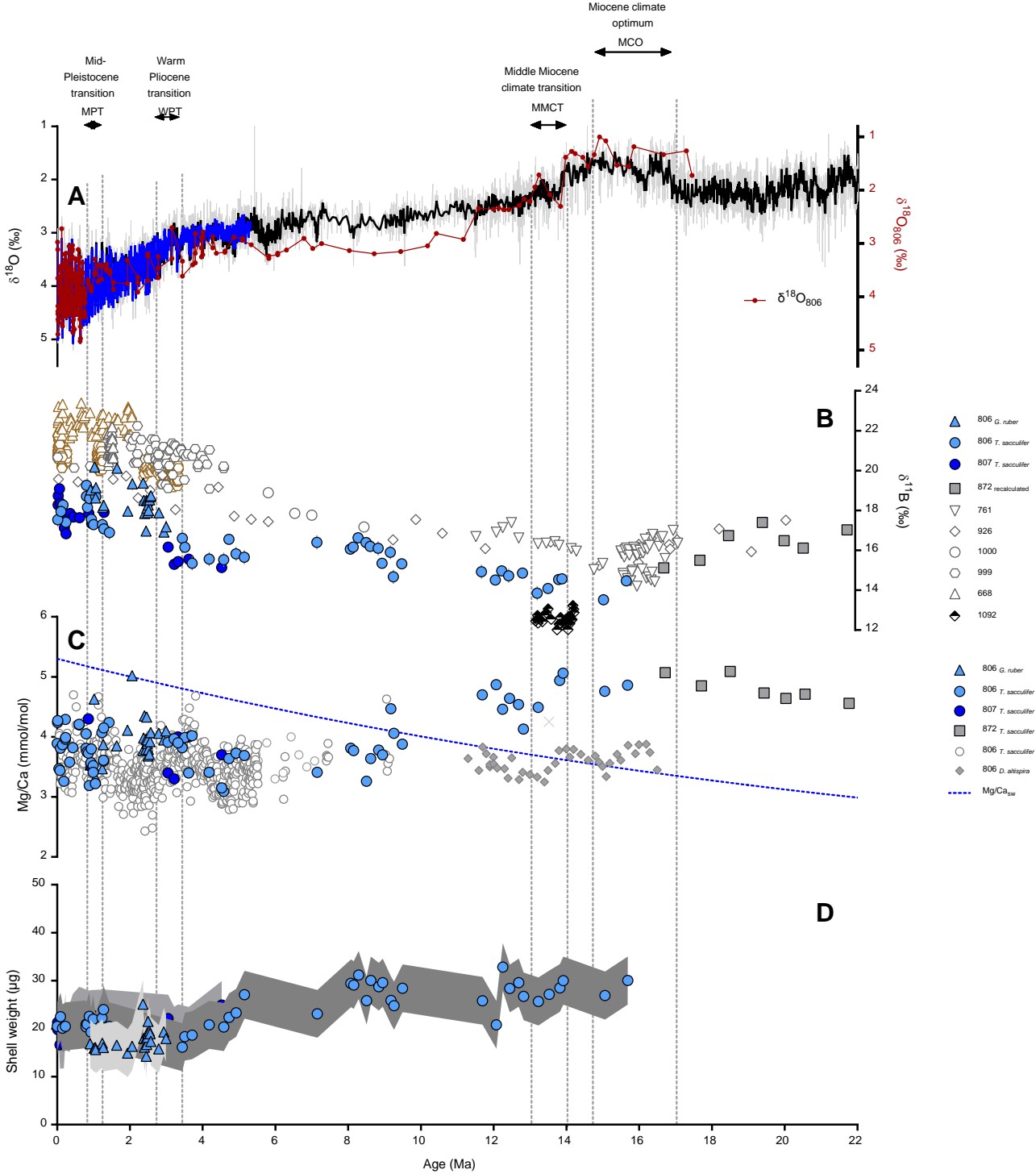

Figure 2

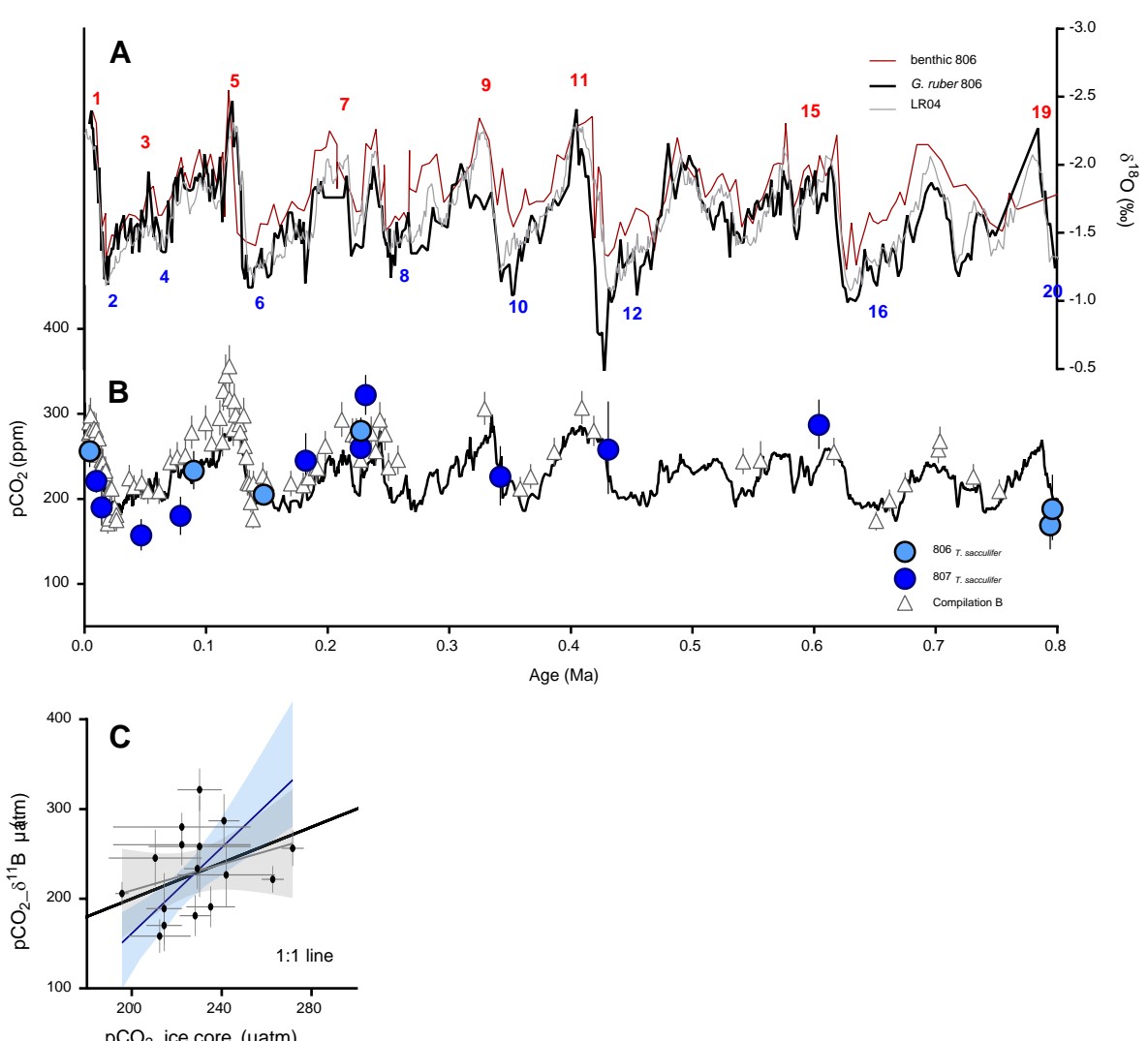

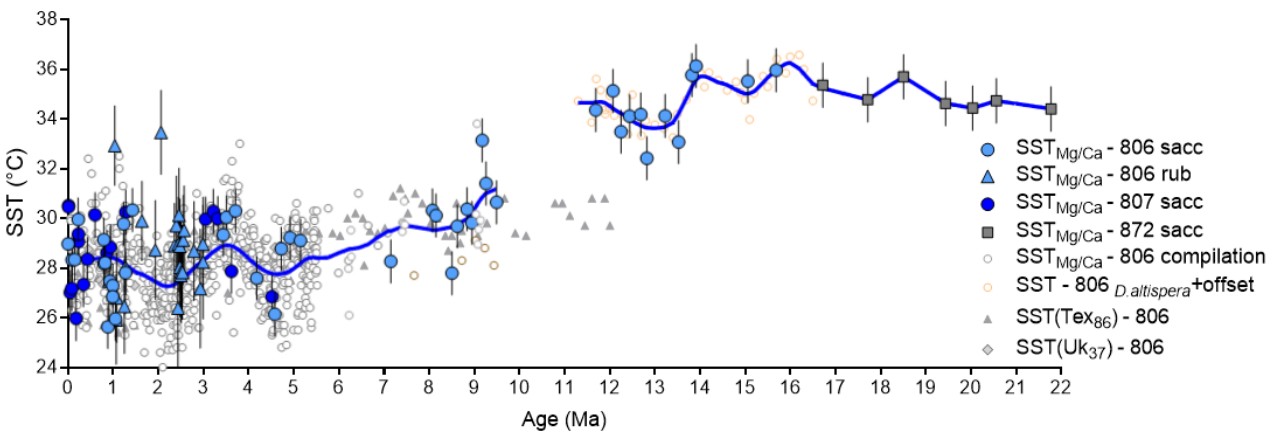

Figure 4

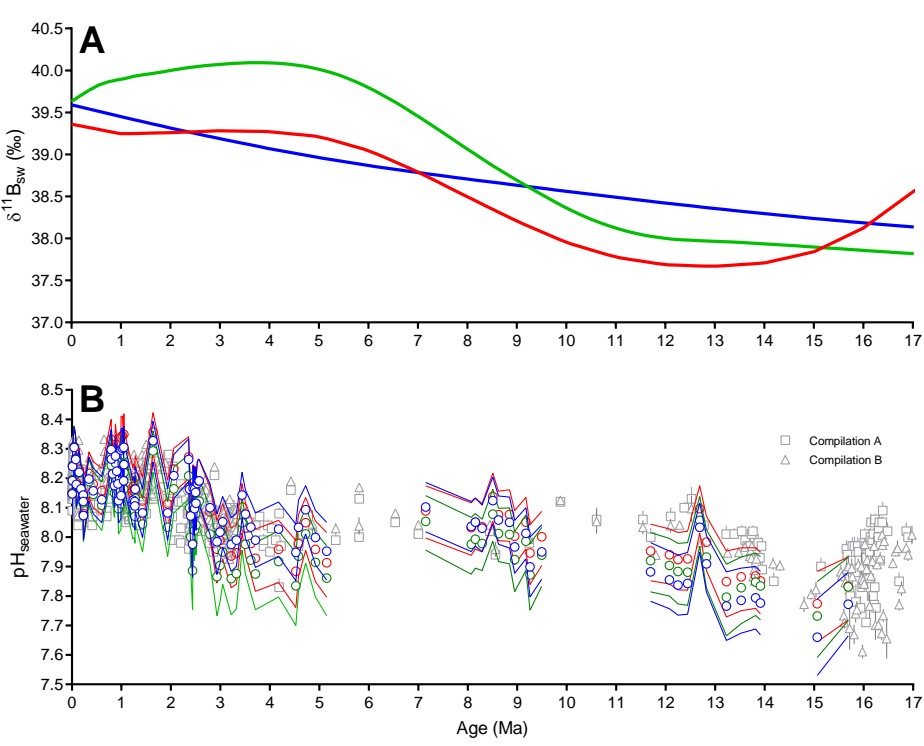

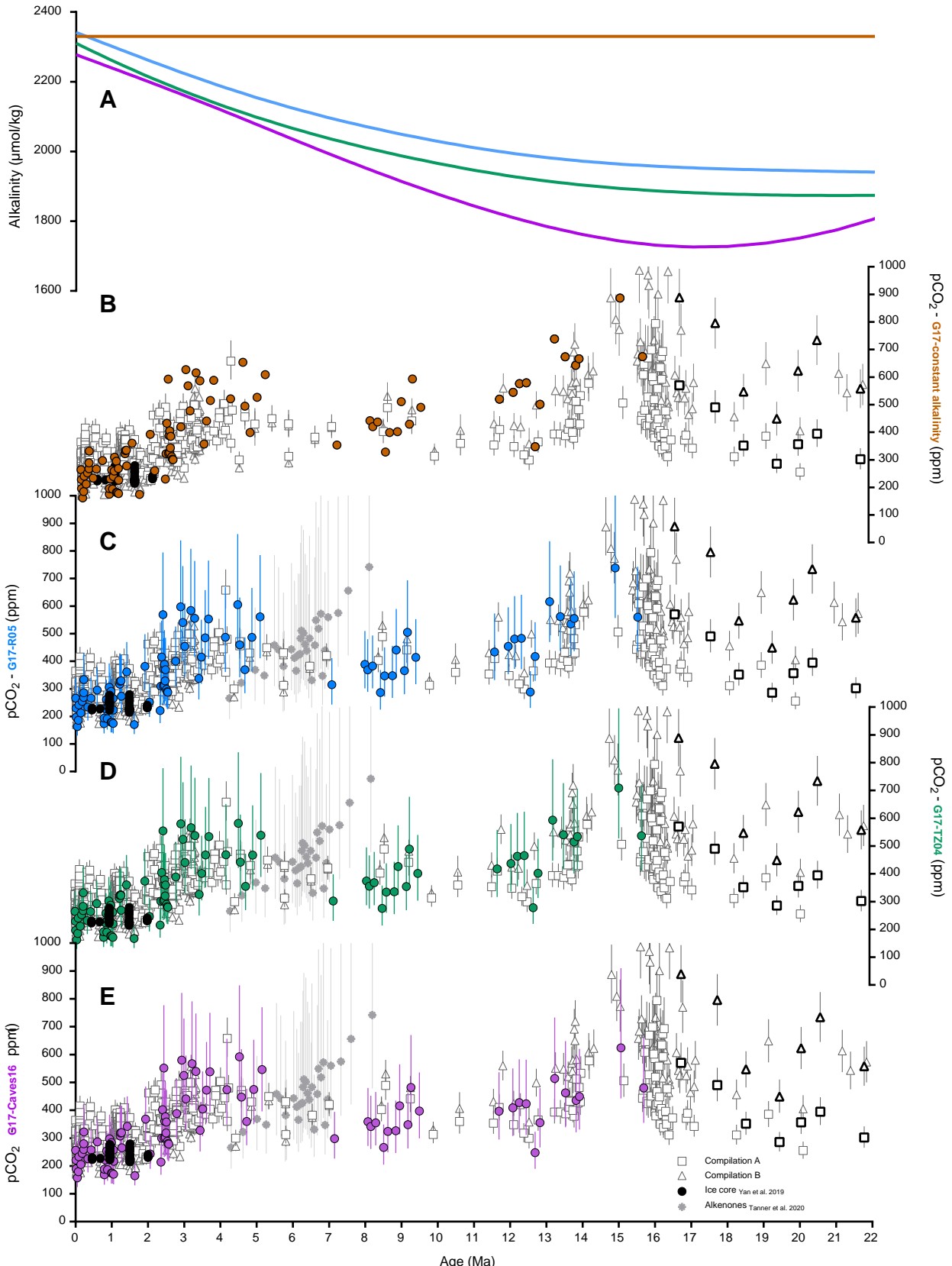

Figure 6

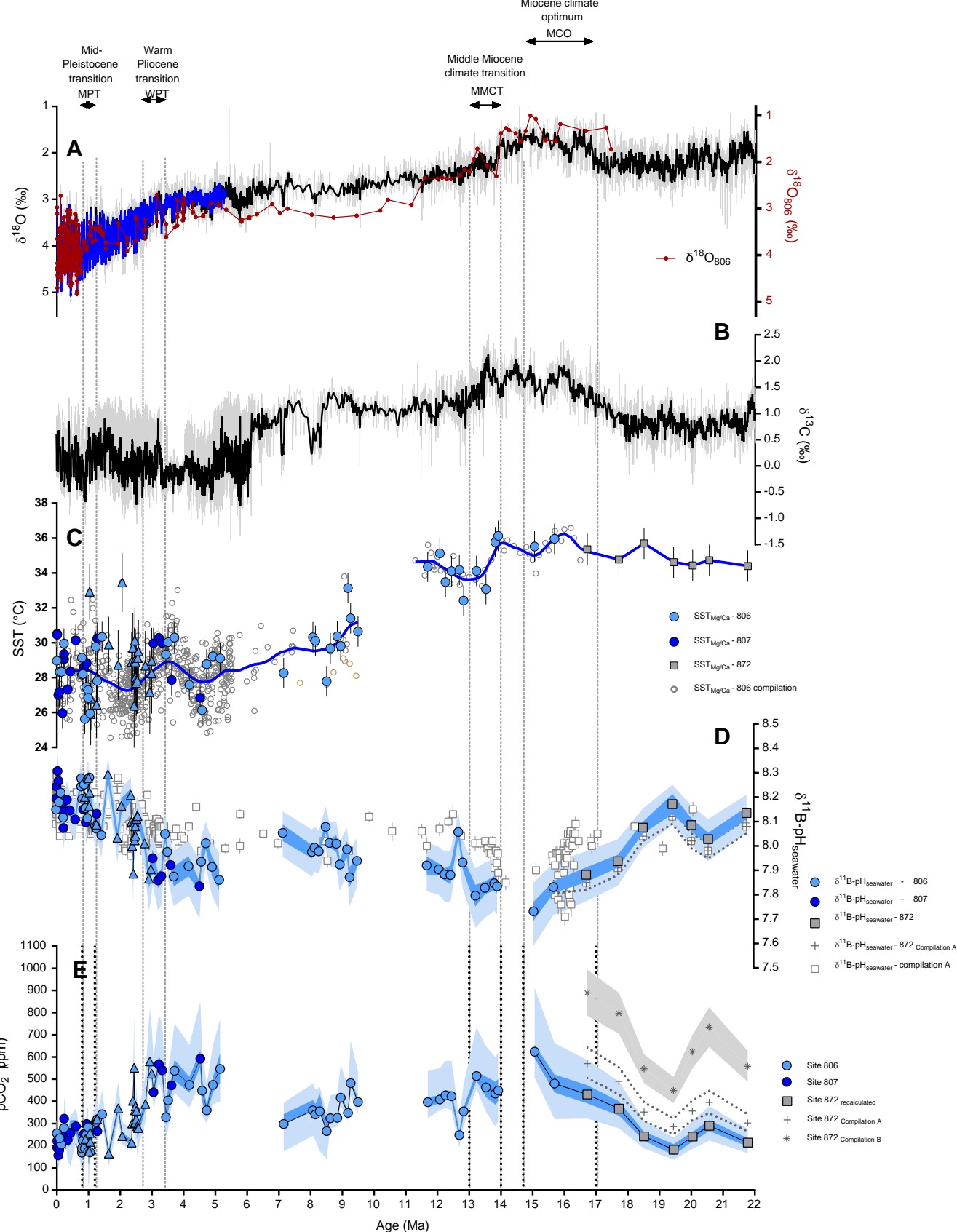

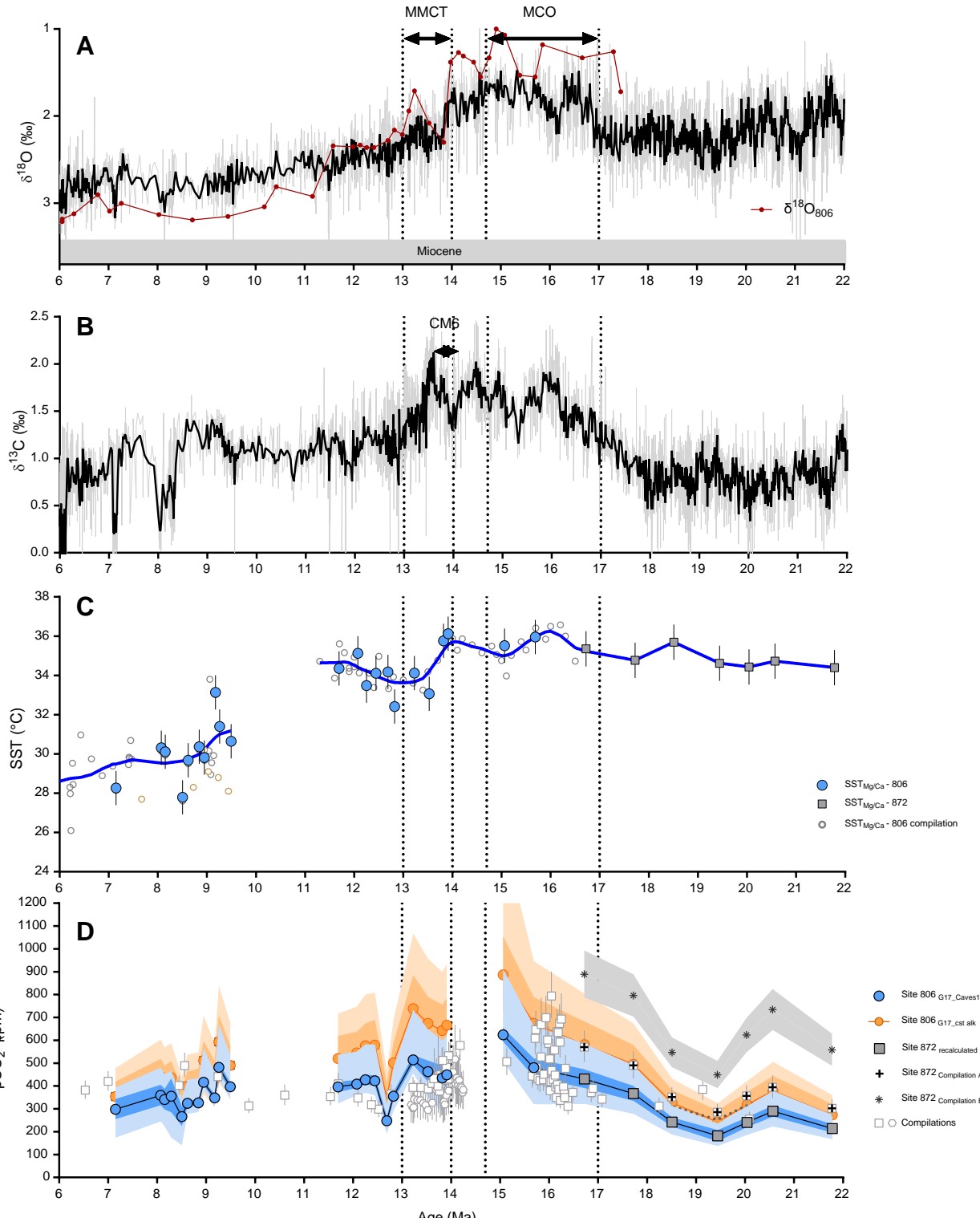

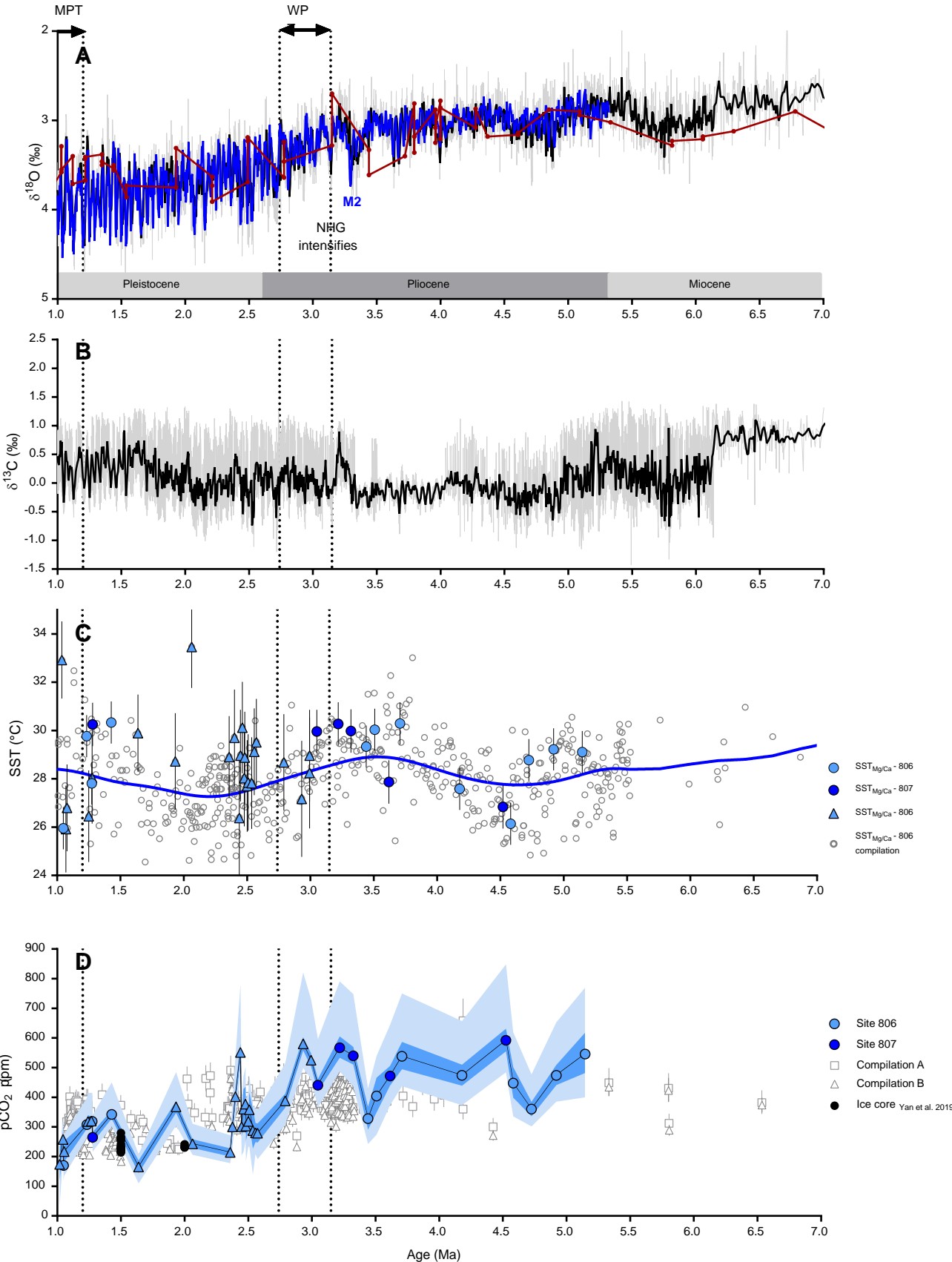

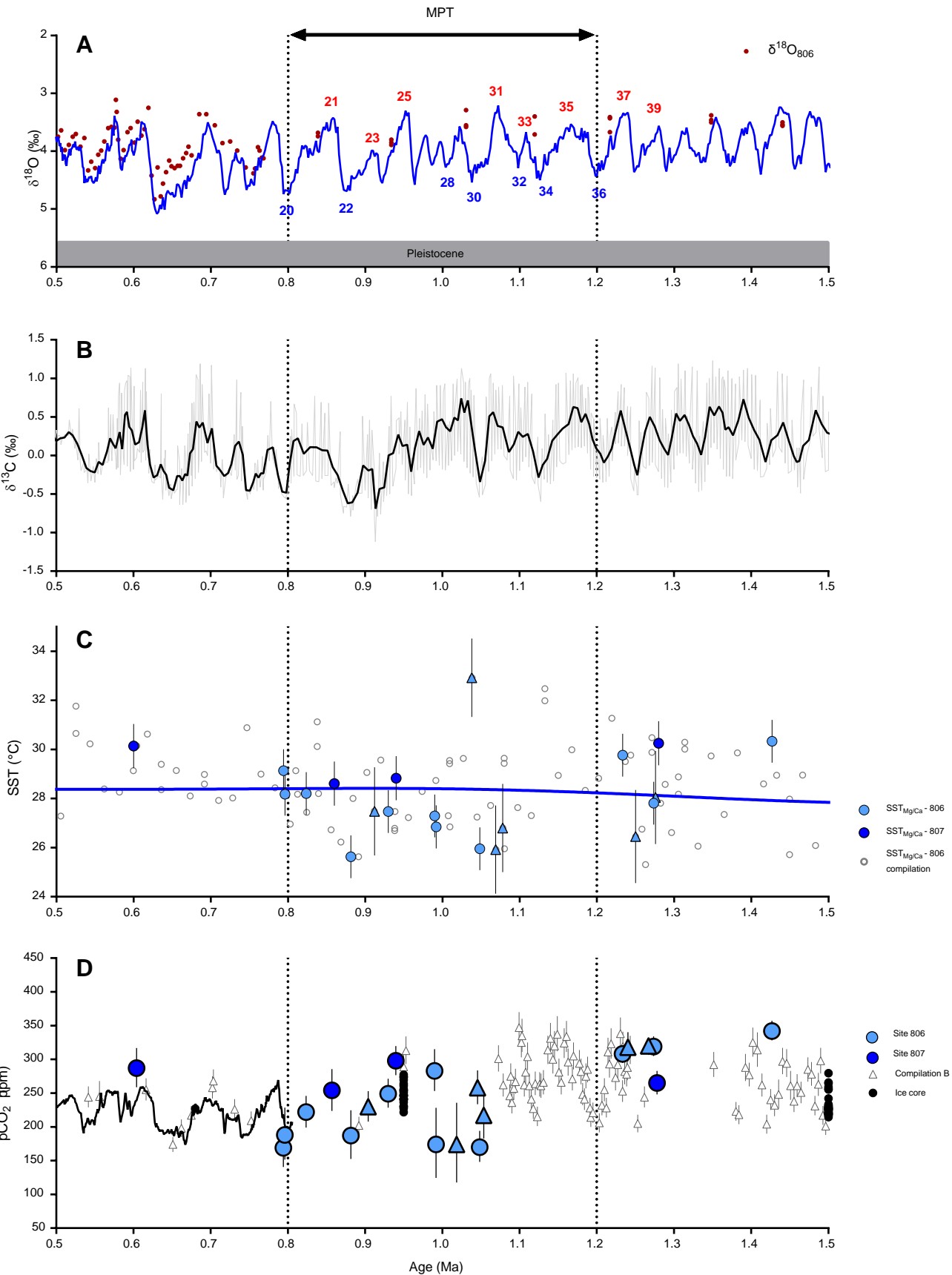

Figure 10

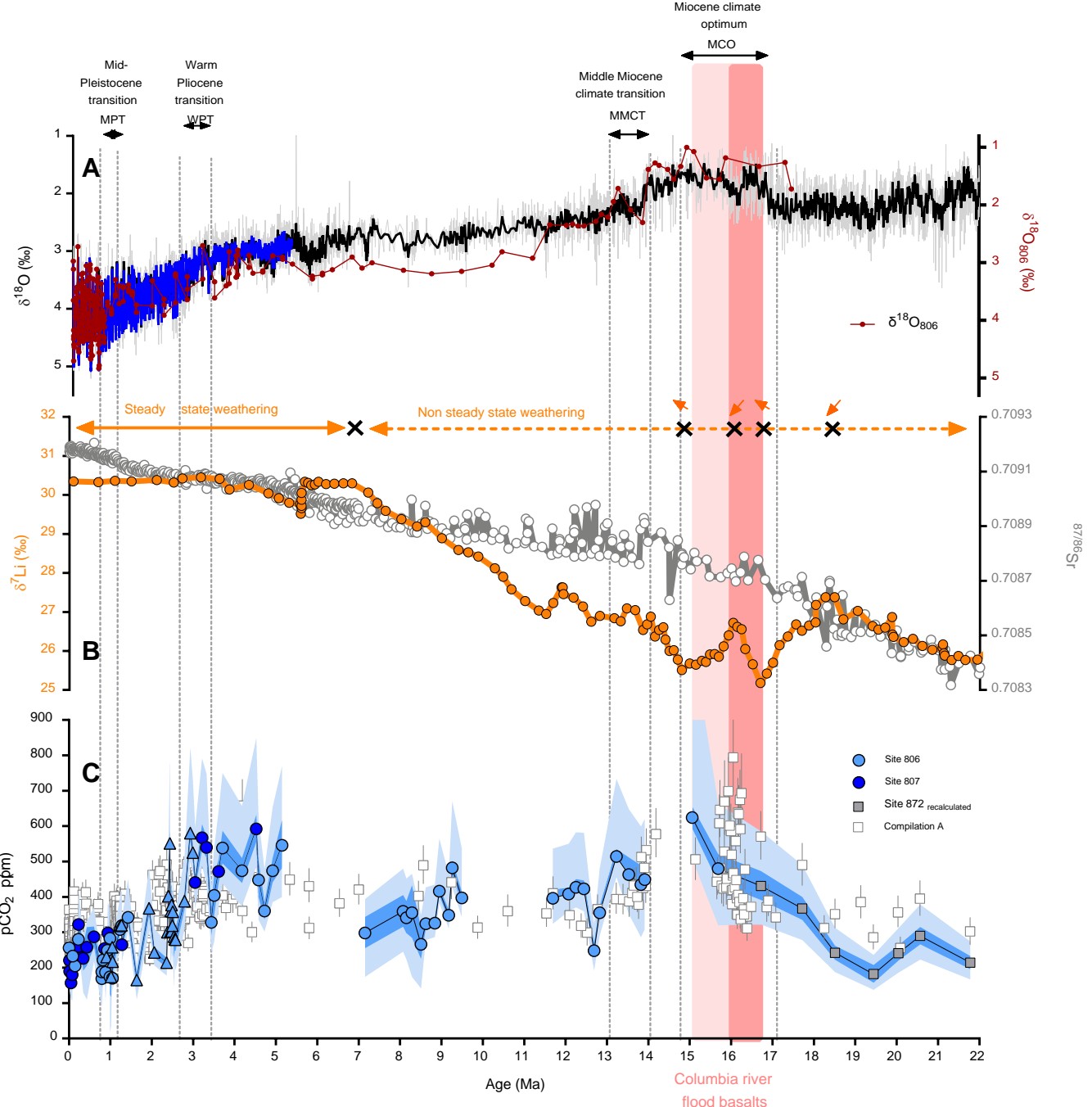