# Peer review of "Atmospheric CO2 estimates for the Miocene to Pleistocene based on foraminiferal $\delta^{11}\text{B}$ at Ocean"

_Climate of the Past, 2020_

## Author Response (AR1)

**UNIVERSITY OF CALIFORNIA, LOS ANGELES**

**UCLA**

BERKELEY • DAVIS • IRVINE • LOS ANGELES • RIVERSIDE • SAN DIEGO • SAN FRANCISCO

SANTA BARBARA • SANTA CRUZ

[Figure]

DEPARTMENT OF EARTH AND SPACE SCIENCES
3806 GEOLOGY BUILDING
BOX 951567
LOS ANGELES, CALIFORNIA 90095-1567
TEL: (310) 825-3880
FAX: (310) 825-2779

Dear Dr. Fischer,

We want to thank you for your handling of our manuscript, and thank the reviewers for their comments. We have made the requested revisions and include a detailed response below. We include the responses for both reviewers in this document. Thank you for consideration of our manuscript.

Sincerely,

Maxence Guillermic and Aradhna Tripati on behalf of the co-authors

Response reviewer 1 starts page 2

Response reviewer 2 starts page 13

**Reviewer 1**

Guillermic et al. present boron isotope data from the Western Equatorial Pacific (WEP) at a variety of resolutions across the Neogene. Whilst new boron isotope data from this region is welcome, their results (and interpretations) are not consistent with major findings from many other studies, a point which is not expressed well in the submitted manuscript.

**Response 1:** We have added comparisons to additional datasets, including Sosdian et al., 2018 and Rae et al., 2021, as seen in Figure 6, which explores the use of different models for the second carbonate system parameter in seawater. This comparison shows that results are broadly consistent between these studies. In addition, we do a direct comparison with data for Site 872 as calculated in both studies, and then recalculated here, in Figure 7, and see that both pH and $pCO_2$ are in strong agreement.

I find that the paper needs some significant reworking, as the message is not clear, it is a big topic to cover and many of the appropriate references are in, but not necessarily called out at the appropriate place.

My biggest concern is with the interpretation of the record. In many places there are large offsets between the WEP data and other published data. I do not suggest that this invalidates the data themselves, but I think a more in depth explanation is often required. The reason given is commonly air-sea offsets, but this cannot be relied upon for >150ppm changes in the mid Pliocene when all the sites are supposedly 'in equilibrium'. Many of the sites used in the previous compilation of Sosdian et al 2018 are close by (particularly ODP 872, which agrees in absolute terms with ODP 871 in the Indian Ocean).

The comparison to ODP site 999 for the Plio-Pleistocene needs some more careful thought, as I do not follow the discussion on changes to air-sea equilibrium there.

**Response 2:** We agree that the variability in the $pCO_2$ mostly comes from the assumptions made for the reconstructions in each of the studies. Therefore, we now explore systematically the impact of choice of carbonate system parameter, and methods used for calculations, for our data and those of Sosdian et al., 2018 and Rae et al., 2021, in Figures 6 and 7, and other studies. Figure 6 explores the use of different models for the second carbonate system parameter in seawater. This comparison shows that results are broadly consistent.

We do a direct comparison with data for western equatorial Pacific ODP Site 872 as calculated in both studies, and then recalculated here, in Figure 7, and see that $\delta^{11}B$, pH, and $pCO_2$ are in strong agreement, that strengthens our confidence in the use of these sites.

We agree with the comment that ODP Site 806 should be more influenced by the constriction of the Indonesian through-flow but our reconstruction is consistent with Site 761 for the Miocene which suggests that the constriction does not support main changes in air-sea equilibrium. We have added this to the text in line 492: "The similarity between our reconstructed values and those published for ODP Site 871 in the Indian Ocean (Sosdian et al., 2018) suggests that changes in Indonesian through-flow do not induce substantial changes in air-sea exchange in the WEP.".

Diagenesis has the potential to affect these records and this is not discussed at all, nor is the reader pointed to the supplement or **data comparisons to compare the raw data**, calculations, and calibrations used to gather the final data. Given the nature of this paper as a low resolution Neogene time-series, I think further validation and comparison are required.

**Response 3:** To further validate and compare, we now show the raw data in Figures 2, and point to this in the section 3.1, line 283 "Geochemical results".

We have now added text on the potential influence of diagenesis to a section in the methods.

"2.2 Preservation

Microfossils in sediments at these sites, as with any sedimentary sequences, have the potential to be influenced by diagenesis. Despite evidence of authigenic carbonate formation, recent modeling work concluded the influence of dissolution and reprecipitation at Sites 806 and 807 was relatively minor (Mitnik et al., 2018). Prior work has also found minimal impacts on the B/Ca ratio of Pliocene foraminifera from Site 806 (White and Ravelo, 2020), and on the Mg/Ca ratio of Miocene *D. altispera* shells at Site 806 (Sosdian et al., 2020). The weight/shell ratio is commonly used to monitor dissolution, and the only published record at Site 806 for the Pliocene does not show a trend consistent with dissolution of *T. sacculifer* (Wara et al., 2005). We do note that while the "coccolith size-free dissolution" index reported in Si and Rosenthal (2019) indicates higher dissolution rates in the Miocene, their records were thought to be biased from changes in foraminifera assemblages as discussed in White and Ravelo (2020).

To further assess the potential impact of dissolution in our geochemical data, the weight/shell ratio was examined in our samples. The weight/shell index used to monitor dissolution does not present any trend within the interval studied consistent with dissolution. Absolute weights/shell are increasing in the Miocene, which is not consistent with dissolution influencing the record (Fig. 2E). Additionally, we note that reconstructed pH and $pCO_2$ values also exhibit reasonable correspondence with the Vostok ice core data. Downcore $\delta^{11}B$ values from Sites 806 and 807 are similar, despite evidence for higher authigenic carbonate at Site 807 relative to Site 806 (Mitnik et al., 2018). Further, the consistency of our boron isotope and Mg/Ca results with at the two sites with each other, and to the published data from Site 872 (Sosdian et al., 2018), each with different sedimentation rates, are consistent with diagenesis not being a primary driver of the record. Comparison of raw data, and derived parameters, is shown in Figs. 2 and 7."

Please note for the below:

MB15, S18, C17, D18, H09, B11, G14, DlV20 refer to the references MartinezBoti et al 2015, Sosdian et al 2018, Chalk et al 2017, Dyez et al. 2018, Hoenisch et al 2009, Bartoli et al. 2011, Greenop et al., 2014 and De la Vega et al 2020 respectively.

Some additional references I have called out to:

Sub‐Permil Interlaboratory Consistency for Solution‐Based Boron Isotope Analyses on Marine Carbonates - Gutjahr - 2021 - Geostandards and Geoanalytical Research - Wiley Online Library

Robust Constraints on Past CO2 Climate Forcing From the Boron Isotope Proxy - Hain - 2018 - Paleoceanography and Paleoclimatology - Wiley Online Library

Decreasing Atmospheric CO2 During the Late Miocene Cooling - Tanner - 2020 - Paleoceanography and Paleoclimatology - Wiley Online Library

Upregulation of phytoplankton carbon concentrating mechanisms during low CO2 glacial periods and implications for the phytoplankton pCO2 proxy - ScienceDirect

North Atlantic temperature and pCO2 coupling in the early-middle Miocene | Geology | GeoScienceWorld

Miocene Evolution of North Atlantic Sea Surface Temperature - Super - 2020 - Paleoceanography and Paleoclimatology - Wiley Online Library

A diatom record of CO2 decline since the late Miocene - ScienceDirect

**Response 4:** We have added these references to the text except Meija et al. (2017) - a reconstruction from the EEP.

Specific points:

Line 26: What is the evidence that these sites are in equilibrium today and for the interval studied. To my knowledge the WEP is not considered to be a stable oceanic environment over time.

**Response 5:** At present, Sites 806 and 807 are in quasi-equilibrium with the atmosphere (annual average offset of +28 ppmv; Takahashi et al..; Tripati et al., 2009, 2011; Shankle et al., 2021). This region is considered to be warm and thermally stable. Our calculated pre-industrial $pCO_2$ is 298 ppm (calculated using the GLODAP database and corrected for anthropogenic values) which when compared to the Vostock value of 282 ppm at 1.08ka to a value of +16ppm. This $pCO_2$ difference is similar to our $pCO_2$ uncertainty (~17ppm on average before taking into account the $\delta^{11}B_{sw}$). This is why we assumed the site was near air-sea equilibrium with the atmosphere.

A few lines of evidence suggest the region was in quasi-equilibrium in the past: 1) zonal temperatures are at a maximum in pre-industrial times and we are able to reconstruct atmospheric $pCO_2$ values, 2) the compilation of temperature proxies shows a weak and stable zonal temperature gradient from ~12 Ma to early Pliocene which would support air-sea stable conditions and air-sea equilibrium (Nathan and Leckie, 2009; Zhang et al. 2014; Liu et al., 2019).

We have added discussion of this to the manuscript on Line 127. "These two sites have been examined in other boron-based studies (Wara et al., 2003; Tripati et al., 2009, 2011; Shankle et al., 2020), as has the region more broadly (Pearson and Palmer, 2000), because they are understood to be in equilibrium with the atmosphere and have relative stable hydrography. The region experiences equatorial divergence but is not strongly affected by upwelling and has a current estimated annual air-sea $CO_2$ flux of +28 ppmv (Takahashi et al., 2014). The pre-industrial air-sea $CO_2$ flux is calculated to be +16 ppm, (GLODAP database corrected from anthropogenic inputs), with a value of 298 ppm, compared to the Vostok ice core value of 282 ppm at 1.08 ka . This $pCO_2$ difference is similar to our $pCO_2$ uncertainty (an average of ~17 ppm for the youngest samples). If trade winds were much stronger, and equatorial divergence greater, in the past, than this could drive some disequilibrium. However, a few lines of evidence suggest the region was in quasi-equilibrium in the intervals we examine: 1) zonal temperatures are at a maximum in pre-industrial times and during the Pleistocene, and we are able to reconstruct atmospheric $pCO_2$ values from the ice cores, 2) temperature proxies indicate the region is relatively stable with respect to temperature compared to other parts of the ocean, and also indicate a weak and stable zonal temperature gradient during the Miocene and Pliocene which would support air-sea stable conditions and air-sea equilibrium (e.g., Nathan and Leckie, 2009; Zhang et al. 2014; Liu et al., 2019).

This work builds on low-resolution prior reconstructions for these sites (Wara et al., 2003; Tripati et al., 2009, 2011; Shankle et al., 2020), Site 872 in the tropical Pacific (Sosdian et al., 2018), and other published boron isotope work, to provide additional data to constrain past seawater pH and $pCO_2$ for the WEP using MC-ICPMS, thereby providing an invaluable new perspective on reconstructing past atmospheric $CO_2$ via marine sediment archives."

Line 29: 'reproduce the ice core record' is very strong language for the comparison which has been carried out. There are only 16 points and no comparative data is produced (e.g. numerical data or crossplotted data).

**Response 6:** We have modified the sentence, line 30 now reads "We use high-precision multi-collector inductively-coupled plasma mass spectrometry and show that data from these sites are consistent with ice core data and other boron-based studies."

We also have added crossplots to Fig. 3, in order to quantitatively compare our data to ice core data. Fig. 3E does not show either a significant difference between the slope or intercept with a 1:1 line.

Line 308: "Crossplots comparing our data are presented in Figs. 3C, 3D, 3E; the slope and intercept are not statistically different from a 1:1 line (p=0.69 and p=0.48)."

Line 31: The Miocene data is higher than other published data, but so is the Pliocene, and arguably the latter is much more important as more data is available to facilitate the comparison.

**Response 7:** As show in Figures 6, 9, and 10, our data for the Pliocene overlap with published estimates from Sosdian et al. (2018) for the Pliocene and Miocene when uncertainties are considered, But Sosdian et al. (2018) do not use the same second carbonate parameter (e.g. DIC) and while Rae et al. (2021) data are higher in the Miocene due to the use of different scenarios of seawater $\delta^{11}B$ and alkalinity. For the Pliocene, our reconstruction is marginally higher than these two studies, but when uncertainties are compared, they overlap.

We also compare our results in Figure 8 to recalculated values from Sosdian et al. (2018) using the same $\delta^{11}B_{sw}$ scenario that we are using in this study. As observed in Fig. 6 or S2, the differences between reconstructions mostly arise in the Miocene, that are eliminated if the same methods are used (Figure 8).

Line 33: A 270 ppm transition during the Pliocene iNHG would be very interesting information. This is a huge change (effectively one halving of CO2). Please discuss this more in the context of the other records if you find your Pliocene data to be valid.

**Response 8:** We now write line 505: "The $pCO_2$ concentrations that we calculate indicates a reduction to 350 ppm by 2.7 Ma, ~280 ppm by 2.6 Ma, and ~210 ppm by 2.4 Ma, in several steps. These results support roughly a halving of $CO_2$ values when compared to values of ~530 ppm at 3.3 Ma. These values are consistent with the $pCO_2$ thresholds proposed by both DeConto et al. (2008) and Koening et al. (2011) for the intensification of Northern Hemisphere glaciation and the low atmospheric $CO_2$ (280 ppmv) scenario from Lunt et al. (2008). Mg/Ca SST decline from 30°C to 26°C, supporting an Earth System sensitivity of ~4°C/doubling of $CO_2$ over this range, although given uncertainties, higher values of ~6°C/doubling of $CO_2$ that have recently been proposed (Tierney et al., 2020) can not be excluded."

Line 68: There is no atmospheric $CO_2$ data from the Pliocene available in the blue ice cores, although they did confirm the presence of ice which is of that age. Please correct this.

**Response 9:** Thank you. We have changed the "Pliocene period" to "the early Pleistocene period".

Line 72: Foster 2008 is not a B/Ca to $CO_2$ paper, and most recent studies have stopped plotting the B/Ca datasets as there were found to be too many divergent controls on the incorporation.

**Response 10:** We wanted to acknowledge the different proxies used for $CO_2$ reconstruction. We agree that the reconstruction from Foster et al. (2008) is primary based on $\delta^{11}B$ but that study does actually use B/Ca to constrain $[CO_3^{2-}]$ as a second carbonate parameter. To acknowledge this we also added this reference when citing the boron isotopes based studies.

Lines 94-102: Use Hain et al. 2018 for the strongest case that $\delta^{11}B$ can be a viable CO2 proxy, regardless of other uncertainties.

**Response 11:** Thank you! We have added the reference.

Lines 103: Here you refer to various $\delta^{11}B$ studies as 'high resolution' and yet earlier refer to the ice cores as 'relatively high resolution', I would suggest being more consistent within the manuscript regarding what is 'high resolution' given the timescales you are talking about. The ice cores, Martinez-Boti, Dyez, Chalk, de la Vega and Greenop studies are high resolution compared with your work here, but Foster, Hoenisch, Sosdian are more similar to your new records. Consistency with this will stop some of the false equivalency made about resolution in this paragraph.

**Response 12:** We removed the "Relatively" from "Relatively high-resolution" for the Vostok ice core. We have modified the statement in Line 106 to read "Given the evolution of the field, there are relatively few studies generating high-precision boron-based records over major climate transitions in the Cenozoic using recent analytical methods, that incorporate our current understanding of the proxy (e.g., Greenop et al., 2014; Martinez-Boti et al., 2015b; Chalk et al., 2017; Dyez et al., 2018; de la Vega et al., 2020).".

Line 126: These studies are problematic in their interpretation of CO2 data and call into question the assurance that these sites have remained in equilibrium. This point is repeated on line 152 but without an explanation to the reasoning behind. Given that this is the key assumption in this manuscript I think it deserves more attention.

Response 13: Comparison with the recent boron isotope-based $pCO_2$ studies including compilations from Sosdian et al. (2018) and Rae et al. (2021) shows that our data are in broad agreement, as discussed in our response above. We do acknowledge that the models used ($\delta^{11}B_{sw}$ and alkalinity) for each reconstruction induces important variability (Fig. S2), and we show that when using a similar set of assumptions, our results are similar to work from Sosdian et al. (2018) for Site 872 in the region. We also now show the raw data, as suggested, and the pH estimates, that further corroborates the fidelity of our reconstruction. With respect to air-sea equilibrium, we have expanded the discussion, as described in an earlier response. There is no evidence or physical oceanographic or biogeochemical explanation for why the region we are examining should have been in more disequilibrium than other regions examined, which is why it has been the target for other $pCO_2$ work (Pearson and Palmer, 2000; Wara et al., 2003; Shankle et al., 2020). For example, the EEP/Carribbean did experience major gateway changes that are much more likely to have influenced the hydrography of the region over the timescales examined. Other areas may have experienced upwelling changes.

Line 155: How much could disequilibrium impact these data? No reference is made to preservation or potential diagenesis changes that may impact the data to a far larger extent.

**Response 14:** Please see our above response on disequilibrium, and the new text starting on Line 135. We have now added several sections that relate to preservation, including the section described above in the methods (section 2.2)

Line 160-166: The age models are not great for these cores, but I do not think that matters given the resolution of the data. It may be worth stating here that no direct comparison of ages between the cores is made

**Response 15:** We changed the age model of site 806 and used the polynomial regression developed in Lear et al. (2015) based on biostratigraphy.

Line 198: "The age model for Site 806 from 0-1.35 Ma is based on Medina-Elizalde and Lea (2005) which corresponds well with ages from the Lisiecki and Raymo LR04 stack (Fig. 2A). The fourth polynomial regression-based biostratigraphy from Lear et al. (2015) was used for the rest of the record, following

other work (Sosdian et al., 2020). Ages for Site 807 are based on published biostratigraphy (Berger et al., 1993) for 807 with additional constraints placed by Zhang et al., (2007) for the interval from 0-0.55 Ma."

Line 170: Missing accent on Plouzané.

**Response 16:** This has been corrected.

Line 194: Gutjahr et al 2020 is the updated reference for this.

**Response 17:** This has been corrected.

Line 197: This is great, where is the data though? Can you add a supplementary table?

**Response 18:** We originally added the data as a supplementary table. You can find it in Table S4 and S5 in the supplemental information.

Line 210: you have 'less than' and '<'.

**Response 19:** We removed the "<". This sentence now reads "contributed to less than 1% of the sample signal".

Line 217: Why is the 2SD for the $\delta^{11}B_{NEP}$ so much larger than for the other data despite the increase in n?

**Response 20:** Good question, so the data come from independent microdistillations which took a small amount of powders each, we think the largest variability can be due to homogeneity in the NEP powder, as the same variability have been observed both at the University of Cambridge and at IUEM for various projects. Similar results were reported by Sutton et al. (2018). Note that the difference was not observed in McCulloch et al. (2014), Rapid, high-precision measurements of boron isotopic compositions in marine carbonates). The JCP-1 seems more homogeneous and is reported around 0.2 permill.

We now provide a table of those standards in a table in the supplementary materials (Table S4).

Section 3.1: This needs to be much more quantitative if you are going to use it to pin all of your results on.

**Response 21:** We now add a few crossplots in Fig. 3 in order to make it more quantitative, the slope and intercept of the linear regression between $pCO_2$ from ice cores and from boron isotopes (Fig. 3E) is not significantly different from the 1:1 line.

Line 309: These alkenone records are out of date, please refer to Tanner et al 2020 and Super et al 2018 for updated interpretations.

**Response 22:** We added those references in the text. We made a compilation of Mg/Ca based SST using our framework in Figure. 4, we also added a section to discuss changes in temperature where we added the work of Super et al. (2018). We have added Section 3.3 : "3.3 Sea surface temperature in the WEP" . We also added the Tanner et al. (2020) $pCO_2$ reconstruction based on alkenones to the Figure 6.

Line 420: "However, reconstructions are still few and discrepancies between boron isotopes and alkenones based reconstructions lead to uncertain $pCO_2$ history. Nevertheless, recent literature was able to get coherent reconstructions from both proxies (Rae et al., 2021). To date, boron isotopes and alkenone-based $pCO_2$ reconstructions support higher $pCO_2$ during the MCO and a decrease over the MMCT (Sosdian et al. 2018; Tanner et al., 2020)."

**Response 23:** We have modified this section. It now reads line 407:

"However, reconstructions for the Miocene are still relatively limited (Sosdian et al., 2018; Rae et al., 2021). Current boron isotope and alkenone-based $pCO_2$ reconstructions support higher $pCO_2$ during the MCO and a decrease over the MMCT (Sosdian et al. 2018; Stoll et al., 2019; Tanner et al., 2020), consistent with what was previously inferred from B/Ca (Tripati et al., 2009, 2011).

We applied the same framework we used for calculations at Sites 806 and 807 to published boron isotope data from Site 872 (Sosdian et al., 2018) in order to extend the WEP record to the early Miocene (Figs. 7, 8). The Miocene data between Sites 806 and 872 do not overlap as both are low in resolution, but do show excellent correspondence in their trends in $\delta^{11}B$ and reconstructed pH. The pH values we reconstruct are within error of published estimates from Site 872 (Sosdian et al. 2018, Figs. 7D and 8D). Collectively, these data suggest the early Miocene WEP was characterized by a mixed-layer pH of 8.1 ± 0.1 (2 SD, n=4) between 19.4 and 21.8 Ma, which decreased to reach a minimum during the MCO of 7.7 ($\pm^{0.11}_{0.14}$).

Given the sensitivity in absolute $pCO_2$ to assumptions about the second carbonate system parameter, a few scenarios were explored for the combined 806/807/872 reconstructed pH values. For all alkalinity scenarios we used, reconstructed $pCO_2$ shows an increase from the Early Miocene to the MCO, with the highest values in the MCO. Recalculated $pCO_2$ for Site 872 between 19.4 and 21.8 Ma is 232 ± 92 ppm (2 SD, n=4), lower but within error of the ones presented in Sosdian et al. (2018) and also within error of a constant alkalinity scenario (Fig 8D). The main difference between reconstructions is when comparing the same data recalculated in Rae et al. (2021) that show higher $pCO_2$ between 19.4 and 21.8 Ma, with an average value of 591 ± 238 ppm (2 SD, n=4) for Site 872, because of the different assumptions used in their study and ours. This difference is important because that would imply a relatively high and stable $pCO_2$ from the early Miocene to MCO, which would imply a decoupling between $pCO_2$ and temperature with no $pCO_2$ change during an interval of decreasing benthic $\delta^{18}O$. However, our reconstructed $pCO_2$ increase towards the MCO is in line with the observed benthic $\delta^{18}O$ decrease and $\delta^{13}C$ increase and suggest a coupling between temperature and $pCO_2$ over this period. We note that overall, Mg/Ca-SSTs are warm (>32 ºC), and there are relatively small changes in Mg/Ca-SST from the early Miocene into the MCO."

Line 314: The S18 study is a reinterpretation of the same data from the other two, so this sentence either needs to explain that or just use the most recent iteration.

**Response 24:** Line 442, we changed for "Published $\delta^{11}B$-based reconstructions also support higher $pCO_2$ for the MCO of ~350-400 ppm (Foster et al., 2012) or 300-500 ppm (Greenop et al., 2014) that was recalculated by Sosdian et al. (2018) to be ~470-630 ppm depending on the model of $\delta^{11}B_{seawater}$ chosen."

Line 321: This section is not a fair representation of the existing data. The $\delta^{11}B$ presented here is ~13.8 ‰ and the minimum in other records e.g. S18 is ~15 ‰, in addition, the data of S18 from ODP 872 is geographically very close to ODP 806. ODP 871 from G14 appears to match the data in S18 quite well, which would imply that another reason is responsible for the difference seen here between 872 and 806 (ocean frontier, preservation or analytical). In addition, G14's raw values (for *T. trilobus*) are fairly similar to the data presented here. This would then suggest the reason for the difference is in interpretation and calculation. I think plotting the available data in raw $\delta^{11}B$ and either recalculating or plotting calculated CO2 would really help with this point. There is not a huge amount of data for this period and it is worth discussing fully where it does and does fit.

Response 25: Raw data $\delta^{11}B$ data are presented in Fig 2. We now provide a comparison of our calculated data with recent compilation of S18 and Rae21 in several places (see Figs. 5, 6, 7 and 8, S1, S2). We

believe that those figures show the variability of the $pCO_2$ values calculated depend on the assumptions made for alkalinity and $\delta^{11}B_{sw}$.

Data from Site 872 were used to extend our record to the early Miocene and were reprocessed using our framework. The closest datapoint we have is Site 806 (15.6 Ma, $\delta^{11}B$=14.47+/- 0.21 permill) and Site 872 (16.7 Ma, $\delta^{11}B$=15.12+/- 0.25 permill) which have boron isotope signatures close to each other which support a minor effect of diagenesis and the variations a real environmental change which would be in line with the $\delta^{18}O$ benthic slack.

Our $\delta^{11}B$ data are typically lower than published data from other ODP sites (926, 761, 1000) located in different oceans basins. While there is no data overlapping in time period between Sites 872 and 806, both sites are located in the WEP, and both records can be combined to provide a continuous record and the additional data provides further constraints to the early Miocene period. To robustly do this, we applied the same framework to the $\delta^{11}B$ data from S18 from Site 872 (Figs. 7 and 8). Those data are in absolute values, slightly lower than ours, but that is to be expected if $CO_2$ was changing in a manner similar to what is seen in the benthic $\delta^{18}O$ stack. The combined records would suggest a larger $CO_2$ increase than recalculated from S18 alone, from the early Miocene into the MCO.

All three sites are located above the lysocline, and please see section 2.2 for the discussion about diagenesis.

Line 327: I'm not an expert on this topic but I think both of these ideas have been updated in Stoll et al 2019 and Tanner at al. 2020.

**Response 26:** Yes thank you for this remark, we have updated this and cited those references, as described in Response 23.

Line 328: these very warm temperatures also appear in the Atlantic TEX86 study of Super et al. 2020 which you could cite here.

**Response 27:** Thank you, we added the reference to section 3.3.

Line 363: Please given the 'marginally consistent' value here as well, rather than just the inconsistent. Also see S18 for reconciliation between the datasets of MB15 and B11.

**Response 28:** We now present comparisons with S18 and Rae201, and we removed this sentence.

Line 367: de la Vega capitalisation.

**Response 29:** We corrected it.

Line 371-373 : please expand on the good agreement here as above. Stap et al. 2016 does not agree particularly well with the other studies.

**Response 30:** The $pCO_2$ reconstructed from the original paper from Stap et al. (2016) appeared to match our reconstruction, however no overlapping data exist. Nevertheless, we note that the data reprocessed by Sosdian et al. (2018) lead to significant changes and lower $pCO_2$ values from Stap et al. (2016) compared to their original publication. We removed Stap et al. (2016) from the references of this sentence.

Lines 376-384: This section reads like it should be in the introduction rather than results.

**Response 31:** As with the other sections in the results and discussion, as we step through each time interval, we provide a broader perspective on the interval for the paleoceanographer/paleoclimatologist to contextualize the data

Line 385: 150 ppm is a lot of disagreement, and as stated it is down to the raw $\delta^{11}B$ values. I do not follow the vague argument about disequilibrium that has been made several times now.

As in theory, increased upwelling, increased respired carbon dissolved in surface water, reduce pH and increase CO2 estimate and vice versa.

One issue with 999 is the Panama Isthmus and potential influx of surface EEP water. Upwelling in EEP makes water more acidic therefore, if there were to be an increased influx into the Caribbean, this would reduce pH and increase CO2 estimates. For the inverse to happen, 999 would need to be a sink for CO2 so barring a huge change (e.g. reversal of AMOC), it is more likely that CO2 estimates at 999 will overestimate rather than underestimate CO2. With this in mind the difference between 806/7 and 999 would then require even more disequilibrium in the WEP.

I would favour that the Pliocene section here is more likely to have been impacted by diagenesis, or by the closure of the Indonesian through-flow, which is not discussed here.

**Response 32:** We agree that the variability in reconstructed $pCO_2$ mostly comes from the assumptions made for the reconstructions from the different studies. Our work on the EEP will help to shed light on the significance of differences in Site 999 compared to 806.

Please see Response 3 for the discussion of diagenesis, and Response 5 for discussions about air-sea equilibrium. We want to emphasize it is unlikely that the WEP is out of equilibrium with the atmosphere during the Pliocene, and the similarity in reconstructed values with Site 871 suggests that changes in Indonesian through-flow did not impact our site.

Line 389: I am not sure you have the resolution to say this, would suggest removing.

**Response 33:** We agree that we may not capture the minimum $pCO_2$ over those time intervals due to our resolution, so we removed this sentence and start the paragraph as followed:

Line 505: "The $pCO_2$ concentrations that we recalculated are consistent with the $pCO_2$ thresholds proposed by both DeConto et al. (2008) and Koening et al. (2011) for the intensification of Northern Hemisphere glaciation and the low atmospheric $CO_2$ (280 ppmv) scenario from Lunt et al. (2008)."

Line 401-405: Please reference this section. I would also incite the logarithmic nature of CO2 forcing here, see multiple papers e.g. MB15, C17, DIV20.

**Response 34:** We have added references to this section, including the three suggested references.

Line 415: All of these papers DO suggest a decline over the MPT, it is the main finding of all of them.

**Response 35:** We changed for Line 536: "Previous boron isotope studies for ODP Sites 668 and 999 in the tropical Atlantic Ocean have suggested that a decline in atmospheric $CO_2$ did occur during glacial periods in the MPT, but not during interglacials (Hönisch et al., 2009; Chalk et al., 2017; Dyez et al., 2018)."

Line 425: When is the end of the MPT, please define.

**Response 36:** We defined the MPT between 1.2 to 0.8 Ma and added this to the text.

Lien 427-437: This whole paragraph is extrapolated from one point. I think just show you agree with the data of H09 and move on.

**Response 37:** We have edited this section to say: "While data resolution are limited, we speculate this could explain why glacial/interglacial amplitudes in WEP $pCO_2$ values decrease from the MPWP towards the Pleistocene, whereas variations in $\delta^{18}O$ are increasing – a speculation that could be tested with increased data resolution."

Line 466: Mejia et al 2017 would be a useful reference here.

**Response 38:** We removed this section following comment from reviewer 2, we did not include Meija et al. (2017) because reconstruction is based on Site 846 located in the EEP, influenced by upwelling.

Line 472: exchange 'many others' for an e.g. prior to the reference list, or give a few more.

**Response 39:** We removed this section following comment from reviewer 2.

Lines 497-500: This is precisely the point of H09, C17 and D18, I do not think that the addition of one data point allows the confirmation of these claims.

**Response 40:** Our data support this statement, but it is true that our resolution is low. We now state line 544: " Although our data are relatively limited, we note they have greater resolution for the middle and later part of the transition than prior publications that have drawn conclusions about the MPT (Hönisch et al., 2009; Chalk et al., 2017; Dyez et al., 2018) (Fig. 10D) and therefore we explore their implications."

DlV20 study missing from Table 2, represents the key reference for the mid-Pliocene period. The ordering of this table is also confusingly out of temporal order.

**Response 41:** DIV20 is now included in Table 2 and this table is now in temporal order.

Figure 1: Watch the formatting on the scale axis for the map figure. Please add contours or change the colour scale to something more friendly for colour blindness. The x-axes are also cut off the other panels.

**Response 42:** We have slightly modified the colors.

Figure 2: Please add a legend to the plot to define the symbols and shades. It would be helpful to colourise your data and add raw data from other studies in grey behind to facilitate easier comparison. A d11Bborate plot would also be useful to account for the different calibrations between species.

**Response 43:** To allow comparison, we added the raw $\delta^{11}B$ data from other studies to Fig. 2 and have added color.

Figure 5: The updated ice core compilation of Bereiter et al. 2015 may facilitate a better comparison for the plot here, or at least including the WAIS divide data from Ahn et al. 2008. A cross plot or calculation of root mean squared error or similar would be a valuable addition.

**Response 44:** We updated the ice core compilation using the composite of Bereiter et al. (2015).

Figure 7: S18 data is missing from this plot.

**Response 45:** We added the S18 data to all graphs.

Figure 9: Much is made of the low CO2 found in MIS 30, but this figure shows lower(?) CO2 at 1-1.1 Ma which is not discussed.

**Response 46:** This decrease between 1-1.1 is MIS30 which we now explicitly mention in the text. Line 550: "We also find evidence that during the MPT, glacial $pCO_2$ declined rapidly from 189 ($\pm$30) ppm at MIS 36 (Chalk et al., 2017) to reach a minimum of 170 ($\pm^{52}_{24}$ ) ppm during MIS 30" and line 555: "In our record for the last 16 Myr, the lowest $pCO_2$ is recorded at MIS 30 during the MPT, with values of 164 ($\pm^{44}_{35}$) ppm, which supports an atmospheric $CO_2$ threshold that leads to ice sheet stability. During this transition, the $pCO_2$ threshold needed to build sufficiently large ice sheets that were able to survive the critical orbital phase of rising obliquity to ultimately switch to a 100 kyr world, was likely reached at MIS 30."

Figure 10: Again S18 is missing from this figure, despite it being the most complete Neogene study to date (including this one!)

**Response 47:** We added the S18 data to all graphs.

**Reviewer 2**

The manuscript by Guillermic et al presents new Mg/Ca and d11B measurements of planktonic foraminifera spanning the last 17 million years from the Western Pacific Warm Pool at ODP 806 and 807 with the aim of estimating past evolution of Sea Surface temperatures and atmospheric pCO2. The most significant new contribution of this study is the addition of measurements between 5 and 17 Ma, as the majority of previous d11B measurements since the mid-Miocene are concentrated in the last 4 Ma whereas low resolution data illustrating long term trends in the 4 to 17 Ma time window are to date more limited. The estimation of pCO2 from d11B of foraminifera in this time period is sensitive to a number of uncertainties, including the assumptions of the evolution of d11B of seawater and alkalinity over time. This contribution employs the current best estimates for these parameters and illustrates the sensitivity of the pCO2 estimate to uncertainties in these parameter choices.

While the processing of the new data is clear and uses up to date alkalinity and d11B seawater estimates, the comparison with previous pCO2 estimates is not presented as clearly as it could be, and the discussion of phytoplankton (alkenone) pCO2 estimates for the Miocene is not up to date. I summarize the main content issues which I believe need to be addressed for this manuscript to provide a coherent step forward in understanding of pCO2 in this time interval. Subsequently, I have some suggestions on the organization and structure which I propose could improve transparency and clarity in the manuscript, as well as some more detailed comments.

Content and interpretation:

While the authors present for their own data an updated estimation of the CO2 considering recent proposals for d11B seawater (eg Greenop et al 2017) and alkalinity history derived from Caves 2016 or Zeebe 2005, they do not include in figures (and therefore not thoroughly consider) the published d11B-CO2 estimates which have been recalculated with these same alkalinity and Greenop et al 2017 d11B seawater parameters, namely those compiled and homogenized in Sosdian et al 2018. This is an essential update to make so that the new data from ODP 806 and 807 can be considered in context, and so that the new data contribute to an integrated better picture of the trends and absolute values. In particular, the latest Pliocene CO2 trend is quite clear in the Sosdian et al compilation. Additionally the early Pliocene and Miocene values in the Sosdian et al compilation are higher than in the original publications, and therefore are more consistent with the results as plotted here. Otherwise there is a disconnect between the figures (not homogenized parameters) and the textual mention of pCO2 values recalculated by Sosdian et al 2018. Once this is updated in figures, the discussion should also be updated and clarified.

**Response 1:** We have made these changes. We added the updated reconstruction from Sosdian et al. (2018) to our figures and compare with our data. We also added a comparison to the compilation published by Rae et al. (2021).

During the Pleistocene period, the reconstruction from Sosdian et al. (2018) does not reproduce the absolute $CO_2$ values from the Vostok ice core. Their results can be offset by ~ +60 to +100 ppm. Their divergence for the Pleistocene is important because it means they have higher absolute $pCO_2$ values for the early Miocene, after adjustment. However, their results are really similar to our reconstruction for the Pliocene and the Miocene periods.

We also present the recalculated data from Rae et al., 2021 for the Pleistocene and Pliocene periods which does accurately reconstruct ice core $pCO_2$ values, and for this interval, choice of alkalinity and $d^{11}B_{sw}$ scenarios have a minimum effect.

The estimation of tropical SST over this time interval is not trivial given the uncertainty in seawater Mg/Ca, and I think this warrants further clarification and transparency. An inferred seawater Mg/Ca history is sketched in Figure 2 but the Figure caption does not specify the origin of this curve. The calculation equations are provided in supplement but the input data on seawater Mg/Ca should be illustrated along with the data forming the basis for its estimation (eg the basis from which it is derived). Figure 7 illustrates some uncertainty around the SST, but the Figure legend does not indicate what is represented by this uncertainty. To what extent does the uncertainty in calculated SST (e.g choice of Mg/Ca seawater correction, and regression) affects the estimated pCO2 due to solubility? Also, have the authors considered the influence of a pH correction to the Mg/Ca SST, as conducted in recent study (Sosdian and Lear, 2020) and shown to be significant across the MCT (Leutert et al 2020)?

**Response 2:** The $Mg/Ca_{sw}$ were based on polynomial regression based on Gothmann et al. (2015) data, however, to be consistent with the literature and because variations were in line with our first regression we now utilized the fourth-order polynomial regression from Sosdian et al. (2020), data and figures are updated. However, the overall difference between the two data processing were <1°C. We applied a pH correction to *G. ruber* utilizing the sensitivity published in Gray and Evans, (2019). We used an iteration based on pH and temperature until difference was negligible. *T. sacculifer* did not yet proved to be influenced by pH so we did not apply a correction. Salinity correction was applied for both species. We now present the Mg/Ca data along the SST reconstruction in Fig. 2C and recalculated the temperature based on published Mg/Ca values at site 806 and 872 (Fig. 4). The temperature compilation was realized using Mg/Ca at Site 806 (Wara et al., 2005; Nathan and Leckie, 2009; Tripati et al., 2009) and Site 872 also located in the WEP (Sosdian et al., 2018).

The references on the alkenone pCO2 reconstruction are not up to date. Lines 308-310 refer only to older publications based on a theoretical diffusive model of CO2. Recent metanalysis of culture carbon isotopic fractionation (epsilon p or ep) data suggested that due to the operation of carbon concentrating mechanisms, ep exhibits a much lower sensitivity to CO2 than originally inferred; application of the sensitivity observed in cultures to sedimentary ep measurements yields a significant pCO2 decline since the mid-Miocene (Stoll et al., 2019). This low ep sensitivity is supported by recent determinations over glaial cycles (eg Badger et al 2019) and further suggests significant pCO2 decline in the late Miocene (Tanner et al., 2020) which would be a relevant reference for comparison in section 3.4. A detailed updated discussion is provided in Rae et al, 2021.

**Response 3:** Thank you for this comment, we updated this paragraph with those references, and we also added Tanner et al. (2020) data to Figure 6.

Line 420: "However, reconstructions are still few and discrepancies between boron isotopes and alkenones based reconstructions lead to uncertain pCO₂ history. Nevertheless, recent literature were able to get coherent reconstructions from both proxies (Rae et al., 2021). To date, boron isotopes and alkenone-based pCO₂ reconstructions supports higher pCO₂ during the MCO and a decrease over the MMCT (Sosdian et al. 2018; Stoll et al., 2019; Tanner et al., 2020)."

The discussion of previously published d11B records is in many places overly superficial. For example in lines 314-316, previously published d11B records of the MCO (Sosdian et al 2018, Greenop et al 2014) are diminished in importance by suggesting " it is unclear if these values accurately reflect the atmosphere given the sites may or may not have been in equilibrium with the atmosphere.." The cited studies reflect multiple sites (ODP 926, 999, 668, 761..), all in comparably reasonable locations to be close to equilibrium with the atmosphere. Unless the authors would like to present clear new evidence that some of all of these sites are less likely to have remained in equilibrium with the atmosphere than ODP 806 or 807, the original interpretation that these sites (of preindustrial pCO2 disequilibrium <25 ppmv) remained close

to equilibrium should be respected, and other potential explanations for the differences should be explored.

**Response 4:** We now present a comparison with data from Sosdian et al. (2018) and Rae et al. (2021). It is clear that the choice of $\delta^{11}Bsw$ and alkalinity scenarios is critical and lead to most of the variability we observed in $pCO_2$ reconstructions between studies. We then removed the disequilibrium argument from this paper as there in no strong evidence for it, especially since we can reconcile data from site 761 and our WEP data.

Site 806 and 807 are sites estimated to have fast rates of diagenetic recrystallization (Mitnik et al 2018). For example, averaged over the upper 80 m of sediment (appx 3 million years given sedimentation rates), authigenic carbonate is estimated to comprise 19% of total carbonate at 806 and 36% at 807; in comparison other sites like ODP 999, the authigenic carbonate is <1% of total carbonate in the same depth and time interval. It might be helpful for the authors to acknowledge this and comment on evidence for how this may or may not affect the 11B and Mg/Ca results of the planktic foraminifera.

**Response 5:** We now present the raw boron data from this study along with other boron isotopes based studies (Figure 2). In our study, the boron isotopes data at sites 806 and 807 are lower than other sites.

The samples picked were all visually preserved, however no SEM images were obtained during this study to assess potential recrystallization or dissolution, we only used the weigh/shell to monitor dissolution. This record is shown in Fig. 2 and present no decreasing trends toward the Miocene which is not in favor of sample dissolution over the time study. Previous literature observed a significant positive relationship with test size and the boron isotopes in coretop samples (Hönish and Hemming, 2004; Ni et al., 2007), to overcome this problem we applied a correction based on empirical relationship between the weigh/shell and the boron isotopes (see supplement).

We acknowledge that the % of authigenic carbonate at sites 806 and 807 is important. However, the lack of porewater boron and boron isotopes data at those sites and the high variability in boron isotopes pore water profiles at other sites (Brumsack and Zuleger, 1992) do not permit to conclusively state a potential issue specifically at these sites. However, considering the proportion of authigenic carbonate, diagenetic processes are happening at those sites, which can effectively affect our data through partial dissolution or preferential dissolution of the test. Because our data are lower than other published data of ~2permill, and our samples going through rigorous cleaning, I would think that preferential dissolution of the heavy isotopes or preferential dissolution of ontogenetic calcite relative to the light gametogenic calcite could be relevant here. However, no trend is observed for the weigh/shell data and the empirical weigh/shell correction we are using is consistent for our $pCO_2$ reconstruction in comparison to the Vostok ice core (Fig. 5). Also, comparison between sites 806 and 807 shows that similar correction needs to be applied which is not going in the direction of a higher influence of authigenic carbonate at site 807. Moreover, recent study from White and Ravelo, (2020) tried to constrain the dissolution effect at site 806 by reconstructing deepwater saturation state through the Pliocene, no trend were observed in their record suggesting minor dissolution effect on the Mg/Ca. Despite evidence of diagenesis processes, the record suggests they have a minor effect on the $\delta^{11}B$ of our shells.

We have added section 2.2 "Preservation" starting line 171 to address the preservation of the samples.

The coherency of the d11B-pCO2 estimates with ice core pCO2 is always a useful comparison, but its effectiveness relies on the precision of the age model used for this portion of the sediment core (as well as the precision of the ice core age model, which cannot be investigated here). Particularly relevant to the last 800 ka, section 2.2. should detail on what the age models are based not just the publication source. From the reference cited for the last 1.35 Ma, it appears the age model is based on d18O of planktic G. ruber -

is it still tuned to SPECMAP chronology as in Lea et al 2000, or is it retuned to LR05? In Figure 5, I think it would be better to show the site 806 d18O G. ruber in the upper panel, eg the metric from the same site and age model as the d11B estimates, rather than the LR05. Then, I would suggest in addition to the time series, a scatterplot of the d18Oplanktic vs d11B-based pCO2 from 806 (assuming that d18O is available for the same core intervals - this gives an estimation of the coherency of pCO2 and glacial cycles in the same core without age uncertainties; were any d18O made on 807?), and also importantly a scatterplot of the d11B-based pCO2 vs ice core pCO2 .

**Response 6:** The age model used between 0-1.35 Ma at site 806 is from Medina-Elizalde and Lea, (2005). We added the high resolution d18O *G. ruber* from the same study in Figure 3 alongside with the stack from Lisiecki and Raymo, (2005) for comparisons. The two records are showing good agreement with each other even if not tuning against each other but are resolving the IG/G cycles. The ages of the deeper samples were calculated based on the fourth-order polynomial regression based biostratigraphy events (Lear et al., 2015). Considering the low resolution in our samples and the availability of d180 data we still present our data with the Lisiecki and Raymo, (2005) and Zachos et al. (2008) stacks as observed in other studies ( Sosdian et al., 2018, 2020).

For site 807, between 0-0.5 Ma, Zhang et al. (2007) derived an age model based on the correlation of the oxygen isotopic curve of C. wuellerstorfi to the SPECMAP stack (Imbrie et al., 1984). In addition, they used one planktonic foraminiferal δ18O event and two nannofossil datum levels (Prentice et al., 1993). Ages of the deeper samples were calculated based on linear interpolation of the biostratigraphy events.

We have added few crossplots in Fig. 3, in order to have a better quantitative comparison between our data, the ice core data and also the d18O. No linear regressions were significant (p<0.05) but the crossplot Fig. 3E does not show either a significant difference between the slope or intercept with the 1:1 line.

Line 308: "Crossplots comparing our data are presented in Figs. 3C, 3D, 3E; the slope and intercept are not statistically different from a 1:1 line (p=0.69 and p=0.48)."

Suggestions on organization:
I recognize the challenge of illustrating the effects of possible assumptions of d11B and alkalinity on the final CO2 calculation, but I am not convinced the current organization is the most effective and it leads to an unusual ordering of figures. The Methods heading "2.7 " effectively starts presenting results and sensitivity analysis.

The authors might consider if a more direct presentation of results and discussion could:
a)  begin with section 3.1, and start with the current Figure 5 - the last 800 ka uses modern d11B sw and alkalinity so is not subject to the uncertainties/sensitivity analysis on both parameters .
b) continue with a section on the measured indices and summarizes the findings in of the current Figure 2 which presents the measured results

**Response 7: a) and b):** We started with Figure 2 showing the raw results, like this we can introduce the comparison between raw published data and talk about potential impact of diagenesis. Then we are presenting the comparison with the ice core data.

c) comment on the inferred trends in SST and uncertainties in their calculations , and comparison with other SST histories both from Mg/Ca (Sosdian and Lear 2020) as well as TEX86 (Zhang et al. 2014, ) Lines 286-288 needs to clarify if the measured Mg/Ca is consistent with the other published Mg/Ca, or if the calculated temperatures are consistent with the published Mg/Ca calculated temperatures; and in the

latter case, have the temperatures for all these studies been recalculated using the same assumptions of Mg/Ca seawater and temperature regression as used for the new data here?

**Response 8:** We now made a compilation of available Mg/Ca at site 806 and 872, all data are on the same age model and were recalculated using the same framework which implies a pH correction for *G. ruber* (only few data), a salinity correction for *G. ruber* and *T. sacculifer* and taking into account the secular changes in seawater Mg/Ca using the fourth-order polynomial relationship derived in Sosdian et al. (2020). The raw Mg/Ca data are presented in Fig. 2 and the temperature reconstruction in Fig. 4.

d) discuss sensitivity of Neogene pCO2 estimations to assumptions of d11B seawater and alkalinity, which could introduce the current Figures 3 and 4 as the sensitivity of the results to d11Bseawater and alkalinity (and is there sensitivity to SST) and incorporate the introduction to this currently in the methods section

-continue with the discussion of temporal trends in calculated pCO2

**Response 9:** Section about sensitivities have been moved as suggested. We believe we present clearly the impact of the different scenarios line 323 in section "Scenarios of reconstructions".

As the manuscript begins to go through the main pCO2 results, I am not fully convinced that the current organization of the discussion is the most straightforward and concise. In the current organization the authors new data seems like it gets buried within the discussion. If the current time interval based structure is used, I believe it would be useful if in each heading of the results/discussion section, the authors presented first the summary of their own new results, and followed it with comparison to other proxy pCO2 results and finally to climate.
Also, if organization based on time periods is used then clearer section headings are needed  For example 3.3 is "Miocene" but 3.4 is "Late Miocene" which is a period nominally included included within the Miocene heading. I am not sure if division of the Pliocene into the warmth then glacial intensification then Pleistocene (3 sections) is really needed to discuss the author's new data, as these are time periods with substantial previously published data and interpretations and the authors new data are largely consistent with and reproduce these earlier results.  Overall, I believe the discussion section can be streamlined and made more concise.  Some sections such as 3.9 seem very extraneous, as there is really limited new SST data and it is not coherently presented to evaluate east west gradients; I suggest this section be eliminated from the discussion.  Section 3.8 is not really clear in advancing a mechanism for the CO2 variation, and I suggest the key points might be effectively commented within the context of the Miocene and Pliocene sections of the text.

**Response 10:** We changed the heading for "Miocene", "Pliocene" and "Pleistocene". We focused on the data comparison with Sosdian et al. 2018 and Rae et al. 2021, especially for the early Miocene period. We removed section 3.9 from the manuscript as we focus on the data.
* * *
Detailed comments:

ALL of the figures should more accurately show the true data density, including Figures 3 and 4 - continuous fill patterns and no symbols is not a clear way to represent the data.  At minimum, symbols are bars are needed to show where there are datapoints (alternatively rather than complete shading, points with error bars could be illustrated to show the sensitivity).

**Response 11:** We added the datapoints in Fig. 3. We kept the format of Fig. 3 but error bars are presented in Fig. 4.

And in Figures 6-8 and 10, the broken shading at least alerts to the data gap, but I think it would be ideal to show no shading over the long intervals without datapoints.

**Response 12:** Figures now represent true data density, we left blank the periods between ~5-7, ~10-12 and ~14-15 Ma.

Please clarify the basis of the age model (eg benthic d18O, biostratigraphy, etc), Section 2.2 is not sufficiently clear.

**Response 13:** We believe we clarified the age model in the main text and response above.

Could a more direct heading for Methods section 2.6 be developed?

**Response 14:** We could reduce the method section but we think it is important to keep this section as it is for data validation from the community.
* * *
References cited (not cited in the manuscript):

Mitnick, Elizabeth H., Laura N. Lammers, Shuo Zhang, Yan Zaretskiy, and Donald J. DePaolo. "Authigenic carbonate formation rates in marine sediments and implications for the marine δ13C record." Earth and Planetary Science Letters 495 (2018): 135-145.

Rae, James WB, Yi Ge Zhang, Xiaoqing Liu, Gavin L. Foster, Heather M. Stoll, and Ross DM Whiteford. "Atmospheric CO2 over the Past 66 Million Years from Marine Archives." Annual Review of Earth and Planetary Sciences 49 (2021).

Sosdian, S. M., and C. H. Lear. "Initiation of the Western Pacific Warm Pool at the Middle Miocene Climate Transition?." Paleoceanography and Paleoclimatology 35, no. 12 (2020): e2020PA003920.

Stoll, Heather M., Jose Guitian, Ivan Hernandez-Almeida, Luz Maria Mejia, Samuel Phelps, Pratigya Polissar, Yair Rosenthal, Hongrui Zhang, and Patrizia Ziveri. "Upregulation of phytoplankton carbon concentrating mechanisms during low CO2 glacial periods and implications for the phytoplankton pCO2 proxy." Quaternary Science Reviews 208 (2019): 1-20.

Tanner, Thomas, Iván Hernández-Almeida, Anna Joy Drury, José Guitián, and Heather Stoll. "Decreasing atmospheric CO2 during the late Miocene Cooling." Paleoceanography and Paleoclimatology (2020): e2020PA003925.

---

## Referee Report (RR1)

Review for 'Atmospheric CO2 estimates for the Miocene to Pleistocene based on foraminiferal δ11B at Ocean Drilling Program Sites 806 and 807 in the Western Equatorial Pacific'

The authors have done a good job at dealing with the previous round of reviewers comments on this manuscript. The discussion and conclusions are more nuanced and more appropriate for the data as presented in this iteration.

I am pleased to see that most, if not all the suggestions from the reviewers have been taken on board as they were important for the accuracy and longevity of this paper. However, I have one new major concern with the manuscript that requires addressing prior to publication, alongside several minor points which are listed below.

My major concern is with the implications from the cross plot presented in figure 3 ( in particular E), these were added as part of the revision which is great, but they are not represented well in the text and the data need a significant reassessment. The r^2 of the 'validation' dataset presented in 3E is 0.09 and there is a p-value of .3 which is nonsignificant and cannot be used to reject the null-hypothesis. I understand the frustration of such figures, but they should be presented in the main test where the plot is mentioned and their ramifications discussed. Numerics like this would typically present difficulty for the rest of the study, but in this case I think there are some alternate options as I do agree that the data are probably better than these performed statistics would imply.

The authors should investigate other methods for assessing their ice core overlapping data (e.g. root mean square error etc.), perhaps focussing on identifying and removing outliers and further exploring the knock-on effects on the rest of the record. I.e. given the prediction error envelope (rather than just a fit error or an analytical and calculation error), what effect does that have on the Pliocene and Miocene $CO_2$ reconstructions? Is there an impact from using the Mg/Ca corrected script of Gray and Evans or the calibrations used (e.g. Raitzsch 2018 vs. Henehan 2013)? This is really crucial for our community understanding and confidence in the proxy data.

I am confused as to the point of the other two cross plots in figure 3, they are all mentioned in the test together but with little explanation as to why or what they show. The layout of the axes is also confusing, why is the same parameter plotted on the x axis in 3C and the y axis in 3d?

In the text (line 275) it is mentioned that only 2 of the data points in the validation dataset (n=16) are outside the uncertainty, this does not appear to be the case with the whole record in figure 3B or in 3E.

Minor points:

Line 33-35: There is a lot of detail here regarding the MPT in the abstract which is still not for me the major finding or thrust of the paper, would suggest slimming this down.

Line 42: some formatting inconsistencies throughout with italics on pCO2.

Line 49: Would 'Neogene' be a better key word than 'Miocene'

Line 70: In reference list like this which are incomplete add 'e.g.' in front.

Line 90: The TE part of the acronym TE-NTIMS is not defined.

Line 95: or if the disequilibrium is known. Not many of the sites used so far are in perfect equilibrium, but the stability is important.

Line 121: Is this 17ppm 1SD or 2SD?

Line 138: insert here if you applied the pH dependency.

Line 144: Hopefully all the figures are of merit, suggest changing to 'principal figures'.

Lines 176-182: In the discussion of age models it is apparent that the two sites have very different qualities of age model, what are the implications of this for age uncertainty?

Line 226: Was the JCp analysed clean or uncleaned?

Line 248: is the capital needed in Alkalinity?

Line 257: I am not clear precisely what time-adjacent means here, please be specific.

Line 264-265: This is a very important point, why can they be lower and yet produce similar estimates of $CO_2$? What is inside the calculation that yields this result? ($2^{nd}$ carbonate parameter?)

Line 270: n=16 here.

Lines 281: missing space between reconstructions and in

Line 312: odd bold formatting.

Line 339: Specify the subplot of Fig 6.

Line 344: Please point to the plot here. Greenop et al. 2017 (e.g. plot 6E)

Line 376-379: What is the point made here, who are we supposed to believe? Please detail the differences. The difference in the calculation of two papers from the same year is huge here >400ppm, and of crucial importance for the viability of the proxy.

Line: 461: MPWP format

Line 468: i.e.., format

Line 488: clarify ice sheet stability over multiple obliquity cycles, or just large sheet generation. They are still not inherently stable.

Line 529: reprocessing data?

Line 542: specify the MCO record produced in this study. Also another line of reasoning is required to explain the difference seen between sites, that cannot be due to the basalts unless you are implying a local effect/ teleconnection

Line 553: typically ESS or ECS not earth system climate sensitivity.

Line 817: format

Figures and captions:

Figure 1: Contours would be good to add to this plot, the rainbow colour bar is not good for colour blind accessibility.

Figure 3:  It is unclear to me what 3C and 3D are supposed to show and why they have been plotted. More explanation required as above. Also it would be easier to see and trends if the $\delta^{18}O$ ruber were plotted on the same axis in both figures.

Figure 4: What is orange?

Figure 6: The ice core reference is not correct.

Figure 7: The site 872 data is calculated by Rae et al. 2021 not from it.

Figure 9: Again ice core reference is not correct.

Figure 10: Rae's compilation is cited throughout these figures and while I appreciate it is difficult to cite all the original references on the longer timescale plots, I think it is unfair to not credit the original authors on any of these plots. None of the data is created by Rae et al. e.g. Figure 10 should credit Bereiter, Hoenisch, Chalk, Dyez, and Yan studies.

---

## Author Response (AR2)

DEPARTMENT OF EARTH AND SPACE SCIENCES
3806 GEOLOGY BUILDING
BOX 951567
LOS ANGELES, CALIFORNIA 90095-1567
TEL: (310) 825-3880
FAX: (310) 825-2779

Dear Dr. Fisher,

Thank you for considering our manuscript for publication in Climate of the Past (manuscript # cp-2020-158). We believe we have addressed the points raised by the reviewers.

We explore multiple regression methods and statistics on the comparison between ice core $pCO_2$ and our $\delta^{11}B$-derived $pCO_2$. We revised this section of the manuscript accordingly. We provide more context to the Li isotope comparison, on the hypotheses for the Miocene climate transition and the discussion of the Columbia River basalts and the Miocene climate optimum.

We want to thank the reviewers for their constructive comments.

Sincerely,

Maxence Guillermic and Aradhna Tripati on behalf of the co-authors

Dear authors

the revised version of your manuscript has now been seen by the two referees that also assessed the initial version. Both referees acknowledge the significant changes that you did in the revision, which over most parts meet the points of criticism initially raised by both the reviewers. Reviewer #2 has added a few more constructive suggestions for improvement in the second review, in particular on the long-term volcanic and weathering carbon imbalance, which I would ask you to include in your re-revisions. Both referees also make a few comments on language/textual improvements etc. (see also my editorial comments below). I would regard these points raised by the referees as minor revisions as they imply mainly textual changes and can be implemented in a straightforward manner.

However, referee #1 also raises a point on the overall accuracy/validity of your pCO2 reconstruction based on the validation against the ice core data and I have to admit this gives me some headache as well. According to the figure caption of Figure 3E your $r^2$ is only 0.09, i.e. this would imply that only 10% of the variance in your d11B-derived pCO2 could be explained by atmospheric CO2 as contained in ice cores. Is this correct or is the $r^2$ of 0.09 only a typo? Looking at this figure, I visually can see a clear correlation although with a regression slope larger than 1.

In the main text you also refer to some p values, but these numbers do not agree with the ones in the caption and I am unable to attribute the numbers to the respective subplots. Please double check all the numbers and change the text to make things crystalclear. Moreover, it is unclear to me what the uncertainties in the x and y component of the scatter plot 3E refer to.

In fact, some of the ice core $CO_2$ data (x-axis) have a larger uncertainty than the d11B-derived data points (y axis) which appears virtually impossible as long as this does not include dating uncertainties.

Please also mention in the caption which evaluation/corrections scenario you used for the d11B-derived pCO2 in Figure 3E.

**Response 1:** Following yours and the reviewers suggestions, we:

- Rechecked, and confirmed the original linear regression performed on the data gave an $R^2$ of 0.09 and a p-value of 0.25.
- Performed both an ordinary least squares regression and a Deming regression followed by bootstrapping (n=1000) in order to take into account the uncertainties in both ice core $CO_2$ and $\delta^{11}B$-derived $CO_2$. The slope and intercept are still within error of the 1:1 line, but the p-value is however still low and this is likely due to the limited resolution of the data. We have rewritten this section of the manuscript.
- We removed the p-values comparing the slope and intercept to the 1:1 line which may have induced some confusion.

  We note the uncertainties in $pCO_2$ from ice core were estimated based on the 2 SD of ± 1 kyr of the age from the sediment sample. Uncertainty for ice core $pCO_2$ is only larger than derived from boron isotopes at 227 ka because +/- 1 kyr in the record is tracking a steep $pCO_2$ transition. The uncertainty for the boron based data are from propagated uncertainties. Data include dating uncertainties, 2 SD of +/- 1 kyr.

Finally, your linear regression is suggesting a slope smaller than 1 although the range in the d11B-derived pCO2 is larger than the range in ice core CO2!!!??? This is likely due to you using a one-sided regression, which in case of considerably uncertainties in x and y underestimates the slope. In fact, you should use a 2-sided regression (York et al., American Journal of Physics, 2004) for this scatter plot. It is also puzzling that 50% of the data points do not fall within the prediction envelope, what is the error this envelope refers to? In any case, for the validation period of the last 800 kyr your data seem to be clearly related to atm. CO2 but seem to overestimate somewhat the true atmospheric CO2 concentration range either due to methodological, correction/evaluation, diagenesis or CO2 surface water saturation issues. Although this systematic overestimation must not hold back in time, it is important to openly communicate that the absolute levels of your pCO2 reconstruction may be biased. However, the relative changes on time scales of 1 Myr, where alkalinity changes do not play such a great role may be more reliable. The issue of uncertain absolute CO2 levels in the past is also nicely

illustrated in your evaluation scenarios, which show that the overall Miocene CO2 levels strongly depend on the assumptions made during the evaluation/correction of the data in particular on alkalinity. This is clearly an issue the d11B community will have to work on in the future.

**Response 2:** Thank you for this comment. In order to develop this section we now present in Fig. 3 a Deming regression with a bootstrapping method taking into account the uncertainties for both x and y axes. You can also find a table summarizing the outputs of the regressions (linear and Deming) in Table S6. The p-value is still not significant (p=0.25). However, the absolute values for reconstructed $\delta^{11}B$ overlap with the ice core record and the amplitude of glacial-interglacial variability is reproduced. Having a larger n with a higher resolution record would make this comparison with the ice core record more statistically robust, as would having more robust constraints on temperature.

In summary, I think the low correlation/weak validation of your data for the ice core period does not justify rejection of the manuscript as the paper, nevertheless, adds substantial information on pH and $CO_2$ changes over the last 20 Myr. However, I strongly feel that the discussion should be more open to the limitations of the reconstruction and the large uncertainties in the overall reconstructed CO2 levels. This is in my eyes overly optimistic in the current version. A more critical assessment of the overall levels especially in the conclusions and the abstract, would likely also satisfy the criticism of referee #1.

**Response 3:** We have revised this section as suggested, and the abstract and conclusions are also revised. The main section in the text now reads Line 261: "We sought to assess if there is evidence for air-sea equilibrium or disequilibrium in the WEP during the large amplitude late Pleistocene glacial/interglacial cycles, in order to validate our approach. We reconstructed $pCO_2$ for the last 800 kyr (n=16, Fig. 3). For the last 800 kyr, reconstructed $pCO_2$ values for Sites 806 and 807 are in the range from ice cores (Fig. 3, Petit et al., 1999, Siegenthaler et al., 2005, Lüthi et al., 2008; compilation from Bereiter et al., 2015). The two critical diagnostics we used for method validation are: 1) that the $\delta^{11}B$-based reconstruction of $pCO_2$ is consistent with ice core atmospheric $CO_2$ and 2) the boron-based reconstruction empirically reproduces interglacial-glacial amplitudes from ice cores. Fig. 3B shows that both of these criteria are met. We also created a crossplot comparing these two independent constraints on $pCO_2$ (Fig. 3C). Two regressions between ice core $pCO_2$ and boron-based $pCO_2$ are shown, with a simple linear regression (grey line) and a Deming regression that factors in error in variables (blue line), and bootstraps of outputs shown (n=1000, Figure 3C, Table S6). While slopes and intercepts are not statistically different from a 1:1 line, the regressions do not reach a high significance level (p=0.25); boosting the resolution of the record could help provide better constraints for this type of comparison. The age models for the site do not provide an explanation for this variability based on comparison of the benthic $\delta^{18}O$ records for both Sites 806 and 807 (Fig. 3A, Zhang et al., 2007; Lear et al., 2003; Lear et al., 2015) to the published isotopic stack (Lisiecki and Raymo, 2004). No significant difference in variability was observed at either site. We also note that reconstructed $pCO_2$ uncertainties (both accuracy and precision) could potentially arise from Mg/Ca-derived estimates of temperature; these uncertainties could be reduced using independent temperature proxies for the WEP such as clumped isotope thermometry (Tripati et al., 2010; 2014), a technique that is not sensitive to the same sources of error as Mg/Ca thermometry, and therefore is an area planned for future work. Other sources of uncertainty that have a larger effect on $pCO_2$ calculations are the weight/shell correction, while the TA and seawater boron isotope composition have a minor effect over this time interval."

I therefore would urge you to include such a more differentiated discussion in the re-revised version. Please add a point-to-point reply to the points raised by the referees and me in this second round of reviews and a track change version of any re-revised manuscript relative to the current revised version. Your replies will decide whether another independent review of the paper is needed. Please find below a few more technical editorial comments that I would ask you to include as well:

- at several instances in the paper you refer to the ice core data as "Vostok data". This is incorrect as only the data younger than 420 kyr are from Vostok, the older data is from the EPICA Dome C ice core. Please change this throughout the manuscript by just referring to "ice core data" and providing the relevant references (Petit et al., 1999, Lüthi et al., 2008, Bereiter et al., 2015). In the main text a

reference such as "as compiled by Bereiter et al., 2015" may be sufficient in many cases but at least in the figure captions the original data references should be provided. A similar point was raised by referee #1 for the d11B compilation by Rae et al. Again, provide the full original references at least in the figure captions. Note also that in several figures you refer to the ice core data as Bereiter et al, but this data set only covers the last 800 kyr. Earlier ice core data comes from blue ice samples. Please correct accordingly and provide the correct references (see also comments on Figures below)

**Response 4:** We changed "Vostok" to "ice core" in the main text. We also changed in the Fig. 3 legend, and axis and acknowledge the original publications. We also added the references from Higgins et al. (2015) and Yan et al. (2019) when those data were presented.

- in line 118 you refer to an air sea flux of xx ppm, this is not a flux but an air sea difference

**Response 5:** We changed for air sea difference. Line 114: "annual air-sea $CO_2$ difference of +28 ppmv (Takahashi et al., 2014). The pre-industrial air-sea $CO_2$ difference is calculated to be +16 ppm, (GLODAP database corrected from anthropogenic inputs),"

- line 172: there is something wrong with this sentence

**Response 6:** We changed line 166: "Further, despite different sedimentation rates, our $\delta^{11}B$ and Mg/Ca results are consistent between Sites 806 and 807 , and with data from Site 872 (Sosdian et al., 2018), which implies that diagenesis is not a primary driver of the reconstructed trends. Comparison of raw data, and derived parameters, is shown in Figs. 2 and 7.

- throughout the manuscript add a space between numbers and units

**Response 7:** We changed this through the text.

- line 411. Here you refer to a pCO2 peak at 9 Myr in Sosdian. I couldn't find this point, maybe refer to the age more precisely?

**Response 8:** Sosdian et al. (2018) reported a peak at ~9 Ma, this peak is discrete and relatively small in comparison the MCO $CO_2$ concentration.

In section 3.4 from Sosdian et al. (2018): «Regardless of the chosen $\delta^{11}B_{sw}$ scenario, all our $CO_2$ reconstructions shows $CO_2$ peak at ~9 Ma that is not seen in the climate records (Fig. 5). This peak in CO2 at ~9 Ma is also seen in recent cell-sized corrected alkenones and pennate diatoms (diffusive and ACTI-CO models) $CO_2$ records (Bolton et al., 2016; Mejía et al., 2017; Supplementary Fig. S12). »

- line 517: there is something wrong with this sentence

**Response 9:** We changed from "Before 18.5 Ma, the pCO$_2$ is relatively stable, $\delta^7Li$ is increasing representative of a non-steady state weathering.  From 18.6 to 16.7 the $\delta^7Li$ decrease of about 2 ‰, this decrease can inform on decreasing weathering rate and this decrease is associated with an increase in pCO$_2$." to line 540: "Before 18.5 Ma, the pCO$_2$ is relatively stable, $\delta^7Li$ is increasing, suggesting non-steady state / incongruent nature of continental chemical weathering.  From 18.6 to 16.7 Ma, the $\delta^7Li$ record decreases by ~2 ‰, consistent with decreasing weathering rates and an associated increase in pCO$_2$."

Figure 2: There are brown triangles in figure 2B and a cross in figure 2C but not in the legend

**Response 10:** Brown triangles are data from *G. ruber*. It is in the figure caption from Fig. 2 "brown open symbols are for *G. ruber*."

Figure 3: see comments on Figure 3 above and in review #1. Please delete the subscript Rae et al in the legend and cite the data and compilation papers in the caption

**Response 11:** We are now calling in the legend the compilation from Sosdian et al. (2018) "compilation A" and the one from Rae et al. (2021) "Compilation B", references from those compilations are now added in the caption."

Figure 4: there are orange circles in this figure but not in the legend

**Response 12:** Caption of Figure 4: "Orange open circles are SST data calculated with our framework from the species *D. altispera* at ODP Site 806 (Sosdian et al., 2020) with an offset of +8°C."

Figure 5: Please delete the subscript to individual papers in the legend and cite the data and compilation papers in the caption

Figure 6: Please delete the subscript to individual papers in the legend and cite the data and compilation papers in the caption. There are bold and thin triangles/squares in the figure but not in the legend.

**Responses 13 and 14:** We added the original references in the figure legend for compilation A (Sosdian et al., 2018) and compilation B (Rae et al., 2021).

Figure 7: Please delete the subscript to individual papers in the legend and cite the data and compilation papers in the caption. There is no legend for the SST and d11B-pH subplots

**Response 15:** We are now calling in the legend the compilation from Sosdian et al. (2018) "compilation A" and the one from Rae et al. (2021) "compilation B" in the legend, references are described in the caption.

Figure 8: Please delete the subscript to individual papers in the legend and cite the data and compilation papers in the caption. There is no legend for the SST subplot

Figure 9: Please delete the subscript to individual papers in the legend and cite the data and compilation papers in the caption. There is no legend for the SST subplot

Figure 10: Please delete the subscript to individual papers in the legend and cite the data and compilation papers in the caption. There is no legend for the SST subplot

Figure 11: Please delete the subscript to individual papers in the legend and cite the data and compilation papers in the caption.s.

**Response 16, 17, 18 and 19:** We deleted the subscript from those figures and added legends for temperature.

**Reviewer 1**

Overall I would like to praise the authors for the careful revision of the figures which has really clarified the presentation of the new data and its comparison with previously published records. The text now reads more clearly in many sections as well. There are a few final issues I would propose the authors clarify, and some proofing issues that I identify.

The text at the end of methods (lines 242-243) summarizes the processing of Mg/Ca to SST, noting that it is corrected for influences of pH and salinity. However, additionally, the SST has been corrected for inferred changes in the Mg/Ca of seawater (detailed in the supplement). This point also needs to be mentioned in lines 242-244 of the main text, so the reader is aware of it, and is directed to the supplement for further details.

**Response 20:** We added line 243: "Detailed calculations can be found in the supplemental materials. Briefly, Mg/Ca was used to reconstruct sea surface temperature (SST) using the framework from Gray and Evans. (2019) correcting for influences of pH, salinity and secular variation in seawater Mg/Ca.".

The figures include detailed dashed lines to show correspondance to specific events such as MMCT defined from the benthic d18O stack of Zachos 2008 (and in younger part Lisieki-Raymo). I would suggest also including the low resolution benthic 18O from Lear et al 2015 from 806, to help document the comparison with these events within the same site, because the benthic 18O at 806 and the d11B-based CO2 and SST can therefore be compared within 806 without the complication of uncertainty in age models due to the normal limitations of precision in biostrat.

**Response 21:** We added the benthic $\delta^{18}O$ from Lear et al. (2003) and (2015), reprocessed in 2020. Even if lower in resolution the record is in line with the benthic compilation from Lisiecki and Raymo, (2005) and Zachos et al. (2008) on the different timescales studied here which supports our age model for 806 and our interpretations (Figure S3). The linear regressions for the cross plots Figs. 3C and D are however still non-significant.

line 406 makes a rather broad statement about the shift in $\delta^{13}C$ and $\delta^{18}O$ over the MMCT supporting drawdown of $CO_2$ by organic carbon burial, with no reference. As these benthic data were previously published and interpreted, and this coincidence and organic burial hypothesis advanced long ago, I would suggest referencing a number of papers which have produced the benthic $\delta^{13}C$ and $\delta^{18}O$ results and originally provided this interpretation, to support your link of organic carbon burial with $CO_2$ drawdown. The recent discussion paper in Climate of the Past by Raitzsch et al. provides detail over this interval and may be worth citing as well.

**Response: 22:** We are now referencing a couple of relevant papers associated with the middle Miocene climate transition. We also added the $pCO_2$ from Raitzsch et al. (2021) in Figure 8. Because our resolution is low we stress to overinterpret our data for this transition however we discuss finding from Raitzsch. (2021).

Line 403: "The synchronous shifts in the $\delta^{13}C$ and $\delta^{18}O$ of benthic foraminifera are consistent with increased carbon burial during colder periods, thus feeding back into decreasing atmospheric $CO_2$, and supporting the hypothesis that the drawdown of atmospheric $CO_2$ can in part, be explained by enhanced export of organic carbon (Flower and Kennett, 1993, 1996). However, given the limited sampling of this study, we are only able to resolve a $pCO_2$ decrease toward the end of the MMCT (~13.5 Ma). The higher resolution $\delta^{11}B$-$pCO_2$ from Site 1092 for the MMCT (Raitzsch et al. 2021) reports eccentricity-scale $pCO_2$ variability; the authors reported that low $pCO_2$ during eccentricity maxima was consistent with an increase in weathering due to strengthened monsoonal circulation, which would increase nutrient delivery and supporting higher productivity that in turn would impact carbon drawdown and burial, in line with modeling from Ma et al. (2011)."

The section 3.6 makes a few general statements that could be better balanced with context and caveats. The linking of the MCO pCO2 increase with the Columbia River is interesting but I think it is better balanced to comment that this is superimposed on temporal variation of the large volcanic CO2 flux from seafloor production, which has been estimated by recent tectonic models with increasing precision (eg Muller et al 2016). Likewise the estimated volume and CO2 flux from the Columbia River Basalts is very modest compared to other LIP in compilations (Courtillot et al 2003), suggesting

that if it is contributing to high CO2 at the MCO, a better understanding of eruptive volume and volatile emissions is needed.

Likewise a bit of clarification about the d7Li is required - eg. note that the authors favored alkalinity model is Caves et al, but Caves is discussing specifically why d7Li is not necessarily a proxy for silicate weathering flux. I agree that it is interesting to note the coincidence of inflections in the d7Li curve and features of the pCO2 record, but I find the current lines 508-524 have gone too far in oversimplifying the d7Li and a few sentences to provide better context to the interpretation of d7Li and debates would make this final section much more useful.

**Response 23 and 24:** Thank you for those references and these comments. We incorporated your comments in section 3.6.

Line 498: "On million-year timescales, atmospheric $CO_2$ is controlled by its input through mantle degassing in the form of sub-aerial and sub-aqueous volcanic activity and its removal by chemical weathering of continental silicate rocks. Over the last 16 Myr, two relative maxima in atmospheric $pCO_2$ are observed in our record, one during the MCO (at 15.67 Ma) and a second around the late Miocene/early Pliocene (beginning at 4.7 and 4.5 Ma) (Fig. 11), though the timing for the latter is not precise. The strong $pCO_2$ increase from the early Miocene to MCO is timely with increasing volcanic activity (Foster al. 2012), associated with the eruption of the Columbia River Flood Basalts (Hooper et al., 2002; Kasbohm and Schoene, 2018), with recent geochronologic evidence published supporting higher eruption activity between 16.7 and 15.9 Ma (Kasbohm and Schoene, 2018) reinforcing the idea of an episodic $pCO_2$ increase during the MCO due to volcanic activity. Underestimation of net $CO_2$ outgassing from specific continental flood basalt eruption is possible, as both sub-aqueous and sub-aerial flood basalts, under right climatic conditions, are prone to enhanced chemical weathering. For example, the 4-5‰ drop in $\delta^7Li$ record at the K-Pg boundary (Misra and Froelich, 2012) is attributed to rapid quasi-congruent weathering of Deccan Traps (Rene et al. 2015) during their eruption. Courtillot and Rene (2003) estimates about 50% of emitted $CO_2$, roughly equivalent to the amount emitted by the eruption of a million cubic kilometers of Deccan Traps, is missing due to chemical and physical weathering. Additionally, the early Eocene (at ~50 Ma) 3-4‰ rise in seawater $\delta^7Li$ at a time where there is not significant uplift of the Himalayas (Misra & Froelich, 2012) is also attributed to incongruent weathering of previously erupted Deccan Trap basalts as the Indian subcontinent moved from arid mid-latitudes to the wet low latitudes (Kent and Muttoni, 2008, PNAS). Thus, a significant part of the outgassed $CO_2$ can be consumed by chemical weathering of freshly erupted hot basalts (Courtillot et al., 2003). However, the congruency of chemical weathering of basalts, depending on regional climatic conditions (warm-wet vs. cold-arid), will determine the nature of observable inflection in the seawater $\delta^7Li$. The possible quantification of increased rates of silicate weathering inferred from $\delta^7Li$ (mentioned below) can be utilized to determine total eruptive volume (missing + existing) and volatile emissions from the Columbia River Flood Basalts. At the same time as continental flood basalt emissions, enhanced seafloor production could also be a second possible source of $CO_2$; however, we note there is evidence that the rate of seafloor production has remained virtually invariant over the last 60 million years (Rowley, 2002; Muller et al. 2016)."

Detail: Lear 2015 is missing in the reference list (it is cited in text line 179).

**Response 25:** We added the reference to the list.
Detail: Supplemental Information (there is no "s" needed at the end of "Information")

**Response 26:** We removed the "s".
Detail: the citation of Tanner et al 2020 in line 360 does not make sense as that sentence refers to MMCT and Tanner et al cover only the Late Miocene after 8 Ma. The citation in line 414 is appropriate although the proposed reason for differences is unclear attributing it to " the oceanographic setting of 1088" without indicating what process about the oceanographic setting is proposed to explain the difference.

**Response 27:** We removed the citation of Tanner from line 360.

Line 421: "$pCO_2$ differences between our reconstruction and that of Sosdian et al. (2018) and Raitzcsh et al. (2021) (Fig. 8) likely reflect assumptions made for calculations (of $\delta^{11}B$, TA) and the specific

mono-specific calibrations used for each study, as well as potential geographic differences in air-sea $pCO_2$. These differences do not invalidate the boron isotope proxy but illustrate the impact that specific seawater parameters and calibrations can have on reconstructed $pCO_2$ values."

Detail - line 436 either "during" or "from" - one of these words needs deleting

**Response 28:** We changed line 436 for: "We calculate high $pCO_2$ values of $419 \pm 119$ ppm (2 SD, n=3, Table 2) between 4.7 to 4.5 Ma during the Early Pliocene warm interval (Figure 9)."
Detail - line 483-484, use consistent formatting for the +/- values of $CO_2$.

**Response 29:** We removed the parentheses to be consistent with the rest of the manuscript.
Raitzsch, Markus, Jelle Bijma, Torsten Bickert, Michael Schulz, Ann Holbourn, and Michal Kučera. "Atmospheric carbon dioxide variations across the middle Miocene climate transition." Climate of the Past 17, no. 2 (2021): 703-719.
Müller, R. Dietmar, Maria Seton, Sabin Zahirovic, Simon E. Williams, Kara J. Matthews, Nicky M. Wright, Grace E. Shephard et al. "Ocean basin evolution and global-scale plate reorganization events since Pangea breakup." Annual Review of Earth and Planetary Sciences 44 (2016): 107-138.
Courtillot, Vincent E., and Paul R. Renne. "On the ages of flood basalt events." Comptes Rendus Geoscience 335, no. 1 (2003): 113-140.

**Reviewer 2**

Review for 'Atmospheric CO2 estimates for the Miocene to Pleistocene based on foraminiferal δ11B at Ocean Drilling Program Sites 806 and 807 in the Western Equatorial Pacific'

The authors have done a good job at dealing with the previous round of reviewers comments on this manuscript. The discussion and conclusions are more nuanced and more appropriate for the data as presented in this iteration. I am pleased to see that most, if not all the suggestions from the reviewers have been taken on board as they were important for the accuracy and longevity of this paper.

However, I have one new major concern with the manuscript that requires addressing prior to publication, alongside several minor points which are listed below. My major concern is with the implications from the cross plot presented in figure 3 ( in particular E), these were added as part of the revision which is great, but they are not represented well in the text and the data need a significant reassessment. The $r^2$ of the 'validation' dataset presented in 3E is 0.09 and there is a p-value of .3 which is nonsignificant and cannot be used to reject the nullhypothesis.

I understand the frustration of such figures, but they should be presented in the main test where the plot is mentioned and their ramifications discussed. Numerics like this would typically present difficulty for the rest of the study, but in this case I think there are some alternate options as I do agree that the data are probably better than these performed statistics would imply.The authors should investigate other methods for assessing their ice core overlapping data (e.g. root mean square error etc.), perhaps focussing on identifying and removing outliers and further exploring the knock-on effects on the rest of the record. I.e. given the prediction error envelope (rather than just a fit error or an analytical and calculation error), what effect does that have on the Pliocene and Miocene $CO_2$ reconstructions?

**Response 30:** We replied in a previous comment, but we used also a Deming regression to take into account the both x and y uncertainties (Fig. 3C). The statistics do not improve (p=0.25). To make sure we didn't have a gap due to the age models we are now plotting in Fig. 3 the benthic record from site 807 (Zhang et al., 2007) and from 806 (Lear et al., 2003, 2015, 2020). Those records are consistent with each other which exclude an issue with the age models (Fig. S3). When plotted together, data from Chalk et al. (2017) and Hönisch et al. (2009), we can see in that our data present more variability (Fig. S4A), this variability doesn't seem characteristic of one site (Fig. S4A) but could be due to the reconstruction itself (weight/shell correction). We still are able to reproduce the interglacial/glacial variability which is encouraging. We do not think those results impact our paper which focus on a low resolution $pCO_2$ across the Neogene but a high resolution IG/G cycle would be needed to identify if the variability is truly associated with the reconstruction or if it representative of a localized $CO_2$ change and this is something we will probably explore in the future.

Is there an impact from using the Mg/Ca corrected script of Gray and Evans or the calibrations used (e.g. Raitzsch 2018 vs. Henehan 2013)? This is really crucial for our community understanding and confidence in the proxy data.

**Response 31:** The sensitivity of $\delta^{11}B_{carbonate}$ to $\delta^{11}B_{borate}$ is really similar (0.82 ± 0.22) reprocessed by Henehan et al. (2016), (0.80 ± 0.17) for Raitzsch et al. (2018) and (0.83 ± 0.44) here from (Guillermic et al., 2020). The Mg/Ca from Gray and Evans does not play a significant role in our record because only *G. ruber* is impacted by pH not *T. sacculifer* (no evidence for now) from which most of the record is derived.

For the Miocene, alkalinity is actually of main importance but also $d^{11}B_{sw}$. Rae et al. (2021) used a constant alkalinity and an average of the 3 seawater $\delta^{11}B$ scenarios.

I am confused as to the point of the other two cross plots in figure 3, they are all mentioned in the test together but with little explanation as to why or what they show. The layout of the axes is also confusing, why is the same parameter plotted on the x axis in 3C and the y axis in 3d?

**Response 32:** We removed those cross-plots as they were not adding information to the manuscript, instead we present other cross-plots in Figure S4 to add information on the differences between ice core and boron isotopes derived $pCO_2$.

In the text (line 275) it is mentioned that only 2 of the data points in the validation dataset (n=16) are outside the uncertainty, this does not appear to be the case with the whole record in figure 3B or in 3E.

**Response 33:** We removed this sentence as the comparison with ice core data is better illustrated with the Figs. 3C and S4.

Minor points:

Line 33-35: There is a lot of detail here regarding the MPT in the abstract which is still not for me the major finding or thrust of the paper, would suggest slimming this down.

**Response 34:** Line 32 "During the Mid-Pleistocene Transition there is a minimum in $pCO_2$ at MIS 30."

Line 42: some formatting inconsistencies throughout with italics on pCO2.

**Response 35:** We changed for "$pCO_2$" and it is now consistent through the text.

Line 49: Would 'Neogene' be a better key word than 'Miocene'

**Response 36:** We provide new data until the past 16 Myrs even if the Miocene is included in the Neogene, we don't want the reader to think we provide measurement for the early Miocene.

Line 70: In reference list like this which are incomplete add 'e.g.' in front.

**Response 37:** We added a couple more recent references in order to complete the list, we still put the 'e.g.' in front of the references.

Line 90: The TE part of the acronym TE-NTIMS is not defined.

**Response 38:** Instead of TE-NTIMS (Ni et al., 2010) we changed for N-TIMS as the $pCO_2$ reconstructions are based on N-TIMS or MC-ICP-MS data. We changed line 74 for "N-TIMS" instead of "TIMS" and Line 85: "The marine CO2 proxy that appears to be subject to the fewest systematic uncertainties, based on our current understanding, is the boron isotopic composition ($\delta^{11}B$) of planktic foraminifera as measured using MC-ICP-MS and N-TIMS (Hain et al., 2018)."

Line 95: or if the disequilibrium is known. Not many of the sites used so far are in perfect

equilibrium, but the stability is important.

**Response 39:** We added line 91 "Atmospheric $pCO_2$ can then be constrained if the site being examined is in air-sea $CO_2$ equilibrium or if the disequilibrium is known and stable through time.".

Line 121: Is this 17ppm 1SD or 2SD?

**Response 40:** It is 2SD, line 117: "an average of ~17 ppm (2 SD) for the youngest samples".

Line 138: insert here if you applied the pH dependency.

**Response 41:** Line 132: "For temperature estimation, we utilize a multi-variable model for Mg/Ca correcting from salinity, pH and seawater Mg/Ca (Gray and Evans, 2019) […]".

Line 144: Hopefully all the figures are of merit, suggest changing to 'principal figures'.

**Response 42:** We changed for "principal figures".

Lines 176-182: In the discussion of age models it is apparent that the two sites have very different qualities of age model, what are the implications of this for age uncertainty?

**Response 43:** To explore this, we now provide in Figure 3A the comparison between benthic $\delta^{18}O$ from site 806 and 807. Both age models are based on biostratigraphy but are in line with each other (only 0.55 Myr constrained) and the stack from Lisicki and Raymo, 2005.

Line 226: Was the JCp analysed clean or uncleaned?

**Response 44:** The JCp-1 was uncleaned.

Line 248: is the capital needed in Alkalinity?

**Response 45:** we removed the capital.

Line 257: I am not clear precisely what time-adjacent means here, please be specific.

**Response 46:** We changed for line 250: "Although the record we generated does not overlap with Site 872, they are 1 myr apart (15.7 and 16.7 Ma); there is a good correspondence between our Mg/Ca data and the published Mg/Ca record from *T. trilobus* at Site 872 (Sosdian et al., 2018). Mg/Ca from a different species, *D. altispira* (Sosdian et al., 2020), is also plotted with an offset, for comparison."

Line 264-265: This is a very important point, why can they be lower and yet produce similar estimates of $CO_2$? What is inside the calculation that yields this result? (2nd carbonate parameter?)

**Response 47:** There is a strong size-dependence at site 806 for $\delta^{11}B$, reported in Hönisch and Hemming, 2004 and Ni et al., 2007, this is why we applied a size (weigh/shell)-$\delta^{11}B$ correction in order to adjust the intercept of the calibration for each individual and correct from this effect. This is a parameter that is not propagated in the error.

Line 270: n=16 here.

**Response 48:** We added line 263: "for the last 800 kyr (n=16, Fig. 3)."

Lines 281: missing space between reconstructions and in

**Response 49:** We added the space.

Line 312: odd bold formatting.

**Response 50:** We fixed this.

Line 339: Specify the subplot of Fig 6.

**Response 51:** We added the subplots Figs. 6B and 6E.

Line 344: Please point to the plot here. Greenop et al. 2017 (e.g. plot 6E)

**Response 52:** We added the subplot 6E. Line 342: "by Greenop et al. (2017) (e.g. Fig. 6E)."

Line 376-379: What is the point made here, who are we supposed to believe? Please detail the differences. The difference in the calculation of two papers from the same year is huge here >400ppm, and of crucial importance for the viability of the proxy.

**Response 53:** Rae et al., 2021 used a constant alkalinity + an average of the three $\delta^{11}B_{sw.}$

This study used the alkalinity from Caves et al. (2016) + $\delta^{11}B_{sw}$ from Greenop et al. (2017).

Even if depending on the timescale studied the assumptions are reasonable between studies, difference in absolute values can be important.

Line 375: "because of the different assumptions used in their calculations. This difference is important because the assumptions from Rae et al. (2021) would imply a relatively high and stable $pCO_2$ from the early Miocene to MCO (Fig. S2), which would imply a decoupling between $pCO_2$ and temperature with no $pCO_2$ change during an interval of decreasing benthic $\delta^{18}O$. However, our reconstructed $pCO_2$ increase towards the MCO is in line with the observed benthic $\delta^{18}O$ decrease and $\delta^{13}C$ increase and suggest a coupling between temperature and $pCO_2$ over this period. This highlights the critical need for the use of a common set of assumptions for studies. Assumptions may vary between studies depending of the timescales studied, but a common framework is needed. In addition, further constraints on the second carbonate system parameter and on secular changes in seawater $\delta^{11}B$ will reduce uncertainties in reconstructed $pCO_2$, with improved precision."

Line: 461: MPWP format

**Response 54:** We changed for "mPWP".

Line 468: i.e.., format

**Response 55:** We changed for :" (i.e, ~100 kyr)".

Line 488: clarify ice sheet stability over multiple obliquity cycles, or just large sheet generation. They are still not inherently stable.

**Response 56:** Line 494: "In our record for the last 16 Myr, the lowest $pCO_2$ is recorded at MIS 30 during the MPT, with values of 164 ($\pm^{44}_{35}$) ppm, which supports an atmospheric $CO_2$ threshold that leads to large sheet generation. During this transition, the $pCO_2$ threshold needed to build sufficiently large ice sheets that were able to survive the critical orbital phase of rising obliquity to ultimately switch to a 100 kyr world, was likely reached at MIS 30, but a higher $pCO_2$ resolution of the MPT is needed for confirmation."

Line 529: reprocessing data?

**Response 57:** We changed for "reprocessing".

Line 542: specify the MCO record produced in this study. Also another line of reasoning is required to explain the difference seen between sites, that cannot be due to the basalts unless you are implying a local effect/ teleconnection

**Response 58**: We developed this section following comment from reviewer 1. I guess our perspective on this is that variability between Sites are obscured by the assumptions of the reconstruction themselves. However, reprocessing the data with a common framework as expected from the boron workshop may highlight significant differences between Sites.

Line 553: typically ESS or ECS not earth system climate sensitivity.

**Response 59:** We changed for "Earth climate sensitivity".

Line 817: format

**Response 60:** We fixed the format of the reference.

Figures and captions:

Figure 1: Contours would be good to add to this plot, the rainbow colour bar is not good for colour blind accessibility.

**Response 61:** Figure 1 now have contours.

Figure 3: It is unclear to me what 3C and 3D are supposed to show and why they have been plotted. More explanation required as above. Also it would be easier to see and trends if the δ18O ruber were plotted on the same axis in both figures.

**Response 62:** Instead different of those figures we are now presenting other cross-plots to add information on the comparison between ice core and boron based $pCO_2$ (Figure S4).

Figure 4: What is orange?

**Response 63:** We added to the legend and details in the caption of Fig. 4, orange open circles are the reprocessed data from Mg/Ca of *D. altispera* from Sosdian et al. (2020) following our framework, those date are however plotted with an offset (8°C) calculated to match overlapping data from *T. sacculifer* Site 806 (this study).

Figure 6: The ice core reference is not correct.

**Response 64:** We changed Bereiter et al. 2015 for "Yan et al. 2019".

Figure 7: The site 872 data is calculated by Rae et al. 2021 not from it.

**Response 65:** We changed for: "asterix symbols are calculated $pCO_2$ at site 872 by Rae et al. (2021)."

Figure 9: Again ice core reference is not correct.

**Response 66:** We changed Bereiter et al. 2015 for "Yan et al. 2019".

Figure 10: Rae's compilation is cited throughout these figures and while I appreciate it is difficult to cite all the original references on the longer timescale plots, I think it is unfair to not credit the original authors on any of these plots. None of the data is created by Rae et al. e.g.

**Response 67:** We agree, we now present the original references in the different captions. "Data for compilation A are from: Hönisch and Hemming, 2009; Seki et al., 2010;  Foster et al., 2012; Badger et al., 2013; Greenop et al., 2014; Martinez-Boti et al., 2015a; Chalk et al., 2017; Sosdian et al., 2018. Data for compilation B are from: Foster et al., 2008; Hönisch and Hemming, 2009; Seki et al., 2010; Foster et al., 2012;  Badger et al., 2013; Greenop et al., 2014; Martinez-Boti et al., 2015a; Chalk et al., 2017; Dyez et al., 2018; Sosdian et al., 2018; Greenop et al., 2019; de la Vega et al., 2020."

Figure 10 should credit Bereiter, Hoenisch, Chalk, Dyez, and Yan studies.

**Response 68:** We now cite all these references.

---

## Author Response (AR3)

**UNIVERSITY OF CALIFORNIA, LOS ANGELES**

[Figure]

**UCLA**

BERKELEY • DAVIS • IRVINE • LOS ANGELES • RIVERSIDE • SAN DIEGO • SAN FRANCISCO

SANTA BARBARA • SANTA CRUZ

DEPARTMENT OF EARTH AND SPACE SCIENCES
3806 GEOLOGY BUILDING
BOX 951567
LOS ANGELES, CALIFORNIA 90095-1567
TEL: (310) 825-3880
FAX: (310) 825-2779

Dear Dr. Fischer,

We thank you for your constructive comments on this manuscript. Please find below the point-to-point response.

On behalf of the co-authors,
Maxence Guillermic and Aradhna Tripati

Dear authors

thanks for all the changes. I think this has greatly improved the paper and made the discussion much more differentiated. I had another read of the paper and found a list of language issues that I would ask you to fix (see list below). Please also check the quality/resolution of your figures. At least Figure 5 is very bad resolution in my pdf.

Please send a short list with the changes you did together with the revised version.

All the best Hubertus

Language edits (please note that the line numbers below refer to your new "Authors tracked changes" version)
line 36: "from the Miocene to the late Quaternary"
**Author's response:** This change has been made.

line 59: I would suggest to delete: "from regions...atmosphere."
**Author's response:** This change has been made.

line 64: "into older atmospheric"
**Author's response:** This change has been made.

line 65: "of CO2 prior to"
**Author's response:** This change has been made.

line 82-85: there is something wrong with this sentence, please fix
**Author's response:** We changed this to: "There was also the realization that temperature-dependent $K_D$ and B/Ca sensitivities reported from sediment trap, core-top, and downcore studies (Yu and Hönisch, 2007; Foster et al., 2008; Tripati et al., 2009, 2011; Babila et al., 2010; Osborne et al., 2020) differ from inferences from foraminiferal culture experiments (Allen et al., 2011, 2012) and inorganic calcite (Mavromatis et al., 2015) which complicates the use of the B/Ca proxy, although this type of discrepancy has also been observed with other elemental proxies (e.g., Mg/Ca). »

line 107: "Moreover, although"
**Author's response:** This change has been made.

line 118: "open ocean region"
**Author's response:** This change has been made.

line 133: "support stable air-sea (dis-)equilibrium conditions"
**Author's response:** This change has been made.

line 135: "on prior low-resolution recosntructions"
**Author's response:** This change has been made.

line 143: "correcting for"
**Author's response:** This change has been made.

line 146: "recent times"

**Author's response:** This change has been made.

line 182: "A comparison of"
**Author's response:** This change has been made.

line 233: "on 11B"
**Author's response:** This change has been made.

line 259: "using d11Bborate"
**Author's response:** This change has been made.

line 262: "are provided in"
**Author's response:** This change has been made.

line 269: "1 Myr"
**Author's response:** This change has been made.

line 289: "are met despite large scatter"
**Author's response:** This change has been made.

line 291-92: unclear, clarify this sentence especially on bootstrapping
**Author's response:** This has been changed to: "Two regressions between ice core $pCO_2$ and boron-based $pCO_2$ are shown, a simple linear regression (grey line) and a Deming regression that takes into account error in variables (blue line). Bootstrapping was used to calculate uncertainties in the regression models (n=1000, Figure 3C, Table S6). »

line 295-96: I do not agree or misunderstand: The age scales for the marine records may be consistent with each other but they are independent of the ice core age scale. Please clarify sentence.
**Author's response:** This now reads: « The age models for the sites are based on comparisons of the benthic $\delta^{18}O$ records for both Sites 806 and 807 (Fig. 3A, Zhang et al., 2007 ; Lear et al., 2003 ; Lear et al., 2015) to the published isotopic stack (Lisiecki and Raymo, 2004). »

line 319: "were reconstructed"
**Author's response:** This change has been made.

line 324: ", respectively"
**Author's response:** This change has been made.

line 326: "small (ca. 1°C) change into"
**Author's response:** This change has been made.

line 339: "used in our calculations"
**Author's response:** This change has been made.

line 340: "are provided in"
**Author's response:** This change has been made.

line 382: the part "in concert with ocean circulation " is unclear, clarify

**Author's response:** We removed this part as we intend to make a more general statement. We changed it to: "The study of Miocene climate is thought to provide insights into drivers and impacts of global warming and melting of polar ice (Flower and Kennett, 1994).".

line 399: "suggest that the"
**Author's response:** This change has been made.

line 413: "pCO2 data increase"
**Author's response:** This change has been made.

line 420: "are found during"
**Author's response:** This change has been made.

line 431: "of pH of"
**Author's response:** This change has been made.

line 461: " "Raitzsch"
**Author's response:** This change has been made.

line 485/86: "regions from 3.3. to 2.7 Ma"
**Author's response:** This change has been made.

line 518: it is debated whether eccentricity directly plays a role. May be write "high amplitude, lower frequency (app. 100 kyr) cycles"
**Author's response:** This change has been made.

line 563: explain K-Pg boundary
**Author's response:** We added: "For example, the 4-5‰ drop in $\delta^7Li$ record at the Cretaceous–Paleogene (K-Pg) boundary (Misra and Froelich, 2012) […]"

line 573-74: fix/clarify sentence "will determine the nature... seawater d7Li"
**Author's response:** We changed this to: "However, the congruency of chemical weathering of basalts, depending on regional climatic conditions (warm-wet vs. cold-arid), will determine the shape and position of inflection points in the seawater $\delta^7Li$ record. "

line 582: "by a change"
**Author's response:** This change has been made.

line 585: "increasing seawater"
**Author's response:** This change has been made.

line 587: "MCO is synchronous"
**Author's response:** This change has been made.

line 619-20: fix sentence: "and the MCO values...higher pCO2"
**Author's response:** We changed Line 621 to: "$pCO_2$ values increase from the early Miocene to the MCO with estimated MCO $pCO_2$ values of $511 \pm 201$ ppm (2 SD, n=3)."

line 621: replace "likely" by "potentially" ???
**Author's response:** This change has been made.

line 626: "for the time interval app. 3.3-3.0 Ma"
**Author's response:** This change has been made.

line 627: " for the Pliocene"
**Author's response:** This change has been made.

line 1072: "corrected for"
**Author's response:** This change has been made.

line 1075: "to recent times"
**Author's response:** This change has been made.

line 1078: "squares represent data"
**Author's response:** This change has been made.

line 1094: "pCO2" SUBSCRIPT "ice core"
**Author's response:** This change has been made.

line 1096: "bootstrapping"
**Author's response:** We changed this sentence, please see edits below.

line 1097-98: fix/clarify sentence*
Line 1101: The sentence now reads "Cross plot for the last 0.8 Myr of $pCO_{2\,\delta11B}$ from this study and $pCO_{2\,ice\,core}$ (from ice core compilation, Bereiter et al., 2015), grey line is a simple linear regression (p = 0.25, $R^2$=0.09), blue line is a Deming regression taking both x and y uncertainties into account (p = 0.25). Details of the regression parameters are in Table S6. Ice core $CO_2$ error was calculated based on 2 SD of reported values, and $\pm$ 1 ky for the age of sediment samples. Boron-based $pCO_2$ error is calculated based on error propagation described by eq. S17."

line 1102: "temperatures were"
**Author's response:** This change has been made.

line 1133: "from the compilation by Rae"
**Author's response:** This change has been made.

line 1135: "using the constant"
**Author's response:** This change has been made.

line 1152: "color is indicating"
**Author's response:** We changed to "color indicates"

line 1156: "temperatures"
**Author's response:** This change has been made.

line 1159: "are compilation data from Rae"
**Author's response:** This change has been made.

line 1163: "pCO2 values"

**Author's response:** This change has been made.

line 1164: "pCO2 values"
**Author's response:** This change has been made.

line 1168: "color is indicating"
**Author's response:** This change has been made.

line 1172: "temperatures"
**Author's response:** This change has been made.

line 1177: " pCO2 values"
**Author's response:** This change has been made.

line 1190: "color is indicating"
**Author's response:** This has been changed to "color indicates"

line 1194: "temperatures"
**Author's response:** This change has been made.

line 1198: "represent the CO2 compilation"
**Author's response:** This change has been made.

line 1199: "from the compilation by Rae"
**Author's response:** This change has been made.

line 1209: "color is indicating"
**Author's response:** This has been changed to "color indicates"

line 1213: " temperatures based on literature"
**Author's response:** This change has been made.

line 1217: "from the compilation by Rae"
**Author's response:** This change has been made.

line 1228: "are indications when"
**Author's response:** This has been changed to "indicates"

line 1230: "color is indicating"
**Author's response:** This has been changed to "color indicates"

line 1236: "shown is the timing of"
**Author's response:** This change has been made.

line 1237: "the eruption of"
**Author's response:** This change has been made.